# Sea level much higher than assumed in most coastal hazard assessments

Katharina Seeger[1,2,3] ✉ & Philip S. J. Minderhoud[1,3,4] ✉

The impacts of sea-level rise and other hazards on the coasts of the world are determined by coastal sea-level height and land elevation[1]. Correct integration of both aspects is fundamental for reliable sea-level rise and coastal hazard impact assessments[2,3], but is often not carefully considered or properly performed. Here we show that more than 99% of the evaluated impact assessments handled sea-level and land elevation data inadequately, thereby misjudging sea level relative to coastal elevation. Based on our literature evaluation, 90% of the hazard assessments assume coastal sea levels based on geoid models, rather than using actual sea-level measurements. Our meta-analyses on global scale show that measured coastal sea level is higher than assumed in most hazard assessments (mean offsets [standard deviation] of 0.27 m [0.76 m] and 0.24 m [0.52 m] for two commonly-used geoids). Regionally, predominantly in the Global South, measured mean sea level can be more than 1 m above global geoids, with the largest differences in the Indo-Pacific. Compared with geoid-based assumptions of coastal sea level, the measured values suggest that with a hypothetical 1 m of relative sea-level rise, 31–37% more land and 48–68% more people (increasing estimates to 77–132 million) would fall below sea level. Our results highlight the need for re-evaluation of existing coastal impact assessments and improvement of research community standards, with possible implications for policymakers, climate finance and coastal adaptation.

Sea-level rise (SLR) poses a high risk to vast coastal lowlands around the world, including low-elevated and populous river deltas and coastal plains. According to the Intergovernmental Panel on Climate Change (IPCC) assessment report 6 (AR6) (shared socioeconomic pathway (SSP)1-1.9 to SSP5-8.5)[4], global mean sea level (MSL) is projected to rise between 0.28 m and 1.01 m by 2100 compared with that in 1995–2014. Including deep uncertainties in the polar ice-sheet dynamics, these projections could increase by several metres[4]. This rise is amplified by negative vertical land motion, that is, land subsidence, a natural but increasingly human-accelerated phenomenon in coastal lowlands[5,6], which drives higher rates of relative sea-level rise (RSLR)[7,8]. Consequently, the impacts of RSLR and coastal (and compound) flooding are closely related to coastal elevation relative to sea level. Assessing coastal exposure and vulnerability consequently requires the use of elevation information, commonly provided by digital elevation models (DEMs). Constituting the fundamental base of any such impact and exposure assessment, the quality (mainly vertical accuracy and spatial resolution) of the DEMs is fundamental to the accuracy and reliability of the derivatives and widely addressed in scientific literature (see, for example, ref. 9).

Although high-quality elevation information, for example, acquired through airborne lidar, is available in several regions of the Global North (see, for example, ref. 10), the best available elevation data for the vast majority of the coastal areas worldwide is satellite-based. The global availability of satellite-based elevation data enabled unprecedented (global) SLR and coastal hazard studies (see, for example, ref. 11). However, spaceborne DEMs can have vertical errors up to several metres, contain sensing or interpolation artefacts, or are outdated, thereby affecting the quality of coastal hazard assessments, especially in flat, subsiding coastal plains and densely populated river deltas[10,12,13]. Apart from considering vertical uncertainty of elevation data, using DEMs in coastal hazard assessments requires correctly combining coastal elevation with local sea-level height and the proper conversion to a common vertical reference frame[2,3]. Through a systematic review evaluating recent SLR impacts and coastal hazard assessment studies, we found that these crucial steps were often not considered or performed incorrectly. Rather than considering actual, local sea-level height, coastal sea level is most often assumed to equal (an often outdated) global geoid (or in some instances even ellipsoid), to which open-access global DEMs are typically referenced when provided.

A geoid is an equipotential surface model that approximates MSL based on gravity and the rotation of Earth. As geoid quality depends on gravity observations, uncertainties in global geoid models can range up to several metres in regions that suffer from gravitational data paucity (see, for instance, refs. 14,15), predominantly located in the Global South. Moreover, actual sea-surface height is not just determined by the gravity and rotation of Earth, but also by, for example, ocean currents and large-scale circulation, winds, tides, seawater temperature

[1]Soil Geography and Landscape Group, Wageningen University and Research, Wageningen, The Netherlands. [2]Institute of Geography, University of Cologne, Cologne, Germany. [3]Department of Civil, Environmental and Architectural Engineering, University of Padova, Padova, Italy. [4]Department of Groundwater and Water Security, Deltares Research Institute, Utrecht, The Netherlands. ✉e-mail: Katharina.Seeger@wur.nl; Philip.Minderhoud@wur.nl

and salinity. As a result, time-average sea-surface height can deviate strongly (up to several metres) from a geoid, and its difference is the so-called mean dynamic topography (MDT).

The widespread omission to properly reference coastal elevation to measured local sea level generally leads to an underrepresentation of actual coastal sea-level height in (global) SLR and coastal hazard assessments and can introduce errors with magnitudes as large as a century of projected SLR for affected regions in the world[2,3]. By means of meta-analyses, we quantify the implications of the most-frequently encountered omissions or errors (that is, incorrect or absent vertical datum conversion) on estimates of exposed people and coastal area at global and regional scales. To evaluate the broader implications of our findings, we quantified the magnitude of potential underestimation or misjudgement in assessment studies included in the most recent IPCC AR6 reports.

To facilitate proper, future (re)assessments of coastal hazard impacts, we converted several state-of-the-art global DEMs to coastal sea-level height and provided them ready for use (see Data availability and Code availability sections). We conclude with concrete recommendations, such as data documentation guidelines and peer-review checklists, to ensure correct vertical datum alignment in future publications and improve community research standards. Herewith, we aim to eliminate future propagation of erroneous methodologies that caused this community-wide blind spot and resulted in widespread underestimations of coastal SLR and hazard impact assessments.

## Most coastal hazard studies lack rigour

We evaluated 385 peer-reviewed, scientific publications (systematically selected through a PRISMA-guided literature search, published between 2009 and 2025, with >53% in the past 5 years) on SLR and/or RSLR and coastal flood exposure, vulnerability and risk, presenting both global and regional hazard assessments. We scrutinized each for the correctness of DEM usage, vertical datum conversion and proper integration of sea-level height and coastal elevation and documented the ubiquity of errors and omissions (see the Methods for details; see also Extended Data Fig. 1, Supplementary Figs. 1–3, Supplementary Tables 1 and 2 and Supplementary Data 1). The evaluated coastal hazard assessment literature investigates coastal settings all over the world and across spatial scales. The impact assessments focused on SLR and/or RSLR (14%), storm surge (8%), tsunami (8%), coastal exposure, vulnerability and/or risk more in general (41%), or included combinations of different single-type coastal hazards, exposure, vulnerability and/or risk assessments and technical aspects and methodological advancements (29%). The evaluated literature includes studies from data-sparse coastal lowlands in Africa (14%) and Asia (58%), which are severely affected by RSLR and coastal flooding[16], poor data availability and accessibility as well as data inaccuracy due to the comparably poor performance of global Earth gravity models (see, for example, ref. 17). Moreover, the evaluated literature includes high-impact, global hazard assessments (10%), which dictate the contemporary scientific understanding of global coastal hazard impacts.

In the bulk of the evaluated publications (73%), documentation of used sea-level height, coastal elevation and vertical datums is either incomplete (13%) or entirely missing (60%) (Fig. 1). Although 27% of the literature correctly documented the vertical datum(s) used, only 1% correctly described and aligned coastal sea-level height information to the land elevation data. For the vast majority of evaluated assessments, sea level and coastal elevation alignment and datum conversion description were absent, and conversion was probably omitted (90.6%). In the remaining studies, the description was often incomplete (making the study irreproducible) or described an incorrectly performed conversion (8.6%). Correct alignment of elevation data from global DEMs to sea level requires a vertical datum conversion in combination with additional sea-level data, either a local sea-level datum (for example, local tide gauge and/or national sea-level-aligned datum) or global ocean surface topography (for example, satellite altimetry and/or buoys combined product such as MDT). We expect that authors aware of the necessity and with the expertise to successfully perform a vertical datum conversion using additional sea-level data do properly document these crucial (and often time-consuming) methodological steps and additional datasets. Therefore, we presume that the absence of any documentation of sea-level information (25%) and/or methodological conversion steps (65%) means that sea-level height data was not included or datum conversion to a sea-level reference was omitted. Repeated evaluation across multiple studies confirmed the validity of this presumption (see, for example, ref. 2) (Supplementary Tables 3–5).

The most prominent issue in the evaluated literature (demonstrably present in 25% and presumably present in 63% more; Figs. 1 and 2 and Supplementary Fig. 3) was the neglection of datum conversion from geoid (in some cases even ellipsoid) to a sea-level reference, thereby implicitly assuming a geoid height of 0 m to match local sea-level height. Of all literature containing this issue, we encountered only two papers that reflected on the potential discrepancy between geoid and actual sea level[8,18]. The second most frequent issue (9%) was incomplete, and thereby erroneous, datum conversion and inadequate alignment of vertical datums of datasets involved (for example, land elevation, bathymetry, sea level and heights of coastal infrastructure). This group of literature also contains few studies that pioneered the use of MDT data to create a sea-level reference[19–22], indicating first signs of community awareness on the necessity for correct land–sea-level alignment. Although these studies arguably improved on the bulk of geoid-based studies neglecting sea-level alignment, they suffer from conversion documentation shortcomings and demonstrated[20,21,23] incorrect datum conversion (Extended Data Fig. 3). Only one (ref. 13) out of the 385 evaluated studies (0.3%) had complete vertical datum and conversion documentation and contained no conversion and alignment errors.

Apart from evaluating individual studies, our literature evaluation also revealed widespread and persistent propagation of erroneous workflows, for example, omitting sea-level datum conversion[24,25] or introducing methodological datum conversion errors[19,22,26,27], to consequently affect follow-up studies that apply the same data and processing approach[20,21,23,28] (Supplementary Data 1). Similarly, assessments using modelling frameworks that include elevation data in the coastal zone also suffer from the investigated issues such as incomplete or absent vertical datum documentation (see ref. 29, for instance) and datum conversion (for example, ref. 30) and/or lack of a sea-level datum (ref. 31, for instance) (Supplementary Data 1).

## Coastal sea level often higher than geoid

We performed several meta-analyses on global and regional scales to quantify the magnitudes of coastal sea-level height misrepresentation stemming from the most-frequently encountered processing omission or errors for the most widely used geoids (Methods and Extended Data Fig. 2). Impact assessments that neglect datum conversion and consequently assume the geoid surface to represent MSL result in a global average underrepresentation of coastal sea-level height as represented by MDT of respectively 0.27 m (median 0.19 m, standard deviation (s.d.) 0.76 m) for the EGM96 and 0.24 m (median 0.16 m, s.d. 0.52 m) for the EGM2008 (Fig. 2, Extended Data Figs. 3 and 4 and Supplementary Figs. 4 and 5), which reflects the values reported in the abstract. In regions in which the respective geoids perform more poorly (for example, higher inaccuracy caused by data paucity), the discrepancies can go up to several metres (5.5–7.6 m for EGM96 and 2.8–3.4 m for EGM 2008) (Fig. 2).

Although the global statistics average out the larger regional and sub-regional discrepancies, these seem to be particularly large for several key regions, most located in the Global South. Largest discrepancies are

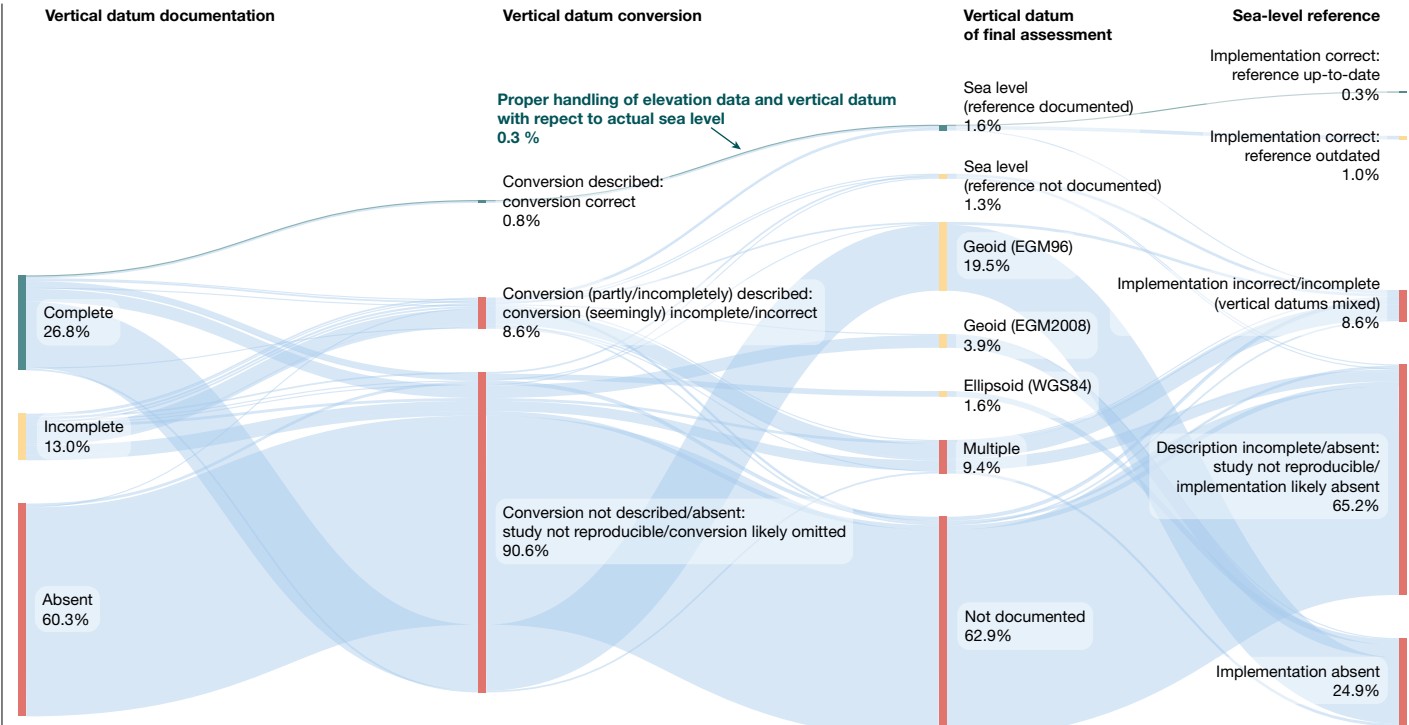

**Fig. 1 | Evaluation of sea-level and coastal elevation data documentation and their proper use and alignment in coastal hazard assessments (n = 385).** Complete vertical datum documentation means all necessary vertical datum information is provided in the study itself or in the cited references of the data used. Vertical datum conversion is correct if all necessary datum conversion steps required to properly align all data to a common vertical reference are properly described and applied. The correct implementation of a sea-level reference involves the use of sea-level information (for example, MDT or tide gauge data), correctly aligned with all other data used in the assessment in a common vertical reference. A sea-level reference is considered up to date if the latest available sea-level data were used. In 73% of the studies, vertical datum documentation was incomplete or completely absent. In nearly all evaluated assessments, sea-level data and their proper alignment to coastal elevation data were either not documented and datum conversion likely omitted (90.6%; see underlying presumption in main text), or (seemingly) incorrectly performed (8.6%). Only 0.3% of the evaluated studies completely documented, converted and properly adjusted coastal elevation data with sea-level information (shown with green colour). The Sankey diagram for the results of this study was created using SankeyMATIC (https://sankeymatic.com/).

observed in Southeast Asia (hosting large, populous and low-lying river deltas; Extended Data Figs. 6 and 7 and Supplementary Figs. 6 and 7) and the Pacific Region (often lowly-elevated atolls), on average amounting to an underrepresentation of coastal sea level of more than 1 m, as previously already highlighted in local studies on the Mekong and Ayeyarwady deltas[2,3]. Other areas with large discrepancies are located in Latin America, the west coast of North America, the Caribbean, Africa, the Middle East and the larger Indo-Pacific. Although, on average, the geoid models underrepresent sea-surface height at global and regional scales, locally the discrepancies can also range in the opposite direction (for example, northern Mediterranean coast, Antarctica and some islands in the Atlantic and the Pacific (EGM2008 only)) (Fig. 2), consequently resulting in an overrepresentation of sea-surface height. The lowest discrepancies between MSL and geoid are prevalent in Eastern North America, as well as Northern and Western Europe (Supplementary Data 2), reflecting the stronger performance of geoid models to approach sea-surface height in data-rich regions in the Global North. Continuous advances in global geoid modelling are reflected by new global geoids (for example, GOCO2025s), but these also contain comparably large discrepancies to MDT-determined MSL (Supplementary Fig. 9), in particular for regions in the Global South. Therefore, omitting to include sea-level information and to properly convert from a geoid to a sea-level datum, especially in data-sparse and remote regions in which geoid and sea level do not align well, immediately transfers these discrepancies (Fig. 3) as errors into hazard and SLR impact assessments[24,32–34] or into the delineation of the low-elevation coastal zone (LECZ)[25].

Studies that do include sea-level information but subsequently omit to convert elevation (DEMs) and sea-level data (for example,

altimetry-based MDT or mean sea surface (MSS)) to a common datum before combining them, introduce datum offset errors (that is, the respective height difference between the different datums, present in 9% of the studies; see, for instance, refs. 19–21,26,27,35). The few studies that did incorporate MDT data but suffer[20,21] from a specific geoid conversion error (Extended Data Fig. 3) statistically nearly equal our global average coastal sea-level height representation (underrepresentation of 0.02 m), but suffer regionally from discrepancies up to several metres, ranging from −4.1 m to +4.3 m (s.d. = 0.76 m) (Supplementary Data 2).

A third source of error stems from using an outdated sea-level reference. Tide gauges provide locally specific information, but their ability to adequately reflect recent sea level can be severely restricted when the time series is too short, incomplete or outdated. As tidal datums may have been established decades to centuries ago, for example, the North American Vertical Datum of 1988 (NAVD 88) in 1985 (ref. 36), or the Amsterdams Peil (AP) in the 17th century (later the Normaal AP in 1875)[37], their usage is valid as long as sea-level changes since establishment are correctly taken into account. The incorrect use of sea-level height information, for example, from tide gauges or satellite altimetry[21,23], without considering past relative sea-level changes since datum establishment and start of the projection period, contradicts the implied actuality of the hazard assessment conducted. Given the high rates of RSLR due to coastal subsidence[5,8], which exceed global SLR in many parts of the world, particularly in densely populated Asian deltas and coastal lowlands[6,8], assessments of contemporary coastal exposure must account for relative sea-level changes since tide gauge datum establishment or, for MDT/MSS, the average observation period[38]. Similarly, as coastal elevation itself is also not static,

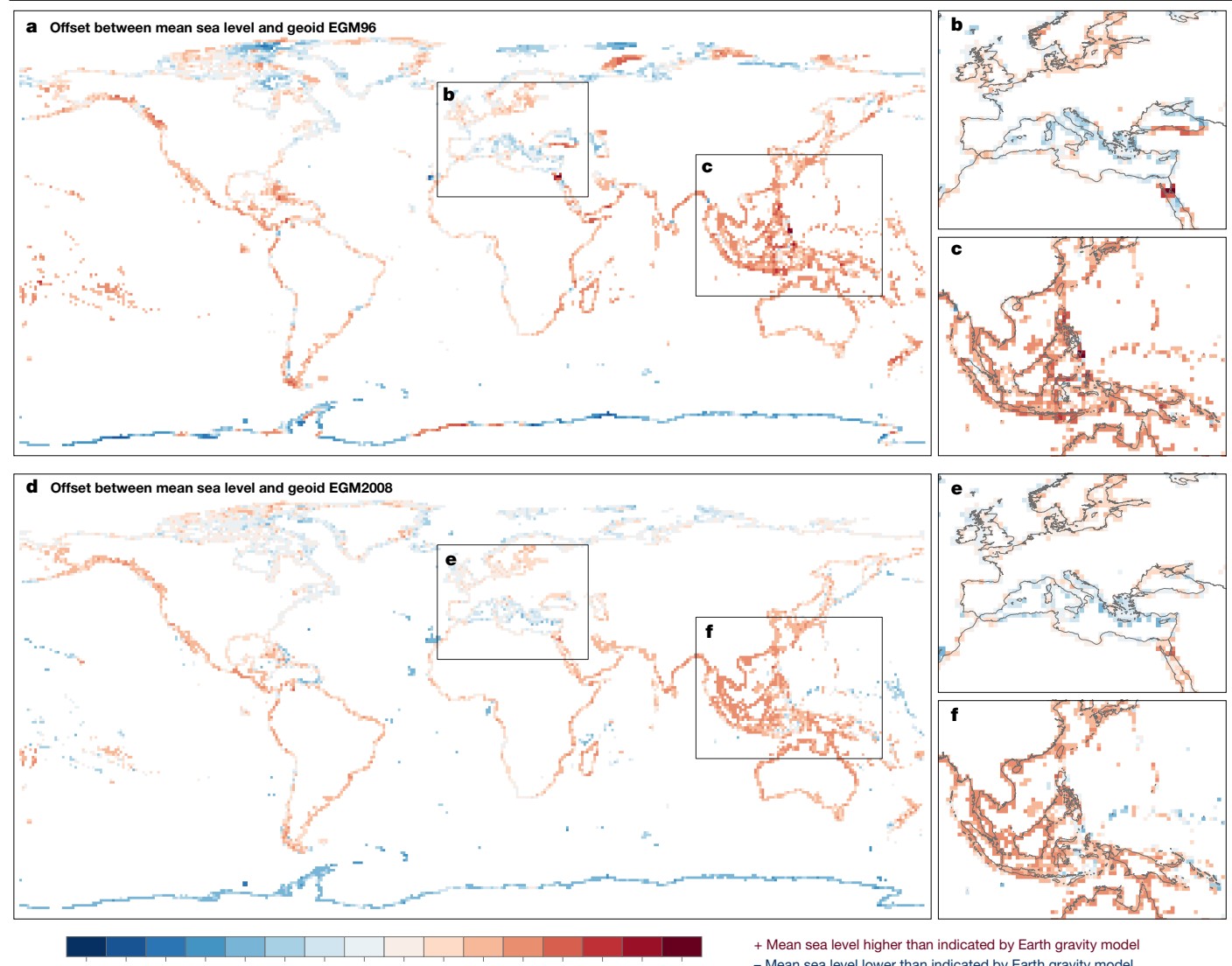

**Fig. 2 | Difference between coastal sea-level height and most-used geoids across the globe. a–c**, The still widely used Earth Gravitational Model 1996 (EGM96) shows large deviations from measured coastal sea level, here indicated by the latest available MDT[44], especially profound at the regional scale. **d–f**, The more recent EGM2008 geoid model shows overall improvement over EGM96 and provides globally a slightly better approximation of local coastal sea level. In data-rich countries in the Global North, the global geoids represent coastal sea level relatively well (for example, Eastern United States, Northern Europe and Western Europe), whereas in the more data-sparse Global South, regions such as Latin America, East Africa and the Indo-Pacific, with Southeast Asia and

Oceania as global hotspots, the geoids substantially underrepresent actual sea-surface height, ranging from several decimetres up to several metres locally. The vast majority of the evaluated literature assumed the geoid surface (0 m) to represent contemporary local MSL, thereby introducing the above discrepancy as error into their respective coastal hazard and SLR impact assessments. For visualization purposes, the spatial scale of the data shown was resampled to 1° using bilinear resampling, whereas all statistics are given at 90 m spatial resolution. The results were visualized using QGIS v.3.28.6 and shapefiles from ref. 51 (Open Government Licence v.3.0).

we highlight the urgency of using the most recent and accurate elevation data, ideally corrected for spatial heterogeneous vertical land motion[39] and consequent elevation change since data acquisition[38]. However, a thorough evaluation and inclusion of relative sea-level or elevation change since datum establishment or elevation acquisition practice is far from common practice, as only 22% of the evaluated studies use actual DEMs (that is, the latest available elevation data by the time of initial submission; Supplementary Table 2), whereas only 9% include DEM accuracy assessments (Extended Data Fig. 1).

## Coastal studies underestimate exposure

We performed several meta-analyses to evaluate the impacts of the most-frequently witnessed issues in existing coastal hazard assessments and quantified the potential exposure misjudgement present in global

relative SLR impact assessments. We used four of the best-performing global DEMs to date as supplied in their data repositories[40–43] and applied a RSLR scenario of 1 m, while omitting proper inclusion of a sea-level datum to mimic the most commonly applied methodology (>90% of all assessments), that is, assuming the global geoid surface to represent contemporary sea-level height (Fig. 1 and Supplementary Data 3). We compared the results with the same DEMs correctly aligned to measured local MSL by properly applying the most recent MDT data[44], showing the discrepancy between the commonly assumed sea-level height (that is, geoid) and measured local mean sea-level height.

Our meta-analyses show that worldwide estimates of land area and population below MSL after a 1 m relative SLR are substantially underestimated when MDT data are not included. Proper sea-level referencing using MDT increases the exposed area from 294,500–431,100 km² to 460,100–670,000 km² (that is, by 31–37%), and population from

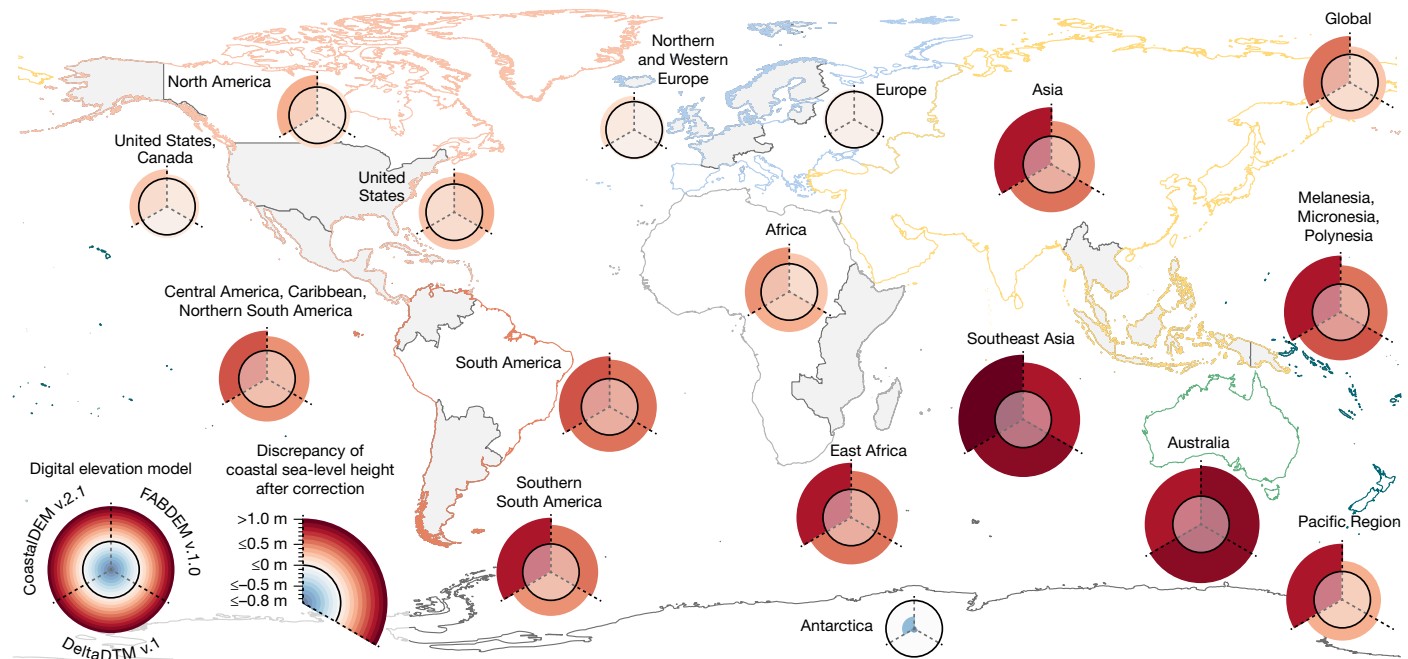

**Fig. 3 | Discrepancy between the commonly assumed coastal sea-level height in hazard assessments and the measured local sea-level height.** Meta-analysis of the most common shortcoming in existing coastal hazard assessments (that is, assuming the geoid to represent local sea-level height), using modern, globally available DEMs (CoastalDEM v.2.1 (ref. 40); FABDEM v.1.0 (ref. 41) and DeltaDTM v.1 (ref. 43)). The difference given is the coastal sea-level height discrepancy for each DEM between assumed coastal sea level (that is, assuming 0 m elevation of the respective geoid-referenced DEM to equal contemporary MSL) and our assessment in which we correctly align coastal elevation to measured mean sea level using the latest MDT data[44]. On average, global coastal sea level is about 0.3 m higher than commonly assumed in coastal hazard assessments, whereas in Southeast Asia, the discrepancies are the largest, with measured sea level exceeding previous assumed levels by, on average, 0.9–1.1 m. All statistics are computed at 90 m spatial resolution. Colour outlines and grey-shaded areas indicate the regions and subregions (Supplementary Fig. 8). The results were visualized using QGIS v.3.28.6 and shapefiles from ref. 51 (Open Government Licence v.3.0).

34.0–49.2 million to 77.0–132.2 million people (that is, by 48–68%) across different geoid-based DEMs and various population datasets (Fig. 4 and Extended Data Fig. 5). This adds, respectively, 287,400–470,700 km$^2$ and 55.1–101.6 million people to the 172,700–234,000 km$^2$ area and 21.9–34.5 million people already below MSL so far (with respect to 114,200–158,600 km$^2$ and 10.5–15.4 million people estimated so far below MSL without MDT referencing). In the most affected region, Southeast Asia, the estimated area and population below MSL following 1 m RSLR increased up to 94% and 96% following proper MDT referencing, increasing the exposure numbers to 78,000–99,700 km$^2$ and 24.2–46.9 million people for this region only. Global assessments of the LECZ (first 10 m of coastal elevation) based on geoid elevation underestimate area up to 4% and people up to 8% globally and proper MSL referencing increases the global LECZ to 3.0–4.1 million km$^2$ being inhabited by 0.82–1.07 billion people (Supplementary Data 3).

When we corrected the encountered geoid conversion error present in existing studies applying MDT data (that is, omitting to correct the offset between the EGM96 and DIR-R4 geoids) (Extended Data Fig. 9), our meta-analysis showed that global coastal area below MSL after a 1 m RSLR decreased by 2% (11,400 km$^2$), from 527,000 km$^2$ to 515,700 km$^2$ (of which, respectively, 133,100 km$^2$ and 117,100 km$^2$ are currently already below MSL). However, the estimated global population below MSL following a 1 m RSLR considerably increased by 10–12% (8.0–14.1 million) from 68.6–108.4 million to 76.6–122.4 million people (of which, respectively, 18.2–27.9 million and 16.5–25.2 million are currently already below MSL). This shows that this specific geoid conversion error (Extended Data Fig. 9) results in underrepresenting MSL disproportionally in populated coastal areas. Consequently, assessments that do use MDT as sea-level reference but suffer from this error (demonstrated for ref. 23 and presumably more, for example, refs. 19,45) underestimate population exposure in affected regions

(Fig. 4, Extended Data Figs. 5–7, Supplementary Figs. 6 and 7 and Supplementary Data 3).

At the local scale, the encountered vertical reference issues particularly affect hazard assessments in large, often populous, low-elevated coastal-deltaic areas (Extended Data Figs. 6 and 7; Supplementary Figs. 6 and 7) and add on other DEM-related vertical uncertainties[16,38]. To evaluate the relative impact of vertical reference issues with respect to other (DEM-dependent) uncertainties, we performed a detailed meta-analysis for the Vietnamese Mekong Delta (VMD). The VMD exemplifies many densely populated coastal landscapes exposed to RSLR worldwide, being one of the largest, flattest and low-lying deltas worldwide[2]. The results indicate that the common assumption that the DEMs' geoids represent local sea level created the largest impact assessment errors. Not including a proper sea-level reference (in our case using MDT data), added 10–60% additional (DEM-specific) error to the relative elevation assessment on top of existing data-inherent inaccuracy, sea-level and elevation change effects[38] (Supplementary Table 3). Proper inclusion of MDT data increased the area and population in the VMD exposed to a 1 m RSLR from 1,400–6,000 km$^2$ to 18,400–24,800 km$^2$ (that is, by 72–95%), and from 312,900–2.4 million to 5.4–10.0 million people (that is, by 74–96%), confirming the findings from earlier research on the elevation of Mekong Delta[2] (Supplementary Table 4). The specific geoid conversion error encountered for incorrect MDT implementation (Extended Data Fig. 9) is particularly prominent in coastal lowlands of Southeast Asia and when corrected for the VMD, exposed area and population following a 1 m RSLR increases by, respectively, 18% (from 20,900 km$^2$ to 25,500 km$^2$) and 19–23% (from 5.5–7.9 million people to 6.8–10.2 million people). These numbers highlight the large spatial variability and locality-specific impacts of geoid conversion errors that are not evident in regional and global statistics.

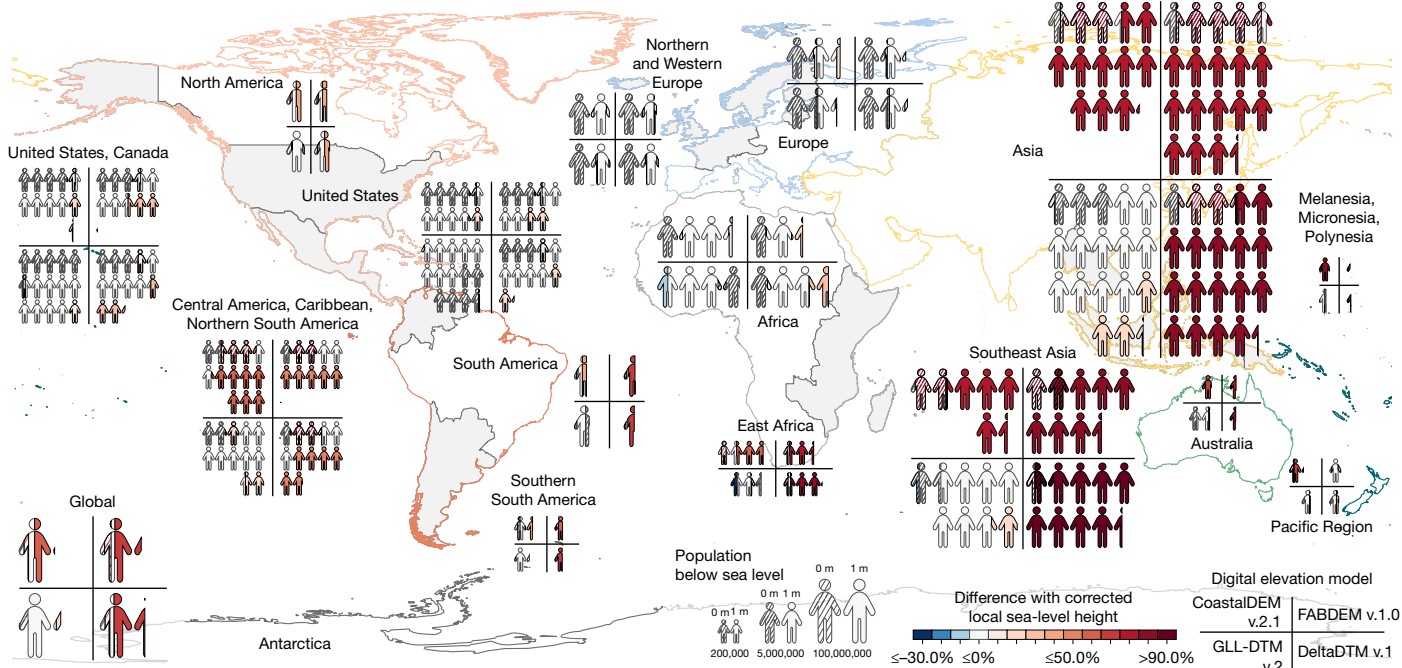

**Fig. 4 | Difference in population falling below sea level following 1 m RSLR between the commonly assumed and measured sea-level height.** Meta-analysis showing the impact of the most-frequently observed errors in existing coastal hazard assessments, that is, omitting or improper alignment of measured coastal sea level to land elevation, using modern, globally available DEMs (CoastalDEM v.2.1 (ref. 40), FABDEM v.1.0 (ref. 41), GLL-DTM v.2 (ref. 42) and DeltaDTM v.1 (ref. 43)). Properly referencing coastal elevation from geoid to local MSL increases global population exposure estimates from 44.0–108.4 million to 102.8–132.2 million people below sea level following 1 m RSLR (that is, by 12–67%). Population estimates are based on WorldPop 2020 (ref. 52) and do not account for future population change. The largest population exposure increase, that is, by 13–96%, is observed in Southeast Asia, where 32.1–46.9 million people (of which 5.0–9.3 million currently already reside below sea level) fall below sea level following 1 m RSLR. All statistics are computed for the respective spatial resolution of the DEMs. Colour outlines and grey-shaded areas indicate the regions and subregions used (Supplementary Fig. 8). All data related to this figure are included in Supplementary Data 3. The results were visualized using QGIS v.3.28.6 and shapefiles from ref. 51 (Open Government Licence v.3.0) and icons modified from Font Awesome Free v.7.1.0 (CC-BY 4.0; https://creativecommons.org/licenses/by/4.0/).

To investigate the potential magnitude and consequences of the revealed issues beyond the individual hazard assessments, we assessed the inclusion of the evaluated studies in the latest IPCC AR6 cycle. A total of 46 evaluated studies (categorized as follows: correct implementation of sea-level reference ($n = 1$); incorrect implementation of sea-level reference ($n = 9$); absence of sea-level reference ($n = 36$)) are included in the screened IPCC AR6 reports, most frequently in chapters from Working Group II and particularly ($n = 25$) in the Cross-Chapter Paper (CCP2) (Supplementary Figs. 10 and 11, Supplementary Table 5 and Supplementary Data 4). Our quantitative comparison of selected assessments indicates that datum conversion errors[23] (potentially for ref. 45; Extended Data Fig. 9) and sea-level reference omissions (see ref. 46, for example) may have led to underestimated coastal exposure in the IPCC reports. For instance, CCP2-reported estimates of people residing in the LECZ (896 million; about 11% of the global population in 2020)[45] are below our estimates following proper MDT referencing (966 million to 1.07 billion; 12.3–13.7% of the 2020 global population) (Supplementary Table 5).

## Discussion and recommendations

Our study reveals fundamental misalignment issues of sea level and coastal elevation throughout a wide body of scientific literature, which introduces errors and creates large uncertainties in the vast majority of coastal hazard and SLR and/or RSLR impact assessments. From all evaluated studies, more than 99% did not use sea-level information, omitted or made errors during sea-level datum conversion and missed crucial datum and processing documentation, rendering the studies irreproducible. In most cases, the encountered methodological issues lead to an underrepresentation of coastal sea-level height, causing existing assessments to underestimate the spatial extent and timing of future RSLR and coastal hazard impacts. This raises concerns about the correctness and reliability of existing assessments and calls for re-evaluation of the workflows and results. At present, we risk that global efforts to improve sea-level measurements and projections to mm accuracy (see, for example, refs. 4,47) are nullified by erroneous sea-level and elevation data implementation in coastal hazard and SLR impact assessments. Our findings reveal a community-wide blind spot, which calls for a systemic change in how we deal with sea-level and (coastal) land elevation data in the global scientific community and beyond.

A potential explanation for the encountered vertical datum issues may be the current constellation that leaves complex geodetic transformations, required to correctly use elevation data for coastal hazard assessments, in the hands of non-specialist end users unfamiliar to the required processing steps. One solution to avoid future errors from omitted or wrongly performed datum conversion and sea-level referencing may lie in the hands of the data providers, which could provide readily combined products of digital terrain with sea level to facilitate proper end use, as we do so in this paper (see Data availability section). This would especially make sense for DEMs that are specifically developed to target the coastal zone and facilitate coastal hazard assessments, such as CoastalDEM[40,48] and DeltaDTM[43]. Although we addressed issues on coastal elevation and sea level, we found indications for similar issues in studies using bathymetry data, suggesting further future research and critical reflection in those domains as well.

The fact that geoid models are developed and performing relatively well in reflecting local sea-level height in the Global North (for example, the United States and Western Europe) may perhaps explain the overconfidence placed in geoid-model performance by Global-North-based

scientists when performing assessments on a global scale or in other regions of the world in which geoid models perform less well. Although we did not further investigate or test this hypothesis, we raise this to call for further scrutiny on the Global North–South transferability of scientific approaches and datasets in future research. A noteworthy example of incompatible North–South transferability occurred with CoastalDEM v.1.1 (ref. 48), of which the neural-network approach used to create the CoastalDEM was Global-North-trained (the United States) and Global-North-validated (Australia), but performed considerably worse elsewhere. The approach, for example, placed half of the entire Mekong Delta already well below present-day sea level, thereby greatly overestimating consequent population exposure to high-water levels in the region (Extended Data Fig. 10).

Our results indicate that scientific peer-review has so far been unsuccessful in withholding the investigated errors from publication and propagation through the literature. Journals could introduce dedicated steps in their submission and peer-review procedures, for example, by providing elevation and datum documentation guidelines, requesting author declarations and adopting review checklists to aid referees. Apart from helping to avoid errors by ensuring proper data use and datum conversion and making studies transparent and replicable, these actions will also raise awareness on proper vertical referencing with the wider research community. This key factor will become even more relevant with the emergence of new elevation datasets (for example, high-accuracy measurements of relative sea level and coastal elevation using ICESat and SWOT data), which will also require proper vertical referencing and documentation when applied for further assessment. Moreover, the correct use of sea-level and land elevation data is also critical to the emerging integration of vertical land motion (VLM) into RSLR projections[5,39,49]. We suggest adopting a proper 'dynamic elevation' approach, combining multiple elevation datasets[50], performing multi-source uncertainty and accuracy evaluations[38] and correctly interpreting and integrating VLM observations[49], to create reliable, state-of-the-art projections of future RSLR.

This study probably reveals only the tip of the iceberg, as the evaluated publications form only a representative selection of the full body of coastal hazard assessments. It is concerning that many of the evaluated studies (see, for example, refs. 22,24) are used to underpin SLR impact and coastal hazard exposure statements in IPCC reports[4,47], which, in turn, inform global disaster risk reduction (United Nations Office for Disaster Risk Reduction) efforts, governments and policymakers worldwide on coastal vulnerability, adaptation needs and timelines, as well as provide quantitative input for climate risk rankings and loss and damage discussions. We recommend that future IPCC reports include a specific review step to verify the methodological validity of referenced coastal hazard assessments. Apart from scientific publications, our investigations suggest that much grey literature, such as policy-forming documents, governmental reports and other consultancy-based assessments, especially in the more data-sparse Global South, focusing on coastal exposure and risk, contain similar issues. We did not investigate whether the issues identified in the above-mentioned reports and assessments have led to misinformed decision-making, but this cannot be ruled out. Our findings may have far-reaching implications for existing coastal adaptation, protection and mitigation strategies, especially those using satellite-derived elevation data as information base. This necessitates re-evaluating existing coastal hazard assessments to rule out vertical reference and sea-level datum issues and, if those assessments informed decision-making, potentially updating and expediting implementation timelines of coastal adaptation strategies, as exposure thresholds may be reached much sooner than previously projected.

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

## Methods

### Literature research and evaluation

We conducted a systematic and reproducible literature research and evaluation, adhering to standards of the Preferred Reporting Items for Systematic Reviews and Meta-Analyses (PRISMA) process. All details of the literature research, selection and evaluation are provided in Supplementary Tables 1 and 2. We used the Scopus search engine and used a search term for titles, abstracts and keywords that covered keywords from low-lying coastal environments, processes and hazards, as well as elevation information and elevation dynamics, impact and/or hazard assessments, projections or evaluations (Supplementary Table 1). The Scopus search engine was used as it constitutes one of the most transparent search engines. We are aware of the limitation that Scopus does not include the entire available (scientific) literature and therefore might introduce a bias into our investigation. Therefore, we performed a second independent literature evaluation on 96 additional studies (90 after excluding publications by the authors; non-systematically collected through other search engines) (Extended Data Fig. 8) to ensure the results from the search were bias-free. All results presented in the main paper are based on the Scopus-based systematically researched literature only. From the initial 35,001 documents that were identified through Scopus, we included only peer-reviewed scientific articles, reviews including original analyses, or data papers. Moreover, we limited our search to the most recent 15 years (2009–2025) to include only the most recent research from the period during which both global DEMs and global MDT data were available and well-established, thereby ensuring a levelled playing field of global data availability across the globe to evaluate datum conversion and sea-level reference methodologies for assessments. Further filters applied refer to the exclusion of certain unrelated subject areas, keywords and limiting the search to the English language. This refined search term reduced the amount of literature to 7,241 documents that were subsequently screened in two consecutive rounds following established inclusion and exclusion criteria (Supplementary Fig. 1 and Supplementary Table 1). The first screening was conducted based on title, abstract and keywords, and the second screening was conducted based on in-depth analysis of the full article and, when needed, its references and supplementary material. The screening was consistently performed by K.S., whereas a subset of 1,000 articles (first screening) was cross-checked and validated by P.S.J.M., hereby not encountering conflicting decisions. Publications that were categorized as 'yes' and 'maybe' in the first screening were included in the second, full-article screening. The inclusion criteria were focus on coastal land and coastal sea level, focus on coastal hazard and impact assessment, focus on coastal lowlands and the use of satellite-borne elevation information. The main exclusion criteria were the absence of elevation data use, use of airborne lidar without combination with satellite-borne elevation data, methodological and technical articles without application, articles referring to elevation data cited in tertiary source or beyond (only primary or secondary sourced studies included) and focus on secondary hazards (cascading effects). The finally included literature, consisting of 385 publications, underwent an in-depth evaluation of their respective research data and methodology, scrutinizing them, and—where needed—also their secondary references, to assess all relevant details related to study area, distance to shore, study focus and/or flood type, number of DEMs, DEMs used, actuality of DEM, DEM type, DEM coverage, acquisition and technique, performance of DEM accuracy assessment, DEM accuracy, spatial resolution, DEM availability, horizontal datum used, vertical datum documentation, vertical datum conversion, vertical datum type, vertical datum used, implementation of sea-level reference, and—where needed—additional comments (Supplementary Table 2).

The individual papers were accessed by either the Scopus search engine, ISI Web of Science, Science Direct, Google Scholar or ResearchGate. The complete evaluation of the selected 385 publications and the subsequent analysis of the results following the developed evaluation criteria and coding are provided in Supplementary Table 2. Only publications that handle all datasets in a consistent vertical reference frame (that is, performing a correct vertical datum conversion) and refer to the latest available sea-level information at their time (that is, using an up-to-date sea-level reference and correct vertical datum conversion) were considered as proper. As vertical datum conversion always requires additional data next to a DEM-based coastal elevation dataset (for example, vertical datum information, offset between geoid/ geoids and tidal level, sea-level height data) as well as a conversion procedure (including a specific GIS or coding environment or conversion service), we expect researchers who perform a datum conversion to document the additional datasets and procedures in their papers and/or supplementary information or supplementary data. In case a paper does not provide any documentation of additional datasets or required conversion steps, we presume that no datum conversion was performed. This assumption was confirmed to be correct for numerous papers for which the absence of datum conversion becomes apparent in the results of the study.

Finally, we evaluated more than 90 additional publications that fitted the selection criteria, but were not included in the Scopus-searched literature. This subset was separately collected by the authors through additional, non-systematic literature searches, for example, to include relevant IPCC-referenced assessments, and served as an independent dataset to evaluate potential bias present in our systematic Scopus-searched literature dataset (Supplementary Data 1). The additional literature dataset underwent the same scrutinizing evaluation (Supplementary Table 1). We found nearly equivalent findings on the occurrence and percentage distribution for the various vertical datum and sea-level reference issues (Extended Data Fig. 8) as for the systematic review (Fig. 1), which suggests that our Scopus-based systematic review is unbiased and provides representative results for the existing body of literature.

### Vertical datum conversion and processing of DEMs

To obtain land elevation above continuous local sea level globally and quantify the discrepancies to land elevation with respect to global geoids or datum conversion errors, we processed four of the most recent DEMs at a global scale by converting them from their original vertical reference system to MSL as indicated by the MDT product. The DEMs included were CoastalDEM v.2.1 (original vertical reference system: EGM96, spatial resolution: 90 m × 90 m)[40], FABDEM v.1.0 (original vertical reference system: EGM2008, spatial resolution: 30 m × 30 m)[41], GLL-DTM v.2 (original vertical reference system: MDT[53], spatial resolution: 1° × 1°, that is, about 1,000 m × 1,000 m)[42] and DeltaDTM v.1 (original vertical reference system: EGM2008, spatial resolution: 30 m × 30 m)[43]. The publicly available GLL-DTM v.2 (referenced to CNES-CLS13 MDT[53]) contains a geoid conversion error (that is, the EGM-DIR R4 geoid was assumed to equal the EGM96) and does not apply the latest available MDT dataset, at the time. Therefore, we obtained the pre-converted GLL-DTM v.2 (referenced to EGM96) from the authors and conducted the vertical datum conversion to MSL[44].

We used the latest available global MDT HYBRID-CNES-CLS2022 dataset[44], which provides sea surface height above geoid (GOCO06s) across the globe, measured by satellite altimetry and combined with gravitational field information, oceanographic data from drifting buoys, high-frequency radar velocities and hydrological profiles. The MDT dataset[44] provides spatially continuous information of sea surface height above the GOCO06s geoid at a resolution of 0.125° averaged over a period from 1993 to 2021. Therefore, it provides the latest available information on global MSL and an accurate substitute for local tide gauge information in those regions. Several studies confirm the accuracy of MDT data in the range of cm (see, for example, refs. 54,55), although vertical accuracy decreases up to about 4 cm within 10 km of the coast[56]. It serves as an open-accessible product that—if properly

aligned with elevation information—can be used to adjust elevation with respect to local sea level continuously along the coastlines of the world.

The following section documents a proper, consistent and reproducible vertical datum conversion of the four DEMs, providing all required datasets, datum information, processing steps and used software environments and may serve as an example for future studies or existing studies aiming to re-evaluate previous assessments. For all our computations, we used the ArcGIS Pro environment. To reference coastal elevation of the DEMs to MSL as given by MDT, the offsets of the underlying respective vertical reference systems (that is, EGM96, EGM2008 and GOCO06s) were determined (Extended Data Fig. 2).

Geoid information was obtained from the openly accessible calculation service of the International Centre for Global Earth Models from GFZ Helmholtz Centre for Geosciences[57]. Point data on geoid height anomaly to the WGS84 ellipsoid was obtained for the entire globe at a resolution of 0.085° for the EGM96, EGM2008 and GOCO06s geoids, respectively. For each geoid, the height anomaly points were interpolated into a global raster by using multiquadric radial basis functions, which gave the most accurate interpolation results and is also used by gravitational field modelling studies[58,59]. Subsequently, the global geoid raster files were resampled to a common spatial resolution of 90 m × 90 m (for CoastalDEM v.2.1, FABDEM v.1.0 and DeltaDTM v.1) and 1,000 m × 1,000 m (for GLL-DTM v2) by using bilinear resampling to be comparable with the spatial resolution of the DEMs throughout the entire datum conversion process. Geoid offsets were determined by subtracting the GOCO06s geoid height anomaly raster from the EGM96 and EGM2008 geoid height anomaly rasters.

Using MSL as indicated by MDT as a vertical datum for land elevation data requires the extrapolation of MDT data over land. We extracted point values from the MDT HYBRID-CNES-CLS2022 raster using bilinear interpolation at point locations and subsequently extrapolated them over land using an inverse distance weighting algorithm and a smooth neighbourhood type with a smoothing factor of 0.5. The resulting raster dataset was resampled to two spatial resolutions of 90 m × 90 m (for CoastalDEM v.2.1, FABDEM v.1.0 and DeltaDTM v.1) and 1,000 m × 1,000 m (for GLL-DTM v.2) by using bilinear resampling to enable comparability with the spatial resolution of the DEMs and to avoid the introduction of potential artefacts stemming from large differences in spatial resolution in the datum conversion process.

The processing of DEMs involved two main processes (Extended Data Fig. 2d). First, the DEMs were converted from the EGM96 and EGM2008 geoid to the GOCO06s geoid by adding the computed respective geoid offsets. Therewith, the vertical references of DEM-derived land elevation and MDT-derived sea surface height are aligned to a common datum (that is, GOCO06s), which is a prerequisite to obtain land elevation above MSL by subtracting the MDT data from the respective DEM (Extended Data Fig. 2d). To automate the processing of the vertical datum conversion of the DEMs, we applied two ArcGIS Pro model workflows to convert DEM tiles from the EGM geoids to GOCO06s and subsequently to MDT, while preserving their original spatial resolution and properties (the Python codes are provided online).

As the performance and the reliability of MDT extrapolation over land decreases with increasing distance from the sea, we applied a distance threshold of 500 km from the coastline (as defined by Open Street Map[60]), for which we consider this approach of DEM vertical datum conversion valid. We converted all land elevation information within the distance threshold to MSL, thereby ensuring adequate extrapolation performance and full coverage of vast low-lying coastal plains and river deltas such as the Ganges–Brahmaputra–Meghna Delta.

### Assessment of coastal land elevation and sea-level height, SLR impact and LECZ from global DEMs

We investigated coastal land elevation and sea-level height globally by extracting point elevation data from the original and vertically converted DEMs and their respective differences, at a 90 m interval along the coastline using bilinear interpolation of values at point locations and excluding no data values and water bodies before rasterization. As some of the evaluated DEMs contain large negative, unrealistic elevation values that are likely artefacts from the source DEM acquisition and post-process steps, we excluded these by applying a minimum elevation threshold of 7 m below MSL, which represents some of the lowest elevations in the coastal lowlands worldwide, such as the Netherlands (see, for example, ref. 21). The elevation statistics (including minimum, maximum, mean, median and standard deviation) were calculated for global (both including and excluding Antarctica), continental and regional scales (using administrative boundaries provided in ref. 51). The global statistics provided in the main text and figures exclude the results for Antarctica, as there are no people living there.

To show the impacts of neglected conversion to a sea-level datum, we investigated the impact of 1 m RSLR on area and population when simulated for DEMs with their original vertical reference system and after conversion to MSL, apart from computing area and population at present already below sea level. Similarly, we investigated the discrepancies in area and population within the 10 m LECZ when different DEMs with and without proper vertical datum alignment are used. After no data values and large negative values (that is, ≤7 m below MSL) were excluded and water bodies masked, we reclassified the DEMs applying thresholds of ≤1 m and ≤10 m, respectively. Area in $km^2$ was calculated for global, continental and regional scales (Supplementary Fig. 8) using zonal statistics. We limit our assessment to relative elevation only, and do not apply a hydrodynamic inundation (for example, bathtub) approach, nor report on flood or inundation extent or impacts in our results.

To estimate population currently below MSL with 1 m RSLR and within the LECZ, we use three global population datasets and thereby avoid potential bias in absolute population counts arising from single datasets[25,61,62]. Uncertainties in and between population data stem from resolution (grid cell size) and quality of input data (for example, census data) and ancillary products as well as models to calculate statistics (ref. 63 and references therein), daily population dynamics and inconsistencies in administrative boundary data[61,62]. Therefore, we apply a multi-dataset approach and use unconstrained WorldPop data from 2020 at 100 m spatial resolution[52] as well as the LandScan Global dataset (800 m spatial resolution) for the years 2020 (to be comparable to the WorldPop 2020 data)[64] and the latest available 2023 data[65]. For the WorldPop dataset, we used data for individual countries as the global dataset provides only aggregated data that would lead to substantial overestimation in population. Population estimates were derived for global, continental and regional scales using zonal statistics. Global population was calculated by computing binary rasters for each DEM (1 if elevation is between 7 m below MSL and ≤0 m, ≤1 m and ≤10 m above MSL, respectively) and aggregating these at the respective (coarser) resolution of the population datasets, using the arithmetic mean. This created the fraction of which each raster cell meets the elevation requirement. Subsequently, the raster was multiplied by the respective population data and spatially summed to estimate population for each extent of interest, implicitly assuming equal distribution of population within a single population data cell. We note that this constitutes an uncertainty factor as people may be distributed disproportionately within a single raster cell, but data resolution restricts further detailing. Another shortcoming is that we do not account for population change in our exposure projections and use static population numbers. Adding a projection of population change in our 1 m RSLR scenarios is not possible, as the projection is spatio-temporally variable. Therefore, the actual number of future exposed population is probably higher, given the predominant projected growth of the human population, particularly in coastal zones of the Global South (see, for instance, refs. 11,66). All reported coastal elevation values, RSLR impact and LECZ statistics, as well as the respective deviations between estimates for DEMs with and without proper datum conversion, were quantified in absolute value and as discrepancies in percentage (Supplementary Data 2 and 3).

## Evaluating the use of investigated studies in IPCC AR6 reports

To investigate links of the evaluated literature to the latest IPCC reporting cycle (AR6), we screened the AR6 working group (WG)I–III and the Special Report on the Ocean and Cryosphere in a Changing Climate (SROCC) reports against our evaluated literature reference list. Through a reproducible, large-language-model-supported, screening protocol (protocol details are available in the Supplementary Information), we used ChatGPT-5 to screen all the chapter-specific literature lists from the AR6 WGI, II, III reports and WGII cross-chapter papers (using .bib files available on https://www.ipcc.ch/report/ar6/wg1/downloads/, https://www.ipcc.ch/report/ar6/wg2/downloads/, https://www.ipcc.ch/report/ar6/wg3/downloads/) and the SROCC report chapters (using PDFs of individual chapters, available on https://www.ipcc.ch/srocc/download/) for using references from our systematic review (385) and additional literature (96). We found that 46 studies in our systematic review and 29 in our additional literature were included as references in the IPCC reporting (Supplementary Table 5, Supplementary Figs. 10 and 11 and Supplementary Data 1 and 4). We then grouped these studies according to their evaluated categories: proper integration of sea-level reference ($n = 1$ and 2, for systematic review and additional literature, respectively); incorrect integration of sea-level reference ($n = 9$ and 7, for systematic review and additional literature, respectively); and absence of sea-level reference ($n = 36$ and 20, for systematic review and additional literature, respectively) (Supplementary Table 5) and performed a quantitative comparison (area and/or population exposure) with several representative and methodologically comparable studies from each category with the results from our meta-analyses, to quantify the magnitude of error in the existing, IPCC-referenced impact assessments as a result of various vertical referencing issues.

## Data availability

The original DEMs, MDT and population data used in this study are available in their respective online repositories[40–44,52]. The processed, global DEMs, converted to local MSL using MDT data and used for the meta-analyses in this study, are available for reuse and are accessible at *Zenodo* (https://doi.org/10.5281/zenodo.17722669). All computations were performed using ArcGIS Pro (https://www.esri.com/en-us/arcgis/products/arcgis-pro/overview) and layouting was done using QGIS (https://qgis.org/). The results of the in-depth literature evaluation, workflow protocol and all statistics of the coastal impact meta-analyses are available in the Supplementary Data and Supplementary Information.

## Code availability

The codes used for this article are available here at *Zenodo* (https://doi.org/10.5281/zenodo.17953234).

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

**Acknowledgements** We thank A. Collins, R. Kopp and Z. Siyum for providing extensive feedback and suggestions that helped to shape our assessments further and considerably strengthened the manuscript. We thank A. Bolten, M. van der Meij and L. Steinbuch for supporting the workstation use and server capacities needed to conduct the global processing, and for support during the data publication, and R. Vernimmen for providing the GLL DTM v.2 referenced to EGM96. For this paper, we went deeper than ever before. We thank J. Pernack and D. Brill for sparking and encouraging an everlasting scientific curiosity about Earth. We acknowledge the inspiration and remember H.-J. Schuurman and F. Verspaget. Here's to the crazy ones and an amazing better half. We thank the brothers Arthur and Luuk Minderhoud for providing P.S.J.M. ample opportunity for nighttime contemplations, which played a key part in shaping this manuscript. We also thank our supportive families and a wonderful grandma. Finally, we acknowledge the inquisitive 'paper tiger' that grew into a graceful lion. May it bask in the morning sun and sleep soundly tonight. P.S.J.M. acknowledges the funding from the Dutch Science Foundation (NWO) under the NWO Veni TTW 2022 (Applied and Technical Sciences) call with the project: 'Drowning Deltas—why deltas sink and what to do about it' (no. 20231). This work received funding to support open-access publication by the Dutch Ministry of Agriculture, Fisheries, Food Security and Nature under the Wageningen University & Research Knowledge Base Programme (KB).

**Author contributions** P.S.J.M. and K.S. conceptualized the study. K.S. and P.S.J.M. devised the methodology. K.S. investigated the literature review and data processing. K.S. and P.S.J.M. investigated the result analysis and interpretation. P.S.J.M. and K.S. investigated the IPCC reference screening. K.S. and P.S.J.M. helped with visualization. P.S.J.M. assisted with funding acquisition. K.S. and P.S.J.M. handled the project administration. P.S.J.M. supervised the study. K.S. and P.S.J.M. wrote the original draft and reviewed and edited the final paper.

**Competing interests** The authors declare no competing interests.

**Additional information**
**Correspondence and requests for materials** should be addressed to Katharina Seeger or Philip S. J. Minderhoud.

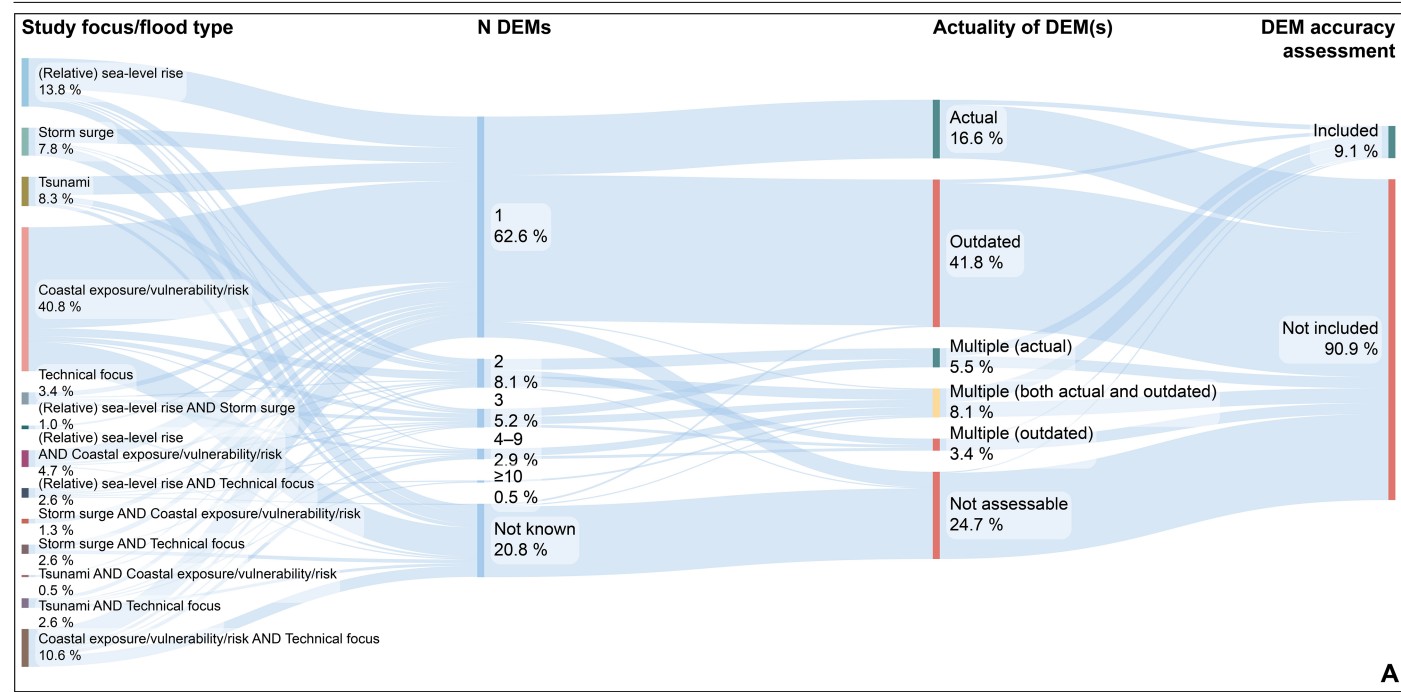

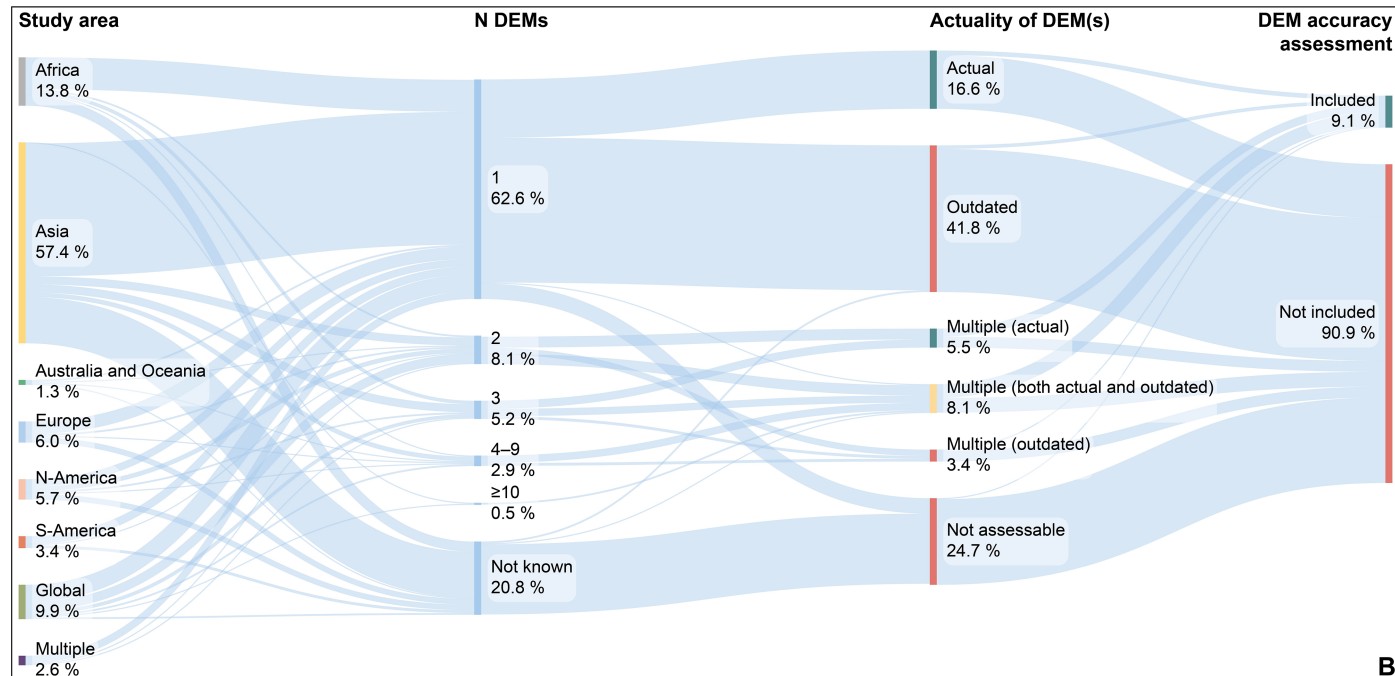

**Extended Data Fig. 1 | Evaluation of the usage of digital elevation models (DEMs) in the literature investigated in this study.** Use of DEMs based on (**A**) study focus/flood type and (**B**) study area. The Sankey diagrams for the results of this study were created using SankeyMATIC (https://sankeymatic.com/).

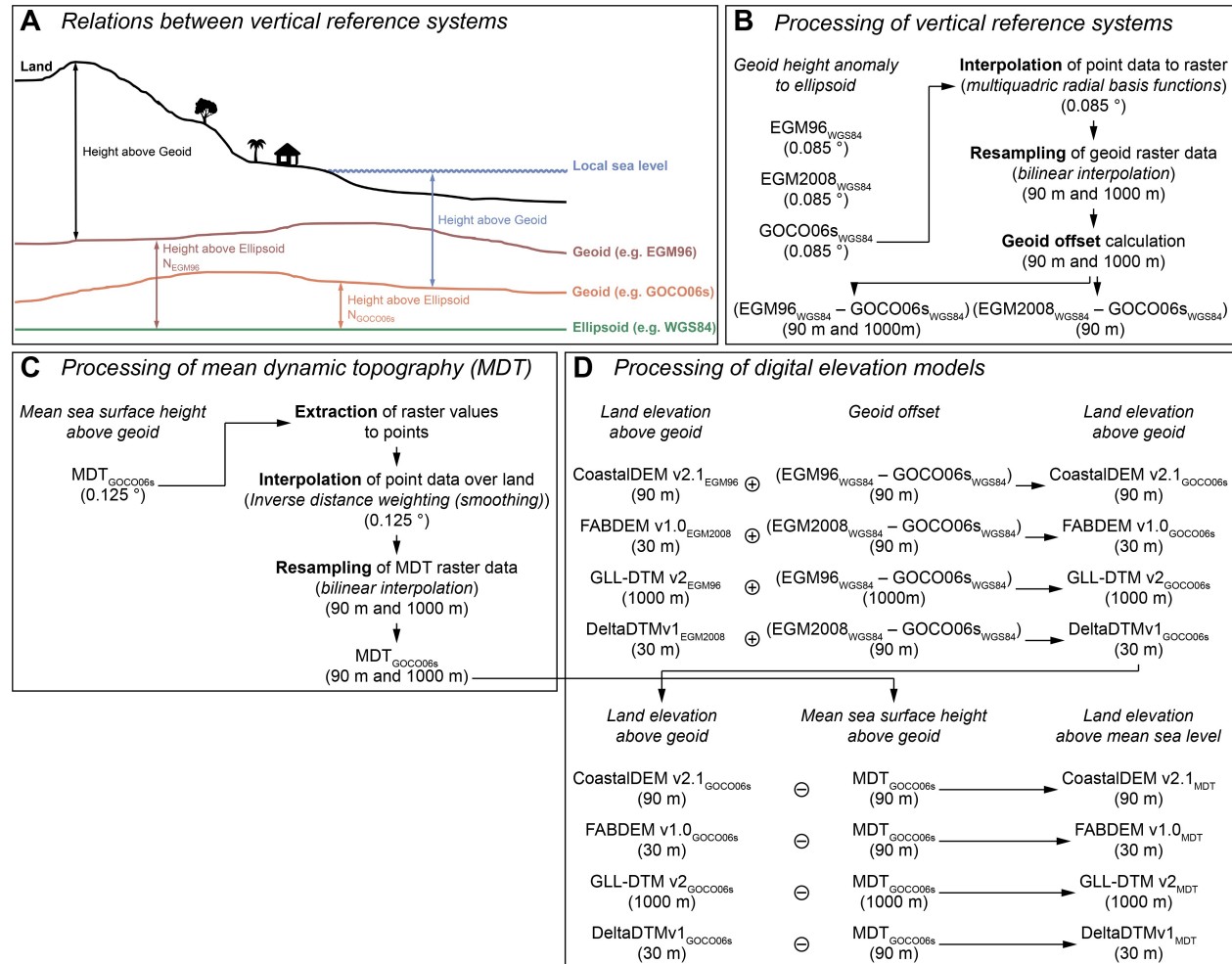

**Extended Data Fig. 2 | Relations between vertical reference systems and processing workflow of vertical datum conversion of elevation data.** **A**, Schematic overview of vertical datums. **B**, Calculation of offsets between vertical reference systems as prerequisite for vertical datum conversion of elevation data. **C**, Preparation of mean dynamic topography (MDT) data as prerequisite for vertical datum conversion of elevation data. **D**, Vertical datum conversion of elevation data from their original vertical reference system to the latest earth gravity model GOCO06s and to mean sea level as indicated by MDT.

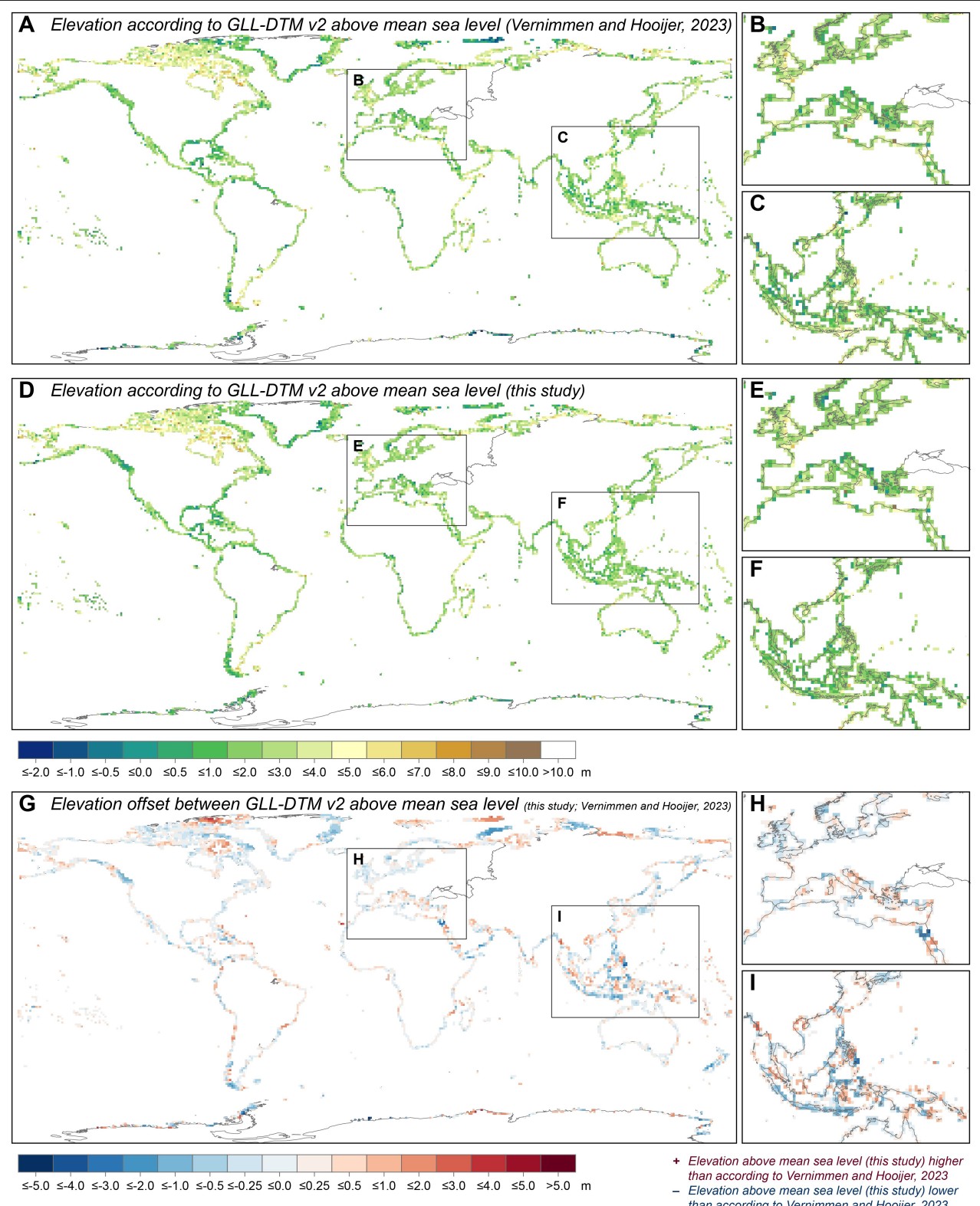

**Extended Data Fig. 3 | Global coastal elevation and discrepancies between sea-level-referenced elevation with and without a geoid conversion error.** Global coastal elevation according to GLL-DTM v2 above (**A**–**C**) mean sea level after refs. 21,42,53 and (**D**–**F**) mean sea level following this study (MDT HYBRID-CNES-CLS2022), as well as (**G**–**I**) the offset between both. The GLL-DTM v2 (refs. 21,42) contains a geoid conversion error (i.e. the EGM-DIR R4 geoid was assumed to equal EGM96, hence introducing the offset between both geoids (Extended Data Fig. 9) when applying the MDT data. In addition, the GLL-DTM v2 applied an, at the time already, outdated MDT dataset (i.e. CNES-CLS13 MDT[53]). The results were visualized using QGIS v.3.28.6 and shapefiles from ref. 51 (Open Government Licence v3.0).

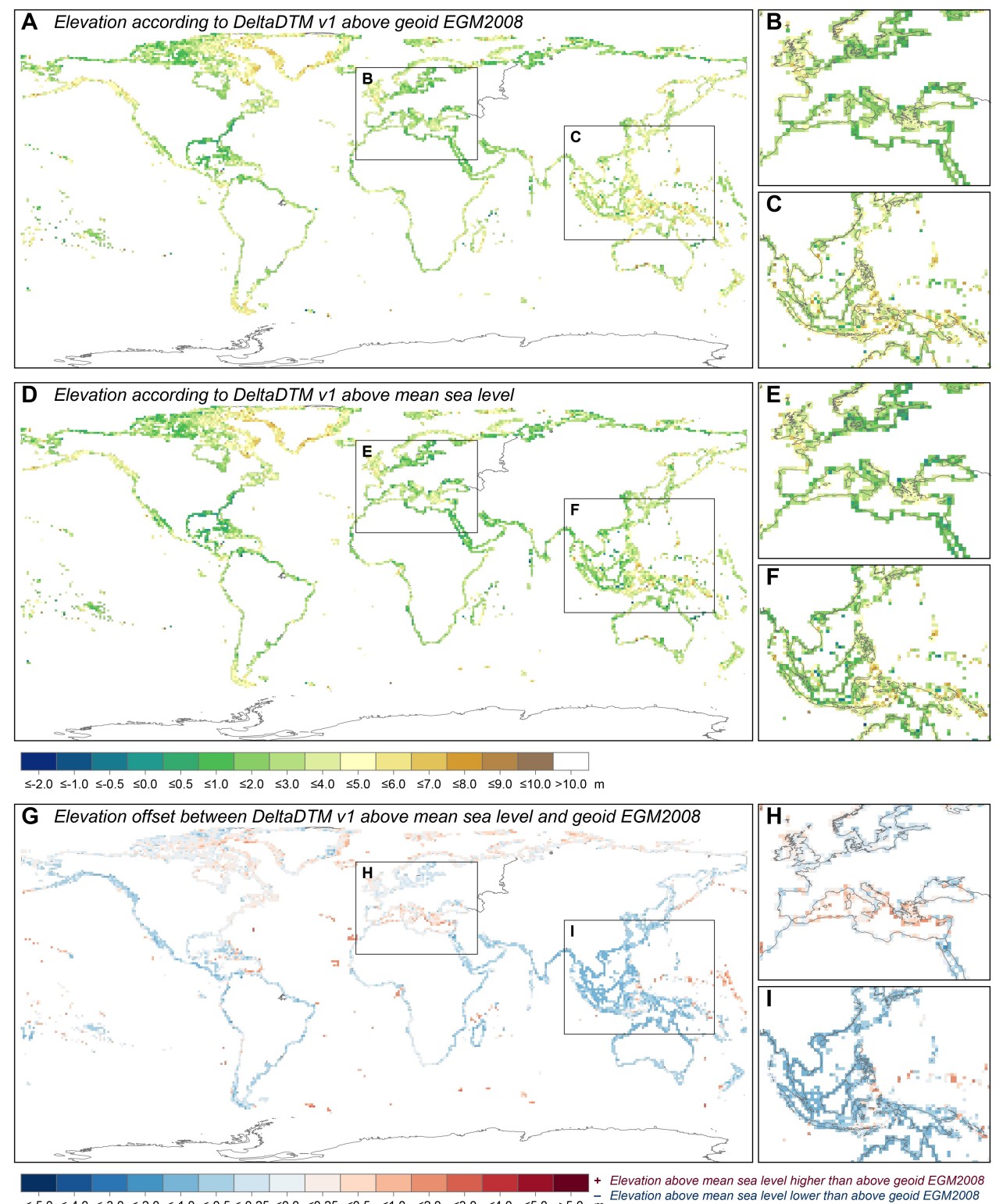

**Extended Data Fig. 4 | Global coastal elevation and discrepancies between geoid-referenced and sea-level-referenced elevation.** Global coastal elevation according to DeltaDTM v1 above (**A–C**) geoid EGM2008 and (**D–F**) mean sea level (MDT HYBRID-CNES-CLS2022), as well as (**G–I**) the offset between both. For visualization purposes, the spatial scale of the data shown was resampled to 1° using bilinear resampling while all statistics are given at 90 m spatial resolution. The results were visualized using QGIS v.3.28.6 and shapefiles from ref. 51 (Open Government Licence v3.0).

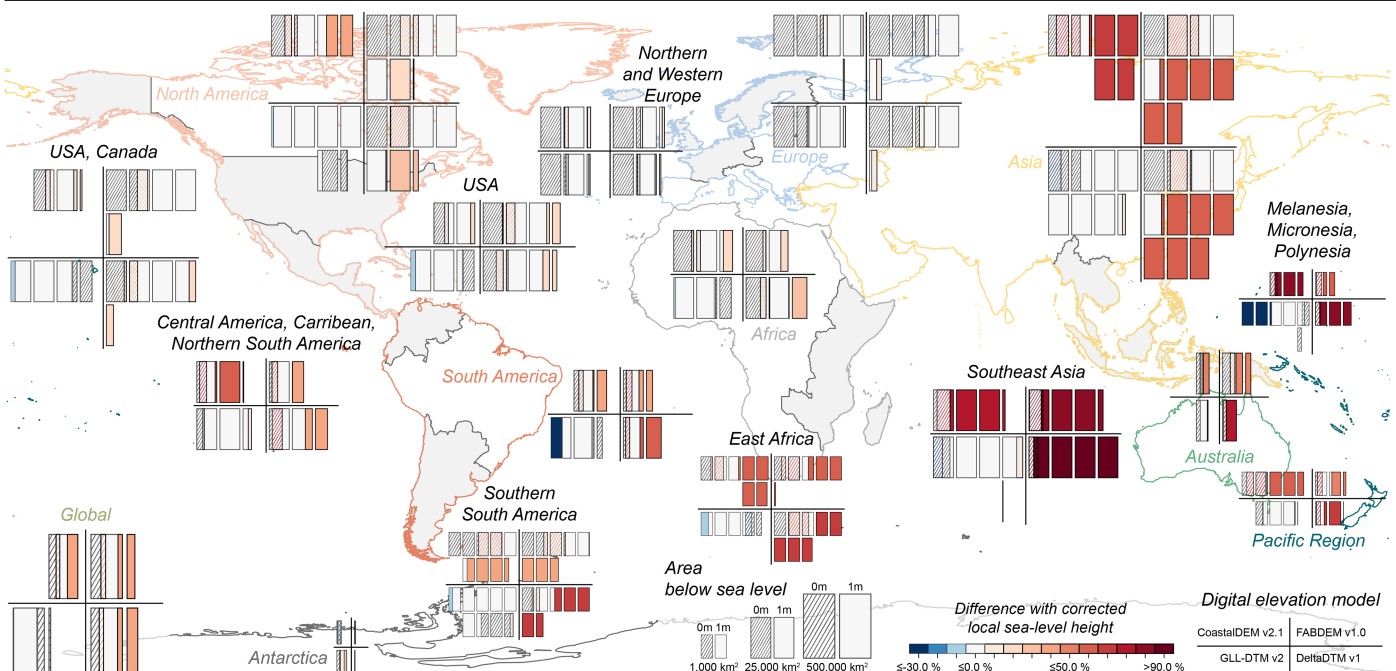

**Extended Data Fig. 5 | Difference in area falling below sea level following 1 m relative sea-level rise (RSLR) between commonly assumed and measured sea-level height.** Meta-analysis showing the impact of the most frequently observed errors in existing coastal hazard assessments, i.e. omitting or improper alignment of measured coastal sea level to land elevation, using modern, globally available digital elevation models (CoastalDEM v2.1 (ref. 40); FABDEM v1.0 (ref. 41); GLL-DTM v2 (ref. 42); DeltaDTM v1 (ref. 43)). Due to the widespread omission to tie elevation data to local sea level instead of global geoid models, area impacted by 1 m RSLR is 31% to 37% larger globally.

Maximum discrepancies of up to 94% (that is up to 93,800 km²) are observed along the coastlines of Southeast Asia where – if properly assessed – up to 100,300 km² will fall below sea level following 1 m RSLR. All statistics are computed for the respective spatial resolution of the DEMs. Colour outlines and grey-shaded areas indicate the regions and subregions used (Supplementary Fig. 8). All data related to this figure is included in Supplementary Data 3. The results were visualized using QGIS v.3.28.6 and shapefiles from ref. 51 (Open Government Licence v3.0).

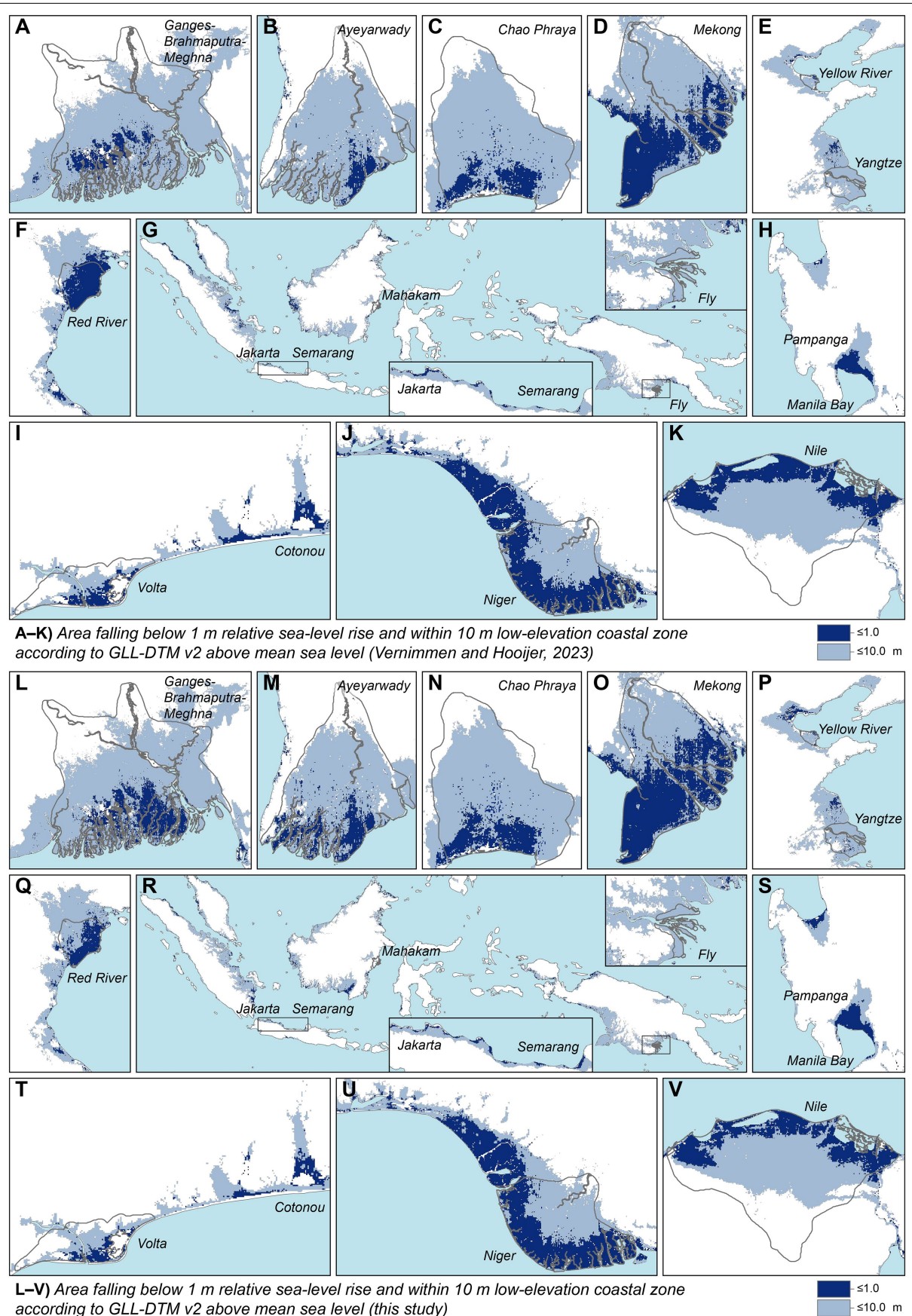

**A–K)** *Area falling below 1 m relative sea-level rise and within 10 m low-elevation coastal zone according to GLL-DTM v2 above mean sea level (Vernimmen and Hooijer, 2023)*

■ ≤1.0
■ ≤10.0 m

**L–V)** *Area falling below 1 m relative sea-level rise and within 10 m low-elevation coastal zone according to GLL-DTM v2 above mean sea level (this study)*

■ ≤1.0
■ ≤10.0 m

**Extended Data Fig. 6 | Assessment of major low-elevated coastal-deltaic areas around the world, showing area falling below sea level by 1 m RSLR and within the LECZ for sea-level-referenced elevation data with and without a geoid conversion error.** Impact of 1 m RSLR and LECZ based on GLL-DTM v2 referenced to (**A–K**) mean sea level after refs. 21,42,53 and (**L–V**) mean sea level following this study (MDT HYBRID-CNES-CLS2022). The results were visualized using QGIS v.3.28.6 and shapefiles from ref. 51 (Open Government Licence v3.0) and ref. 67.

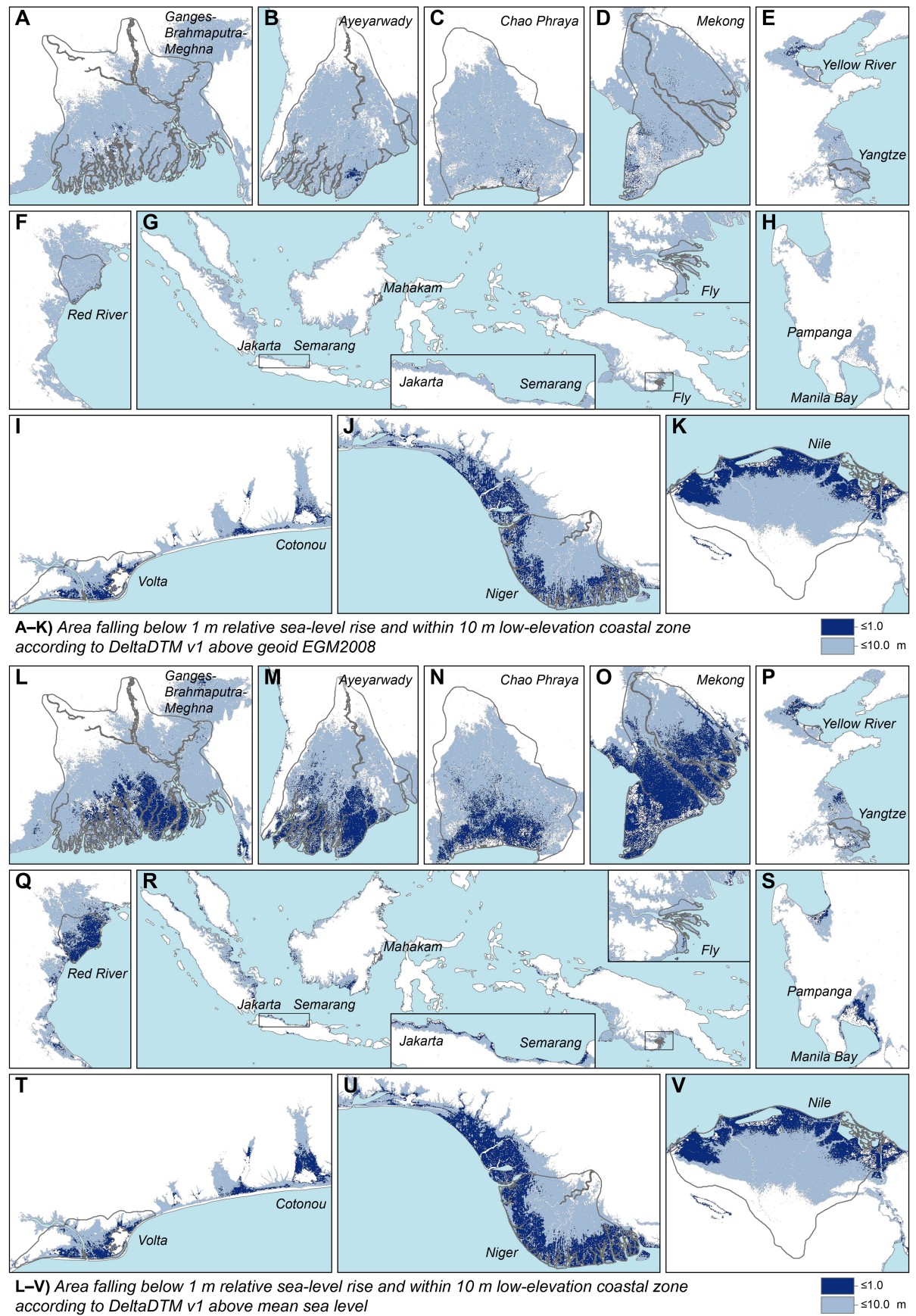

**A–K)** *Area falling below 1 m relative sea-level rise and within 10 m low-elevation coastal zone according to DeltaDTM v1 above geoid EGM2008*

≤1.0
≤10.0 m

**L–V)** *Area falling below 1 m relative sea-level rise and within 10 m low-elevation coastal zone according to DeltaDTM v1 above mean sea level*

≤1.0
≤10.0 m

**Extended Data Fig. 7 | Assessment of major low-elevated coastal-deltaic areas around the world, showing area falling below sea level by 1 m RSLR and within the LECZ for geoid-referenced and sea-level-referenced elevation data.** Impact of 1 m RSLR and LECZ based on DeltaDTM v1 referenced to (**A–K**) EGM2008 and (**L–V**) mean sea level (MDT HYBRID-CNES-CLS2022). The results were visualized using QGIS v.3.28.6 and shapefiles from ref. 51 (Open Government Licence v3.0) and ref. 67.

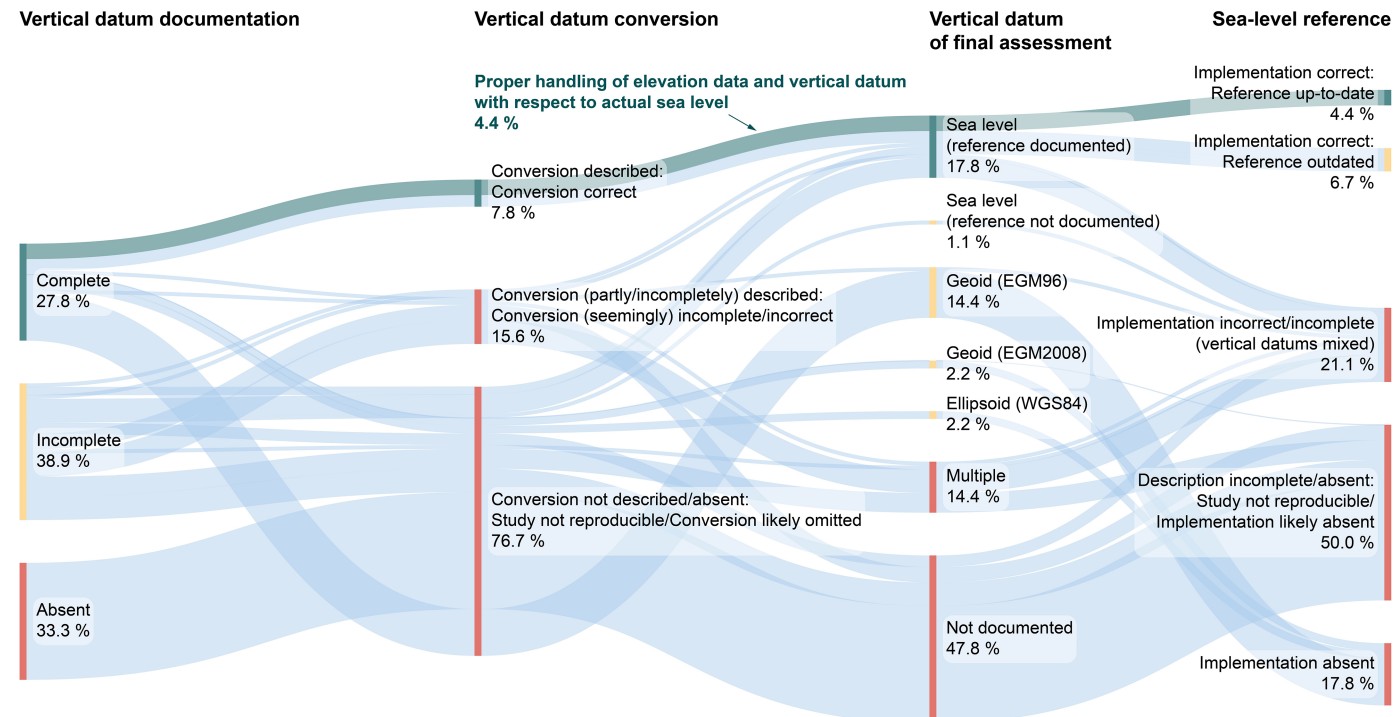

**Vertical datum documentation**

Complete
27.8 %

Incomplete
38.9 %

Absent
33.3 %

**Vertical datum conversion**

Proper handling of elevation data and vertical datum with respect to actual sea level
4.4 %

Conversion described:
Conversion correct
7.8 %

Conversion (partly/incompletely) described:
Conversion (seemingly) incomplete/incorrect
15.6 %

Conversion not described/absent:
Study not reproducible/Conversion likely omitted
76.7 %

**Vertical datum of final assessment**

Sea level
(reference documented)
17.8 %

Sea level
(reference not documented)
1.1 %

Geoid (EGM96)
14.4 %

Geoid (EGM2008)
2.2 %

Ellipsoid (WGS84)
2.2 %

Multiple
14.4 %

Not documented
47.8 %

**Sea-level reference**

Implementation correct:
Reference up-to-date
4.4 %

Implementation correct:
Reference outdated
6.7 %

Implementation incorrect/incomplete
(vertical datums mixed)
21.1 %

Description incomplete/absent:
Study not reproducible/
Implementation likely absent
50.0 %

Implementation absent
17.8 %

**Extended Data Fig. 8 | Sankey Diagram for additionally reviewed literature (Supplementary Data 1) on sea-level and coastal elevation data documentation and their proper use and alignment in coastal hazard assessments (similar to Fig. 1).** The Sankey diagram for the results of this study was created using SankeyMATIC (https://sankeymatic.com/).

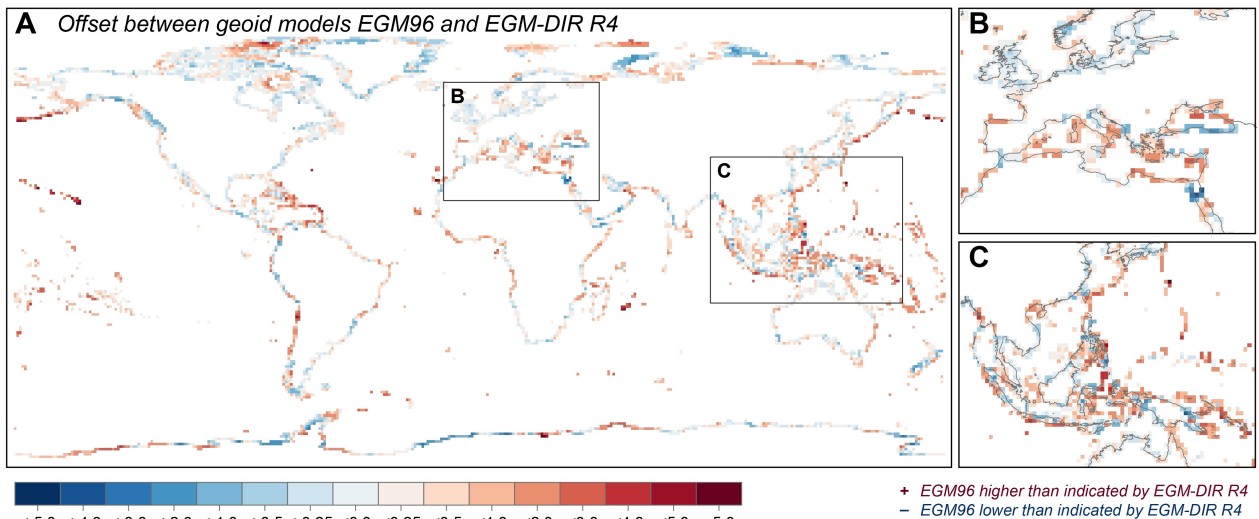

**Extended Data Fig. 9 | Offset between EGM-DIR R4 and EGM96 geoid models across the globe.** We show this discrepancy to quantify potential residual errors stemming from incomplete vertical datum conversion (erroneously assuming the EGM-DIR R4 geoid of the CNES-CLS13 MDT[53] to equal the EGM96 geoid and thereby omitting this geoid-to-geoid datum conversion). This error is present in ref. 21 who based their processing approach on ref. 19 (presumably containing the same conversion error, but this is not irrefutable due to incompleteness of conversion documentation). This error is likely present in other papers (e.g. ref. 45) that follow the described approach of MDT implementation. The results were visualized using QGIS v.3.28.6 and shapefiles from ref. 51 (Open Government Licence v3.0).

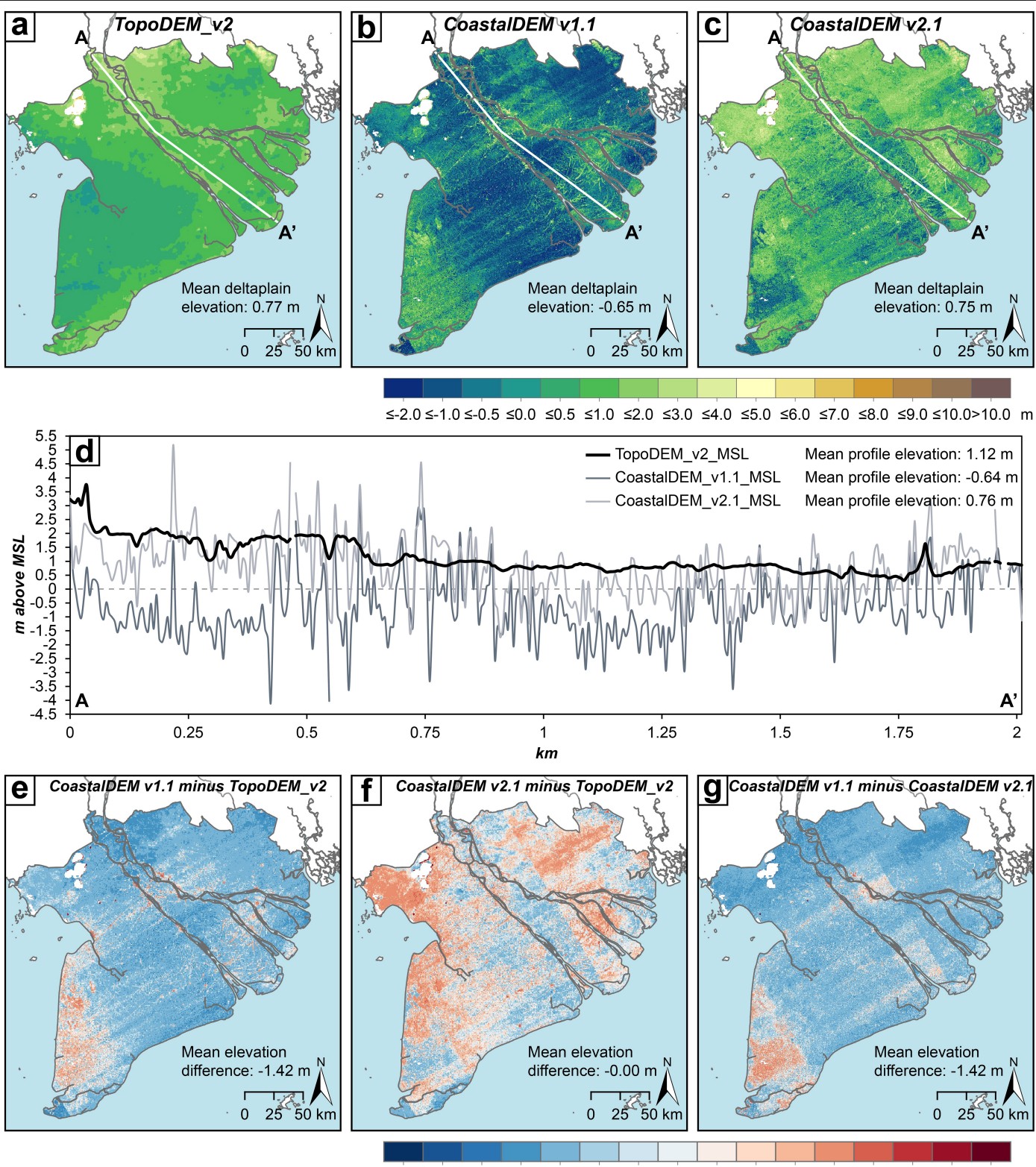

**Extended Data Fig. 10** | See next page for caption.

**Extended Data Fig. 10 | Example of incompatible North–South transferability shown by the comparison between CoastalDEM v1.1 (ref. 48), of which the neural-network approach used to create the CoastalDEM was Global-North-trained (the United States) and Global-North-validated (Australia), and the local TopoDEM_v2 (ref. 38), together with CoastalDEM v2.1 (ref. 40).** CoastalDEM v1.1 and v2.1 (90 m spatial resolution, respectively) were acquired from Climate Central (https://go.climatecentral.org/coastaldem/). The vertical datum was converted to local MSL by properly applying the most recent MDT data[44] as described in the Methods section (see also Extended Data Fig. 2). TopoDEM_v2 was created by interpolation of topographical elevation points referenced to the same local MSL and validated locally using independent geodetic measurements[38]. TopoDEM_v2_MSL provides a mean delta plain elevation of about 0.77 m, while CoastalDEM_v1.1_MSL documents the mean elevation of the delta plain more than a meter lower, at about −0.65 m. This showcases the neural network to correct the SRTM DEM, which was trained on US coastlines and validated in Australia, to be performing very inadequate in other places of the world, thereby, at least for the Mekong Delta, largely overestimating the amount of people exposed to high water levels as reported in ref. 68. These issues were largely addressed in CoastalDEM 2.1, which was released in 2021 (ref. 40). However, it still suffers from inaccuracies due to sensing artefacts. The elevation profiles of CoastalDEM v1.1 and v2.1 over transect A–A' were binned to 500 m (median) for visualization purpose to match the spatial resolution (500 m) of TopoDEM_v2. The results were visualized using QGIS v.3.28.6 and shapefiles from ref. 51 (Open Government Licence v3.0).