## [Peer Review file · Nature]

Sea level much higher than assumed in most coastal hazard assessments

Corresponding Author: Ms Katharina Seeger

Version 1:

Reviewer comments:

Referee #1

(Remarks to the Author)

In this important technical contribution, the authors perform a systematic review of literature assessing coastal population exposure, both globally and regionally, and identify that the large majority of this literature fails to adequately document the conversion between a geoid reference frame and a tidal reference frame, with the potential for consequent substantial impacts on estimated exposure.

While I have not replicated the authors' analysis, they have documented it well and provided their data and code consistent with open research practices, and it seems important to highlight this gap in the literature and to introduce procedures to avoid it in the future.

I need more convincing of the broader impact of this gap, however. My concern is that, by focusing on one source of error, they are not contextualizing that source of uncertainty relative to other sources of uncertainty in the underlying study.

To evaluate the impact of this gap, we need to partition the literature by application. For regional studies that constitute 90% of the literature review, it is not clear what the best metric to evaluate the import of the discrepancy is. It would be helpful to walk through some specific case studies that highlight how this error, relative to other sources of uncertainty, impacts the final results that the authors of the critiqued study think are important.

For the global studies, this metric is clearer, and the authors focus upon it: namely, number of people exposed at a global scale, under different 'scenarios' (1 m RSL rise or 10 m LECZ height.)

However, the authors do not highlight a well-cited erroneous study to make their point about the import of their results; rather, they do their own calculation, picking a comparison (assuming geoid = MTL) that maximizes the resulting discrepancy. It would be helpful to contextualize the discrepancy relative to actual range of numbers reported in the literature, based on errors as actually implemented by those studies. (E.g., Depsky et al., 2023, includes an incompletely documented conversion between geoid and tidal datums, thus one would not expect the error associated with this study to be as large as Lincke et al., 2022, which does not consider the datum offset correction to sea level.)

Toward this end, it might be helpful to have a version of Figure 1 that focuses on this easier-to-compare set of global studies, and how different outcomes on the right-hand side of this Sankey diagram are expected to correspond to differences in global exposure numbers. If possible to calculate, a histogram of errors for the global studies would be useful.

Notably, IPCC AR6 (Cross-Chapter Paper 2) relies on Haasnoot et al., 2021 (doi:10.1016/j.crm.2021.100355) for their statement that about 11% of global population currently lives within 10 m of MSL, which compares with the 12.3-13.6% estimate in this paper. This would be a natural specific point of comparison, but this paper does not appear in the authors' analysis.

Moreover, despite their focus on this one source of uncertainty, the authors take as given a single source of gridded population data, which is surely another substantial source of error. (How well do we really know where people on the planet live, at a 100 m scale?) At a minimum, the authors could compare results from WorldPop and the slightly lower resolution LandScan (or possibly the new, comparable resolution LandScan HD), as well as discuss the literature comparing these and other population data sets. It would be important to know how the error introduced by imperfect tidal corrections (across the range of imperfections actually in the literature, as opposed to only in the extreme case where it is not done at all) compares to uncertainty in population distribution.

The authors also omit one important calculation: namely, for each data set (DEM x population), the population estimated as being below sea level today. In economic modeling studies (e.g., Diaz, 2016; Depsky et al., 2023) it is clear that the combination of DEM, tidal corrections, and population used identifies a population who are below sea level under present conditions. In some cases, this is correct, and the identified population is below sea level but protected (e.g., in the Netherlands); in other cases, this reflects data errors. These economic models allow these people to adapt (move or protect) before the start of other calculations, adjusting for this bias in the underlying representation of the current state of the world.

Of interest to me, therefore, is the *additional* population below sea level with 1 m RSL rise relative to 0 m RSL rise. (Similarly, with the LECZ, the additional population is of greater interest.) It would be helpful to know how much the errors the authors focus on affect this incremental quantity. If the errors on which the authors focus affect primarily the estimate of the currently exposed population as opposed to the additional population exposed, this has less effect on economic analysis (because the marginal increase in exposure with sea level rise may be unchanged) than if the errors primarily affect the additional exposure with 1 m RSL rise.

(Remarks on code availability)

Referee #2

(Remarks to the Author)

Review of Seeger and Minderhoud, "Sea-level much higher than assumed in most coastal hazard assessments"

The manuscript first highlights discrepancies in sea-level rise assessments as they affect low-lying coastal landscapes. The findings are based on an extensive literature review wherein inaccuracies in sea-level rise related flooding/inundation are highlighted largely due to incorrect or absent datum referencing and transformations. The authors then conduct their own global analysis of errors/discrepancies using uncorrected geoid and ellipsoid results and comparing them against correctly transformed and corrected global datasets to demonstrate how these errors can manifest in underpredictions, particularly in southeast Asia and the southern hemisphere more generally where the geoid tends to be more inaccurate as compared with the northern latitudes. The authors raise concerns that the inclusion of these studies in larger scale assessments (e.g., the IPCC) may underrepresent risk to people and populations that may be affected, particularly in low lying areas such as deltas and coastal plains. The manuscript concludes by suggesting several ways awareness and correction can be used to address these issues, including more explicit guidance from journals to peer reviewers on aspects to be mindful of when evaluating these studies, as well as broader awareness of standards and errors within the scientific community.

In sum, the authors highlight what appears to be a pervasive and problematic issue that needs to be brought to the attention of the scientific community, and a high impact journal like Nature is a good place to do that. Although the manuscript appears scientifically robust and I think will be of interest to many in the field, I have major concerns with regards to caveats and limitations in the work need to be more transparently discussed. As a consequence, much of the paper currently reads as sensationalized, with the same findings mentioned multiple times, and without adequate context given to them. Moreover, the paper could also be reframed around some of their conclusions—the factors that have led to the abundance of seemingly "flawed" studies—that would give it a much more engaging and constructive impact within the scientific community.

Specifically, I do not recommend publication until and unless these issues are comprehensively addressed:

- 1) Literature screening issue: In the literature review, which uses PRISMA as a screening tool, it appears there may be an error with one of the exclusion/inclusion literature screening elements included in the supplement (p 18-19 in supplement). Both internal and external screening lists include 'focus on coastal land and coastal sea level' as the first criterion listed; if this is not an error, the authors need to explain to the reader why this criterion is included in both. I highlight this because I had questions as to why some well-known studies that seemed they would be relevant and made the first order screening were ultimately excluded from the review, and I could not resolve this based on the information in these categories, making me question the reproducibility of the approach.
- 2) Population findings: Better justification/explanation needs to be provided for the population dataset that is used in the global reanalysis. Because conclusions rely so heavily on this dataset, the reader needs to be clear on a) why this was selected, above any other, to evaluate population impacts; and b) the limitations in basing conclusions on one global dataset (specific to 2020) and/or any caveats associated with associated results/outcomes (e.g., noting that a 2020 population estimate is being used to compare against a future condition that is unlikely to be reached for many decades).
- 3) Novelty: By conducting their own global meta-analysis, the authors generate helpful conclusions to illustrate how underprediction may affect many coastal locations. While this was undoubtedly a considerable undertaking, the approach used is not novel (the code evaluated showed two instances of simple raster subtractions for a GIS environment) and benefitted from the use of many global datasets that have recently been produced. The novelty of the work comes not in the approach but in the findings from the literature review, that they then use the meta-analysis to help to support; again however, transparency regarding meta analysis limitations is needed, as well as better distinctions made between their findings and those from their literature screening (see comments on abstract below).
- 4) Argument framing: The paper would be a stronger and more productive contribution to the scientific discourse (and the

studies they critique) if the authors offered more context on a) poor data quality and/or paucity and the advances new global datasets, like those used in their reanalysis provide (and that were arguably unavailable during the time many of the studies published in the last five years they critique were conducted); b) the tradeoffs their results might have with more localized studies that can be produced at higher resolutions and/or include nuance that a global approach cannot; c) their decision to use a 1m SLR increment in their analysis, and how that compares to what the other studies they reference have used in their sea-level rise impact assessments. I especially liked some of the recommendations made for improvement for future work in the conclusions (e.g., constellation issues adding to complexity of transformations; peer-review guidance, including explicit instructions for datum review); stating these earlier in the paper could be used to more productively frame the contributions this manuscript offers.

5) Limitations of inundation approach: The authors do not address the limitations of bathtub or inundation approaches in general (of which theirs is one), which have been well documented in the literature as underrepresenting dynamic coastal change present in many of the low-lying areas in which they focus (see: Passeri et al., 2015). At the very least, their results should be caveated with this information.

6) Evaluation of use of flawed studies: Finally, it seems a missed opportunity that the authors did not conduct a comparison of how many of the flawed studies that they highlight are cited in larger scale assessments (for example, the most recent IPCC) to provide substantiation for their claims that these studies often form the basis for larger assessment findings. Even the evaluation of just one assessment as an exemplar would be of benefit.

Line Items:

In the abstract, as well as in many places in the text, sweeping generalizations about the flawed studies they cite are made, which I believe are instead based on the findings from their global meta-analysis. For example, Lines 18-20 discuss how evaluated studies improperly handled data and transformations, but lines 20-23 seem to refer to findings based on the meta-analysis the authors themselves conducted, and this distinction should be made more clearly. Similarly, in line 25, the authors state: "67% more people (increase of 88 million to 132 million people) will fall below sea level following 1 m of relative sea-level rise," but it is unclear this increase is in comparison to what; again I'm inferring this is the comparison of their global reanalysis findings, but if so, this should be stated for context in the abstract.

Line 114: "Not accessible or demonstrably.."

Line 135: correct to "demonstrably"

Line 190: past "relative" sea-level changes?

Line 194: "of actual coastal exposure necessitate the accounting of relative sea-level changes..."

Lines 274: These statements have been made at least two other times in the article and the repetition is getting tedious.

Line 290: Can you say something more about how these data gaps might get filled?

Line 295: Replace "suffer from" with "be affected by" as more neutral terminology.

Lines 333-334: See concerns above regarding the secondary screening.

Line 375: What are "the very coastal stretches"?

Line 380: Replace "allows to investigate" with "can be used to adjust".

Lines 401-405: This is verbatim the same sentence as the last in the paragraph above; is this intentional?

Line 416: Awkward phrasing: "a distance threshold for which we deem this approach"

Line 418: Replace "We considered to transpose" with "We transposed".

Line 431: Elevation threshold of -7m above mean sea level?

Line 448: Again, see comments above regarding justification and caveats around using this dataset and conclusions drawn from it.

References:

Passeri, D. L., Hagen, S. C., Medeiros, S. C., Bilskie, M. V., Alizad, K., & Wang, D. (2015). The dynamic effects of sea level rise on low-gradient coastal landscapes: A review. *Earth's Future*, 3(6), 159-181.

(Remarks on code availability)

The code evaluated showed two instances of simple raster subtractions for a GIS environment; it is a usable resource in current form though very straightforward and something most that use GIS could create very easily.

Referee #3

(Remarks to the Author)

I am pleased about the opportunity to review this interesting manuscript. Building on a large structured literature review and own modeling work reproducing errors found in existing studies, the paper finds that the majority of the current literature on sea level assessments and sea level rise show severe shortcomings in their methodological reporting and data use. The paper therefore argues to find that sea levels are higher than assumed in most coastal hazard and impact assessments. Interestingly, the paper finds that the discrepancies are widest in many regions of the Global South which suffer from particularly high sea level rise exposure but also from data scarcity. Looking into the future, the paper therefore argues based on its results that current sea level rise impact assessments underrepresent the actual exposure and risk from sea level rise, meaning that more locations and more people will be at risk than previously assumed.

As such, the paper provides a daring approach and innovative findings. However, I feel that the paper suffers from a few substantial shortcomings in terms of its coherency, the way it presents its data and the conclusions that are drawn from it. In my view, these points would need to be revised before the manuscript can be considered for publication.

In lines 38 to 40, the authors argue that "less than 1% of the evaluated literature provided thorough documentation and handled coastal sea-level and elevation data properly". I find this statement problematic, inconsistent and misleading – at the very least imprecise. How can the authors know that studies allegedly did not handle data properly, if thorough

documentation was lacking? Two sentences further, a similar formulation is likewise very problematic. The later parts of the paper provide more precise numbers on the reporting shortcomings in the evaluated papers. Nevertheless, the main concern I raise here remains. Figure 1, for example, shows the large proportion of studies with documentation shortcomings. But how can the authors deduce from there, that all these studies have errors and only one out of the 385 evaluated studies contained no conversion and alignment errors (lines 122-123)? I cannot follow this conclusion and find the negative interpretation very problematic – maybe/potentially invalid. In fact, the authors themselves argue in the conclusion that in many cases, the missing documentation makes it impossible to evaluate the correctness of the analysis undertaken in the respective paper. However, this should be more carefully separated from the sweeping allegation of “widespread misuse” which is raised two sentences earlier.

Rather than repeating the numbers of the Sankey charts provided in Fig. 1, the text should concentrate on explaining the problematic convolution of missing documentation and the assumption on erroneous data treatment.

For non-technical readers, a conceptual figure or infographic explaining how the discrepancies between sea level and most-used geoid height arise might be helpful.

Also, it would be helpful to explain more clearly on how two findings go together: Figure 2, for example, suggests that there are positive and negative discrepancies between coastal sea level and widely used geoids, particularly EGM96. This applies also to regions identified as most problematic such as Southeast Asia. However, in the later course of the paper, only the findings regarding the alleged underestimation of current sea levels are highlighted, arguing coastal sea level worldwide is on average 0.2-0.5 m higher than assumed in coastal hazard assessments with errors.

In addition, the role of data scarcity would deserve a more considerate discussion. On the one hand the authors argue that there is a widespread misuse of data, generating most severe discrepancies in areas of the Global South which also suffer from high data scarcity. However, the question whether and to which extent it is justified to talk about the “misuse” of data in situations in which data is insufficient or not available is not discussed. But this discussion is of great relevance for the overall direction of the paper and its messaging: While the paper currently accuses the overarching part of the research community for the alleged misuse of data, it might be the data scarcity that is to blame. Taking this perspective would of course shift the conclusions and recommendations of the paper quite a bit.

The discussion on the potential reasons behind the alleged misuse should be strengthened. The authors put forward the interesting hypothesis that Global North authors might be overconfident in the general validity of the geoid models used, based on the fact that in the Global North those models perform well. Here, the authors should further elaborate on whether they have tested this hypothesis as part of their work.

(Remarks on code availability)

Referee #4

(Remarks to the Author)

General

This study offers timely and valuable insights for researchers in the field, as well as the broader scientific and policy-making communities. Through a systematic/meta-analysis of peer-reviewed publications using the PRISMA framework, it looks to have identified key methodological and empirical gaps in sea-level and hazard assessment research. These gaps highlight the potential for serious misinterpretations and poorly informed decision-making, emphasizing the need for closer scrutiny and more rigorous approaches in future studies. The manuscript is coherent, well-written, and supported by comprehensive supplementary materials, including data extraction tables, raw data, literature search protocols, codes, and additional figures and tables. Its rigorous approach and broad relevance make it a significant contribution, suited for publication in Nature. However, a few minor issues remain that warrant clarification and further justification to strengthen the paper's quality.

Specific comments and issues for clarification

Title:

- The title is informative and effectively conveys the main message of the paper. However, it would benefit from explicitly highlighting that sea level has been underestimated, as this is a key finding clearly demonstrated in the study. Using the term “much higher” may be misleading, since no clear threshold is provided to determine whether the values fall within that range, and no evidence on statistical significance was provided.

Abstract:

The abstract is also informative, but it could be improved by including a brief overview of the approach followed to synthesize evidence. The novelty of this synthesis lies in its data handling and critical evaluation of the methodologies commonly used in previous studies, an aspect that should be more clearly emphasized. For example, on line 17, instead of stating here we.... It would be better if the methodology employed was stated e.g. Using a systematic meta-analysis of peer-reviewed publications we reveal.....Also on line 70 e.g. through a systematic review of 385 papers contemporary methods for SLR impacts and coastal assessments were evaluated, finding that these crucial steps were often not considered

Reflection on the key message – strong criticism on the reviewed papers!

- The claim that all (except one) assessed peer-reviewed publications on sea-level rise (SLR) and hazard assessments are erroneous and invalid seems overly bold. These studies were conducted under diverse natural and human-influenced contexts. The systematic review and meta-analysis attempts to compare them across varying spatial and temporal scales which introduces inherent complexity and necessitates careful contextual consideration, although the concern on proper

application of standardization approaches is valid. Details of the contexts that were considered for each paper could be better described.

- Advances in modelling approaches have improved uncertainty handling, but the use of newer models alone does not automatically invalidate earlier work. Similarly, the finding that about 60% of the reviewed studies lacked documentation on datum conversion is indeed a methodological shortcoming, but it does not necessarily render their findings invalid. This should be highlighted in the paper
- The review rightly highlights challenges posed by data scarcity, lack of methodological clarity, transparency, and reproducibility, which can lead to underestimation of SLR. It also raises a critical concern about applying Global North models (developed and tested in the data-rich regions) in Global South contexts without adequate adaptation to local contexts and dynamics. This practice can lead to serious misinterpretations and poorly informed decisions and deserves closer scrutiny in future research.
- As mentioned in the conclusion section, while this review identifies significant gaps, completely dismissing earlier studies, and, by implication, questioning the integrity of the peer-review process in general seems inappropriate. A more balanced approach would be to advocate for systems that enhance assessments and projections with localized, real-time data, and to emphasize the importance of addressing methodological weaknesses, such as lack of clarity, uncertainty quantification, and reproducibility, issues that can have far-reaching consequences for decision-making and policy. The conclusion would be better framed as identifying current gaps and forward looking to improvements that could be made by authors, reviewers, editors etc. This would create a more useful and positive outcome of the work.
- Finally, while presenting strong criticisms, the review does not clarify whether the assessed studies acknowledged their own methodological and contextual limitations. Recognizing such boundaries reflects scientific integrity and transparency, helping to guide informed decisions based on the findings. If some of the reviewed studies included such disclaimers, this should be acknowledged, as it fosters constructive dialogue and progress in the field.
- One final question for the authors: Do you identify any limitations in your review that may have influenced your findings or conclusions? It be worthwhile to include a separate section discussing these limitations

Methodological comments/questions:

- The methodology is comprehensively described and there is good supporting documentation to enable transparency which give confidence in the results. There are however, a few additional details and clarifications that are needed
- Could you elaborate on whether and how you accounted for possible biases associated with the study period and geographic location of the included studies in your meta-analysis? These factors may have influenced the outcomes and warrant discussion.
- Since the literature review was conducted following the PRISMA guidelines, it is recommended to include a PRISMA flow chart to clearly illustrate the screening and selection process.
- The methods section indicates inclusion of review papers alongside peer-reviewed articles and data papers to narrow the dataset after 2008. However, review papers are usually excluded in systematic reviews to prevent double counting of primary research. A recommended approach is to extract and include the primary studies referenced in those reviews. This could also help reduce the volume of documents and possibly enable expanding the publication timeframe, unless the cutoff at 2008 is specifically justified. Could you please elaborate on the 2008 cutoff and why this was chosen?
- Since SCOPUS was used as the only search engine, it would be helpful to provide justification for this choice. Considering that other databases and repositories exist, please explain how SCOPUS sufficiently captures the breadth of relevant literature for your study.
- The review process, evaluation procedures, and the codes used are clearly and appropriately described; however, there are some issues that need correction. For example, the inclusion criterion “focus on coastal land and coastal sea level” is also listed under the exclusion criteria, which is contradictory. This needs to be changed and clarified
- The use of “maybe” during the screening of titles, abstracts, and keywords is appropriate, as uncertain cases can be passed on to full-text analysis for detailed review. However, it should be stated in the paper that the maybes were carried forward for full text review due to ambiguity
- Line 83, the sub-title needs to be improved as “Most coastal hazard assessments contain errors or are not reproducible” or otherwise!
- Line 103 Is 14% ample? how was this judged? I would avoid this descriptor and just state that 14% of evaluated studies were from Africa
- Line 131-132, the statement begins with a numeral (%), which is not generally recommended, and the statement from line 132 – 137 is quite long and it would be good to break it down for more clarity.
- Line 201, the sub-title “Sea level higher than previously assumed in most coastal hazard assessments” is almost same as the main title, thus, may need some kind of rephrasing to specifically align it with the specific contents of the sub-section, or change the general title of the paper (as also suggested above) to reflect the entire content of the paper
- Line 218, change the word “effected” to “affected” to make it more formal.
- In the meta-analysis, it would be advisable to provide standard errors or confidence intervals (if possible) when discussing the average estimates of the coastal sea level values
- Line 240, figure 4 are the grey outlines unable to be calculated? This needs stating
- Line 319 If the guidelines were followed this should be identified as a systematic review, i.e. a systematic adhering to the standards of the Preferred Reporting Items for Systematic Reviews and Meta-Analyses (PRISMA) process was conducted
- Line 328 some examples of the exclusion of certain subject areas, keywords here would be useful and need full detail in supplementary
- Line 334 Who undertook the screening? Both authors? Initials should be provided, i.e. screen was undertaken by KS. Was consistency assessed? Which would be best practice and if so how was it judged. Details are needed on this
- 349 Again, who conducted the evaluation and how was consistency ensured if there was more than one person undertaking this?
- Line 350, Could you clarify the purpose of including the >90 papers accessed outside the search engine? As they were not

incorporated into the statistical analysis, it would be helpful to either justify their inclusion and identify how they were used or omit this information to avoid confusion.

- In the supplementary under the exclusion criteria it is stated that $n = 326$ (4.5%) met the scope "less well" how was well and less well determined? This seems strange, either the papers met the criteria, did not, or it was unclear. At 1st screening perhaps it is better phrased as unclear and then explained that these were carried through to 2nd screening. Is this the same as the "maybe" papers mentioned in the main document? If so then please align the terminology
- Vertical datum documentation evaluation – how was completeness judged? What documentation did you need to see for this to be judged complete? Needs to be precise
- Finally, to address some minor editorials, it is advisable to strictly follow the journal guidelines while revising the manuscript. For example, the captions and the main text are mixed up in most cases, which obscures the flow and clarity, and it is also good to clearly indicate the results and/or discussion sections!

(Remarks on code availability)

The code is reported openly and transparently. A few details highlighted above are needed to report the methodology conducted more transparently e.g. how was completeness judged

Referee #5

(Remarks to the Author)

I co-reviewed this manuscript with one of the reviewers who provided the listed reports.

(Remarks on code availability)

The codes used to evaluate and synthesize the reviewed literature are included in the supplementary materials to ensure reproducibility. A few additional comments are noted in the reviewer report I co-reviewed for this paper.

Version 2:

Reviewer comments:

Referee #1

(Remarks to the Author)

I thank the authors for their serious consideration of my feedback. My overall assessment remains the same: I think this article is certainly an important technical contribution, but I am not entirely convinced of the broader import of this gap. Thus, I think it is an editorial judgement call as to whether this paper is a good fit for a journal like Nature, or would be better suited for a journal like Nature Geoscience.

For example, in response to reviewers' recommendations, the authors now include sensitivity analyses with respect to different population data sets. For studies that do not correctly convert from the geoid to MDT, the problem is certainly large: across DEMs and population data sets, the number of additional people exposed to 1 m of SLR grows from 34-49 million to 77-132 million from correcting this error: clearly a first order error.

For studies with geoid conversion errors, the case is less convincing: if I am interpreting the results correctly, the number of people grows from 69-108 million to 77-122 million: not insignificant, but clearly a second order error compared to uncertainty in population distribution.

Regardless, I do think the authors need to be more precise in their critique of the literature and statements about implications. For example, they write: "It is concerning that many of the evaluated studies are used to underpin IPCC SLR impact reports (1,19,45,53)." Since the IPCC does not produce 'SLR impact reports,' I was curious as to what they meant by this. Ref. 1 and 53 are to the entire of the IPCC AR6 Synthesis Report and Working Group 1 report. The Working Group 1 report does not assess impacts, so I am unclear as to the relevance.

The Synthesis Report does state "The low-lying coastal zone is currently home to around 896 million people (nearly 11% of the 2020 global population), projected to reach more than one billion by 2050 across all five SSPs." This is the sole relevant sentence, so I am not sure I would call these a "SLR impact report" either. This sentence is based upon the AR6 WG2 Cities by the Sea CCP, which in turn cites Haasnoot et al. (2021) and the IPCC SROCC.

It seems to me very unlikely that the WG2 statement "In 2020, almost 11% of the global population—896 million people—resided in C&S within the low-elevation coastal zone (LECZ; coastal areas below 10 m of elevation above sea level that are hydrologically connected to the sea; Haasnoot et al., 2021b), a figure which will potentially increase beyond 1 billion by 2050 (Oppenheimer et al., 2019)" will have a substantially different policy import than the authors' implied proposed replacement, "In 2020, about 12-14% of the global population—970 million to 1.07 billion people—resided in C&S within the low-elevation coastal zone (LECZ; coastal areas below 10 m of elevation above sea level that are hydrologically connected to the sea), a figure which will potentially increase beyond 1.XX billion by 2050." Notably, this is not a formal assessment statement, which would be marked by use of formal confidence/likelihood language; rather, it provides context for this CCP. However, the statement does merit correction based on the authors' analysis, and the authors' analysis does indicate that a formal assessment of this topic would be warranted.

Refs. 19 and 45, meanwhile, are not IPCC reports at all.

So, in short, I think this is an important technical contribution that merits publication. I am concerned that the writing may be sensationalized for publication in Nature. The key question with respect to IPCC, for example, is not simply whether a paper cited by IPCC contains the errors that the authors identified, but the extent to which correcting the errors would influence the resulting impact assessment (or, even beyond that, the extent to which correcting these errors would influence policy or L&D negotiations). For the authors' strongest claims of import to hold, they need to show that more than one contextual, non-assessment sentence is at stake. Perhaps it is: knowing the context in which each of the references the authors are analyzing are used by AR6, and which if any assessment statements they support, would be key to interpreting this.

(Remarks on code availability)

Referee #2

(Remarks to the Author)

Re-review of: "Sea level much higher than assumed in most coastal hazards assessments" by Katharina Seeger and Philip Minderhoud.

I commend the authors on their responsiveness to review comments, many of which required additional analyses and/or substantive additions and references to the manuscript to validate their statements and conclusions. I also appreciate the more moderated tone of the piece as it now has the potential to serve as a more constructive contribution to the scientific community as to the importance of proper inclusion and handling of datum transformations.

While the overhaul is very thorough, and I was glad to see the detailed responses and changes made in response to my feedback (and that of others) such that I feel they are sufficiently addressed, the manuscript now requires some editing to streamline and tighten the piece. As I read, I found many examples of typos, passive voice, and generally awkward grammar and phrasing that I found distracting from the message they are trying to convey.

For example, the second sentence in the abstract (line 12) starts with "Their..." as written, it is unclear to the new reader what this was referring to in the previous sentence, sea-level rise and other hazards, or elevation and land cover (I'm aware it is the latter; a new reader might not be). Line 22 the sentences reads "potentially necessitating the accelerated implementation of coastal adaptation strategies" which is both awkward as written and a step too far; I think what the authors intend is that the findings show that more people could be in harm's way. And although I think the message is better in the sentences starting on line 86, this could be tightened considerably, thereby strengthening the message and impact. The sentences on lines 365-372 provide other examples of awkward/cumbersome phrasing that could be made more succinct. To address these concerns, I suggest that in addition to perhaps recruiting a reader with a keen editorial eye, that the authors evaluate sentences in the paragraph bodies carefully for clarity, as well as adding some paragraph breaks (there are many instances where a single paragraph spans nearly two full pages, for example lines 199-250) to ensure the messaging is as clear, direct, and succinct as possible to maximize their message (and the impact of the piece).

The authors should also pay particular attention to how they describe "complete documentation" vs. "correct conversion and adjustment" such as in figure 1; there is nuance that is captured in the text that gets at this (they describe further in the paper how with incomplete documentation they are able to infer where conversation might be correct/incorrect), but stating this more clearly at the outset would further strengthen their argument.

In addition, the caption for Figure 1, the Sankey diagram, needs to be revised to include explainers on what the different categorizations in the figure represent. Although the surrounding text goes into some detail, there are elements of the figure that need to be specifically and clearly defined, and the caption is an appropriate place to do this. For example, what does a sea-level reference with a correct implementation (correct implementation with respect to what) and reference up to date (current?) mean exactly?

Again, the authors are commended on the considerable work they have put into revision; it is an impressive effort and good to see how it has evolved.

(Remarks on code availability)

I reviewed the code the first time around; as it doesn't appear this was revised, I have not re-reviewed.

Referee #3

(Remarks to the Author)

The authors have provided thorough and detailed responses to the comments provided by the reviewers. Overall, the manuscript has improved a lot based on the extensive revisions. The methodological approach of the review and its limitations are now described in a much clearer manner. The language is more nuanced and less accusatory. The assumptions about omissions in the studies reviewed are now laid out more clearly. Most importantly, the additional text around the interpretation and sense-making of the findings is very helpful. Particularly, the additional analysis with a view towards IPCC AR6 is helpful. Also, the limits of the approach are now more clearly discussed. The deeper dive into the Mekong Delta case study supports to illustrate the relevance of the findings. The additional discussion of future steps the research community can take is also of value. The new charts in the appendix add clarity and illustration and hence strengthen the overall contribution.

I have a few minor comments the authors still might need to consider in order to improve the manuscript further:

- The choice of the Mekong Delta case study needs to be explained more thoroughly. Is being the third largest delta globally really the only – and most convincing – reason? Or can the discrepancy in sea level and impact effects be shown most

clearly here? If so, how does VMD compare roughly to other regions?

- In Figure 4, the sea level height lines (assumed vs. corrected) are not very clear and should be made more visible. In the same figure, the global results (bottom left) might be more clearly provided as numbers rather than fractions of the person icons. Overall, numbers in addition to the icons might be helpful.

- The terms omnipresent and omnipotent should be avoided. For the former, formulations such as “typical” or “common” would be better.

(Remarks on code availability)

Referee #4

(Remarks to the Author)

We thank the authors for providing a thorough, point-by-point response document to our previous comments. This gives more justifications and perspectives which is helpful and well explained and/or resolved with modifications. Below we outline specific responses to the key comments/modifications:

Regarding the title, it is clear that detailed thought went in to this and there appears to be strong rebuttal and justification for approach. However, we are not in our specific field of expertise and so defer to other reviews who are.

We are happy to see the argumentation in the paper is more “moderate, less criticising and nuanced” presentation of the findings by the investigated literature using the latest, at that time, available elevation and sea-level data

Happy that the assumptions made regarding the absence of any documentation or calculation steps in a manuscript is equivalent to the absence of datum conversion has been made more explicit

Happy more on the need for global-south research highlighted more in revised manuscript

Happy for more positive, forward-looking discussion regarding improving community research practices, as well as this being highlighted in the conclusion

Happy limitations of the review are highlighted more thoroughly in revised manuscript

Happy that a PRISMA diagram has now been added - However, the inclusion/exclusion criteria text in the figure is just a repeat, it would be better if figures for the reasons for exclusion could be added for transparency

Happy for clarification on use of review papers in this review, this is a sensible approach

Including the supplementary papers as a way of checking for one database/Scopus bias is unconventional but seems adequate, happy to see the approach detailed in the methods and the additional records provided as supplementary material and the limitation of this mentioned in the discussion. We believe that including another database at this stage is a lot of extra work that would probably not alter the results and so is not necessary as long as the limitations of this approach are highlighted, which the authors have now done.

Changes to the description of the inclusion/exclusion criteria have been made satisfactorily, as well as other methodological details/concerns

Overall, it is clear a lot of work has been done in this review, and the authors have addressed the comments thoroughly and provided clear justifications in the response document and manuscript. However, we must defer to subject experts regarding the boldness of the claim and the appropriateness of the title.

(Remarks on code availability)

Referee #5

(Remarks to the Author)

I co-reviewed this manuscript with one of the reviewers who provided the listed reports.

(Remarks on code availability)

The codes used to evaluate and synthesize the reviewed literature are included, detailed, and described in the supplementary materials.

Response to Reviews

+++

Referee expertise:

Referees #1-2: sea level rise, coastal processes

Referee #3-5: environmental assessments, systematic reviews

+++

Point-to-point response to referees

General response to referees:

We would like to sincerely thank all referees for their thoughtful, constructive, and detailed feedback. We deeply appreciate the time and effort invested in reviewing our manuscript and based on these valuable comments we were able to substantially strengthen our evaluation and improve the argumentation of our paper. We carefully considered all comments, performed a multitude of additional assessments and revised the manuscript and supporting information documentation. We are grateful for your support enabling us to develop this work further and feel that this thorough revision and the additional quantifications have considerably strengthened our paper in its argumentation and its overall robustness.

Below we provide a point-wise response to all point raised and suggestions made.

Referee #1 (Remarks to the Author)

In this important technical contribution, the authors perform a systematic review of literature assessing coastal population exposure, both globally and regionally, and identify that the large majority of this literature fails to adequately document the conversion between a geoid reference frame and a tidal reference frame, with the potential for consequent substantial impacts on estimated exposure.

While I have not replicated the authors' analysis, they have documented it well and provided their data and code consistent with open research practices, and it seems important to highlight this gap in the literature and to introduce procedures to avoid it in the future.

I need more convincing of the broader impact of this gap, however. My concern is that, by focusing on one source of error, they are not contextualizing that source of uncertainty relative to other sources of uncertainty in the underlying study.

We very much appreciate the referee's positive feedback and thank the referee for raising the point on potential uncertainties stemming from other sources than vertical datum. We agree on the need for contextualising of such uncertainties with respect to the main error we identified and focus on (i.e. vertical datum). During our revision, we undertook multiple actions to address this point and substantiate the relative magnitude of the identified (vertical reference) error with respect to other uncertainties related to the impact assessments. In addition to our existing quantifications, which were initially specifically designed to investigate and isolate vertical datum conversion errors only, we have now also included several additional evaluation to quantify the magnitude of the reference error in comparison to other sources of uncertainty (i.e. individual DEM vertical uncertainties, differences between DEMs, differences between population datasets) and compared our exposure quantifications to exemplar (IPCC-referenced) global and regional studies following your specific suggests made below.

To evaluate the impact of this gap, we need to partition the literature by application. For regional studies that constitute 90% of the literature review, it is not clear what the best metric to evaluate the import of the discrepancy is. It would be helpful to walk through some specific case studies that highlight how this error, relative to other sources of uncertainty, impacts the final results that the authors of the critiqued study think are important.

Thank you for this suggestion to detail the additional uncertainties with respect to the vertical datum issues at regional scale and we gladly provide such an assessment. We selected the populous Vietnamese Mekong Delta (third largest delta on Earth, 18 million inhabitants), building on existing literature and a recently available study (in review, preprint: <https://doi.org/10.21203/rs.3.rs-7706762/v1>). This study, focused on the delta's elevation, investigates the different sources of uncertainty on coastal elevation for a large number of available DEMs (including also widely-used older DEMs which we deliberately did not include in this study). Here we elaborate this study with a dedicated relative SLR impact assessment using population datasets and compare the findings with mimicked vertical reference issues as present in literature (see also Minderhoud et al., 2019). We provide an elaborate quantification and comparison of all related uncertainties (i.e. DEM vertical uncertainties, vertical reference, actuality of sea-level datum, differences between population datasets) and attributed the relative contribution of each respective uncertainty to the consequent impact assessment. The results for the Mekong delta underscore the large relative importance of the vertical datum and showcasing the large relative impact of a vertical datum error, which further increases with the newer, more accurate DEMs (suffering from less internal vertical uncertainty and other errors). The regional case study clearly underpins the main findings of

the paper. The metrics used are such that they are comparable to the new global comparisons that have been added as well (see following point).

The quantifications of this assessment are provided in Supplementary Tables 3 and 4 and insights from this assessment in relation to the DEMs in our current study are added to the manuscript:

“We also evaluated the encountered vertical reference issues on local impact assessments through assessments of major low-elevated coastal-deltaic areas around the world (Supplementary Figs. 11–14) and a meta-analysis for the Vietnamese Mekong Delta (VMD), the third largest delta in the world, inhabited by ~18 million people. When not correctly accounted for, vertical datum offset to MSL adds 10–60% additional (DEM-specific) error to the relative elevation assessment on top of existing data-inherent inaccuracy, sea-level and elevation change effects (Supplementary Table 3; Seeger and Minderhoud, 2025). With respect to vertical uncertainties from the other sources (e.g. DEM quality, remote sensing errors), the vertical datum offset present when omitting a conversion to MSL is responsible for the largest errors in impact assessments and increases the area and population in the VMD exposed to 1m RSLR from 1,400–6,000 km² to 18,400–24,800 km² (i.e. by 72–95%), and from 312,900–2.4 million to 5.4–10.0 million people (i.e. by 74–96%) (Supplementary Table 4). Correcting the encountered geoid error stemming from incorrect MDT implementation (Supplementary Fig. 18), which is particularly prominent in coastal lowlands of Southeast Asia, area and population falling below MSL following 1m RSLR increases by respectively 18% (from 20,900 km² to 25,500 km²) and 19–23% (from 5.5–7.9 million people to 6.8–10.2 million people), confirming the findings from earlier research on the VMD elevation¹².”

For the global studies, this metric is clearer, and the authors focus upon it: namely, number of people exposed at a global scale, under different 'scenarios' (1 m RSL rise or 10 m LECZ height.)

However, the authors do not highlight a well-cited erroneous study to make their point about the import of their results; rather, they do their own calculation, picking a comparison (assuming geoid = MTL) that maximizes the resulting discrepancy. It would be helpful to contextualize the discrepancy relative to actual range of numbers reported in the literature, based on errors as actually implemented by those studies. (E.g., Depsky et al., 2023, includes an incompletely documented conversion between geoid and tidal datums, thus one would not expect the error associated with this study to be as large as Lincke et al., 2022, which does not consider the datum offset correction to sea level.)

We appreciate these excellent suggestions on how to further evaluate the impacts of datum conversion issues on impact assessments. In our first submission, we were reserved in directly comparing our results to individual studies as the errors are so widespread and we wanted to avoid scapegoating individual papers or author groups, but we agree a more direct comparison on the impacts of specific errors is warranted to substantiate our paper.

In our revision, we added direct comparison (i.e. area and people below MSL) between our impact assessment and exemplar global and regional impact assessment studies. To have a more objective selection of which study to evaluate and simultaneously investigate the potential downstream implications of erroneous studies, we first explicitly investigated inclusion of the reviewed literature into IPCC reports (AR6). We have cross-searched the entire AR6 WG1–3 and the SROCC reports against our systematically-identified literature (385 papers) and found that 46 were included as references in the IPCC reporting (Search protocol in the Supplementary Information; Results in Supplementary Table 5 and Supplementary Figs. 20 and 21). We then grouped these studies according to their evaluated categories (correct implementation of sea-level reference (n=1); incorrect implementation of sea-level reference (n=9); absence of sea-level reference (n=36)) (Supplementary Table 5) and performed a comparison with several representative studies from each category and compared them to our results to show the potential magnitude of error in the existing, IPCC-referenced impact assessments as a result of vertical datum issues. As each study follows their own specific data and processing chain (sometimes not completely documented), for some comparisons we were unable to exactly replicate the analysis (we documented the differences for full disclosure) as they use slightly different datasets (e.g. population) which may introduce some discrepancy and the comparison provides an order of magnitude indication of potential error rather than exact values (for this reason, we designed our own controlled meta-analysis to rule out effects not related to vertical reference). The exact specific re-evaluation of each individual impact assessment is beyond our capability and should, and in many cases due to incomplete documentation can only, be performed by the original authors. The substantiations of the comparisons are provided in Supplementary Table 5, together with all the other IPCC-referenced studies (both global and regional) in the same assessment category. As such, the magnitude of error related to the specific vertical reference issues for the entire body of evaluated, IPCC-referenced studies is substantiated.

Toward this end, it might be helpful to have a version of Figure 1 that focuses on this easier-to-compare set of global studies, and how different outcomes on the right-hand side of this Sankey diagram are expected to correspond to differences in global exposure numbers. If possible to calculate, a histogram of errors for the global studies would be useful.

We see the value of substantiating the finding of this study and have tried various ways to integrate this into the Sankey diagram but we were unsuccessful in doing this. The problem we encountered is that each study has its own specific trajectory within the Sankey diagram, which comes on top of additional factors that influence the final impact quantification (e.g. different DEMs used, specific spatial extent to which the study was performed, specific local tidal reference levels, varying population data etc). Hence, studies grouped in similar trajectory in the Sankey diagram may each have study-specific and potentially widely ranging exposure statistics and discrepancies with our assessment and would require individual re-computation, which is clearly beyond our capacity.

We provide an achievable alternative through the newly added comparisons and discrepancy/error quantification of specific IPCC-referenced studies representing the different main categories (e.g. proper SL reference, incorrect SL reference, absence of SL reference). Supplementary Table 5 shows selected representative studies from which we directly compare to our results and add a column that links to similar studies in the same categories, which presumably have comparable differences with our corrected assessment. We have added the specific evaluation criteria corresponding to a specific Sankey diagram trajectory for each IPCC-referenced literature reference in Supplementary Data 1 to enable a comparison to the studies position in the Sankey diagram. We hope this provides an acceptable substitute to the referee.

Notably, IPCC AR6 (Cross-Chapter Paper 2) relies on Haasnoot et al., 2021 (doi:10.1016/j.crm.2021.100355) for their statement that about 11% of global population currently lives within 10 m of MSL, which compares with the 12.3-13.6% estimate in this paper. This would be a natural specific point of comparison, but this paper does not appear in the authors' analysis.

We thank the reviewer for the excellent suggestion to present a comparison of our results to Haasnoot et al. (2021). We have included the study in our additionally evaluated literature and included a direct comparison of the study to our results by including it as one of the reference assessments in our new the IPCC-referenced literature assessment (Supplementary Table 5). In addition, we have made a direct reference to the statistical comparison of the global population, that you so kindly already provided above, in the main manuscript.

Moreover, despite their focus on this one source of uncertainty, the authors take as given a single source of gridded population data, which is surely another substantial source of error. (How well do we really know where people on the planet live, at a 100 m scale?) At a minimum, the authors could compare results from WorldPop and the slightly lower resolution

LandScan (or possibly the new, comparable resolution LandScan HD), as well as discuss the literature comparing these and other population data sets. It would be important to know how the error introduced by imperfect tidal corrections (across the range of imperfections actually in the literature, as opposed to only in the extreme case where it is not done at all) compares to uncertainty in population distribution.

We thank the referee for highlighting this and we agree. Initially, we chose to not introduce other variables in our assessment and focus exclusively on the vertical datum error (illustrated using a population number), we realise that elaborating the population assessment provides a more complete understanding of the magnitude and range of the errors across vertical datum issues and literature. For this revision, we now include also two LandScan Global (2020 and 2023) datasets in addition to WorldPop 2020 (N.B. LandScan HD does not provide continuous global coverage). We have computed all global and regional statistics for each of the evaluated DEMs for the two additional datasets and include the results of the additional populations dataset (and their differences) throughout the reporting of population exposed in the data and manuscript. We also included a general section in the main paper on the different existing population datasets and uncertainties herein (Methods). The elaborated assessments now provide a more inclusive range of potential population residing in the effected low-elevated coastal zone and this is as such integrated in the quantifications and statistics throughout the paper.

The authors also omit one important calculation: namely, for each data set (DEM x population), the population estimated as being below sea level today. In economic modeling studies (e.g., Diaz, 2016; Depsky et al., 2023) it is clear that the combination of DEM, tidal corrections, and population used identifies a population who are below sea level under present conditions. In some cases, this is correct, and the identified population is below sea level but protected (e.g., in the Netherlands); in other cases, this reflects data errors. These economic models allow these people to adapt (move or protect) before the start of other calculations, adjusting for this bias in the underlying representation of the current state of the world. Of interest to me, therefore, is the *additional* population below sea level with 1 m RSL rise relative to 0 m RSL rise. (Similarly, with the LECZ, the additional population is of greater interest.) It would be helpful to know how much the errors the authors focus on affect this incremental quantity. If the errors on which the authors focus affect primarily the estimate of the currently exposed population as opposed to the additional population exposed, this has less effect on economic analysis (because the marginal increase in exposure with sea level rise may be unchanged) than if the errors primarily affect the additional exposure with 1 m RSL rise.

We thank the referee for pointing out this omission and we fully agree on the importance to differentiate between the estimated population (and area) already below sea level and additional population (and area) under a certain relative sea-level rise scenario and in the low-elevation coastal zone. In this revision, we computed these statistics, both in terms of area and population for all DEMs and using all three population datasets. The resulting quantifications are added to the Supplementary Data 5 and Information (Supplementary Fig. 10) and are included where relevant in the evaluation and reporting in the main manuscript and in Figure 4.

Referee #2 (Remarks to the Author)

Review of Seeger and Minderhoud, "Sea-level much higher than assumed in most coastal hazard assessments"

The manuscript first highlights discrepancies in sea-level rise assessments as they affect low-lying coastal landscapes. The findings are based on an extensive literature review wherein inaccuracies in sea-level rise related flooding/inundation are highlighted largely due to incorrect or absent datum referencing and transformations. The authors then conduct their own global analysis of errors/discrepancies using uncorrected geoid and ellipsoid results and comparing them against correctly transformed and corrected global datasets to demonstrate how these errors can manifest in underpredictions, particularly in southeast Asia and the southern hemisphere more generally where the geoid tends to be more inaccurate as compared with the northern latitudes. The authors raise concerns that the inclusion of these studies in larger scale assessments (e.g., the IPCC) may underrepresent risk to people and populations that may be affected, particularly in low lying areas such as deltas and coastal plains. The manuscript concludes by suggesting several ways awareness and correction can be used to address these issues, including more explicit guidance from journals to peer reviewers on aspects to be mindful of when evaluating these studies, as well as broader awareness of standards and errors within the scientific community.

In sum, the authors highlight what appears to be a pervasive and problematic issue that needs to be brought to the attention of the scientific community, and a high impact journal like Nature is a good place to do that.

We thank the referee for this elaborate assessment and valuation of our paper and we value the constructive feedback and concrete suggestions to substantially strengthen our paper and its value to the scientific community.

Although the manuscript appears scientifically robust and I think will be of interest to many in the field, I have major concerns with regards to caveats and limitations in the work need to be more transparently discussed. As a consequence, much of the paper currently reads as sensationalized, with the same findings mentioned multiple times, and without adequate context given to them. Moreover, the paper could also be reframed around some of their conclusions—the factors that have led to the abundance of seemingly “flawed” studies—that would give it a much more engaging and constructive impact within the scientific community. Specifically, I do not recommend publication until and unless these issues are comprehensively addressed:

- 1) Literature screening issue: In the literature review, which uses PRISMA as a screening tool, it appears there may be an error with one of the exclusion/inclusion literature screening elements included in the supplement (p 18-19 in supplement). Both internal and external screening lists include ‘focus on coastal land and coastal sea level’ as the first criterion listed; if this is not an error, the authors need to explain to the reader why this criterion is included in both. I highlight this because I had questions as to why some well-known studies that seemed they would be relevant and made the first order screening were ultimately excluded from the review, and I could not resolve this based on the information in these categories, making me question the reproducibility of the approach.

We thank the referee for pointing out this error in Supplementary Table 1. The referee is right, the inclusion of the screening criteria in both lists indeed is an (copy-paste for layout related) error and we apologise that we did not notice this during proof-reading prior to submission. We corrected the error and removed ‘focus on coastal land and coastal sea level’ from the exclusion criteria.

In addition, based on the referee’s comment that some well-known studies that seemed relevant got excluded in the second screening, we re-analysed both screenings but came to the same results as the submitted version. In conjunction, we have also further detailed the inclusion and exclusion criteria, to enable the reproducibility of the screening process. Although the referee does not provide information on the mentioned well-known studies, we deem it likely that the referee points to papers that may be initially included in the first screening round because of their relevant context, however, that do not combine data from coastal sea level and land elevation. In addition, to aid the assessment of the screening process by the referee, below we present the 10 most cited papers which made the first screening but did not make the second and detail per study why they were excluded during consequent screening:

Publication title	Cited by	Literature screening I	Literature screening II	Reason for exclusion
Increasing risk of compound flooding from storm surge and rainfall for major US cities	607	1	3	Absence of elevation data use
The dynamic effects of sea level rise on low-gradient coastal landscapes: A review	278	2	3	Review article without primary data calculation
Evaluation of the combined risk of sea level rise, land subsidence, and storm surges on the coastal areas of Shanghai, China	236	2	3	Exclusive use of local elevation data
Assessment of groundwater inundation as a consequence of sea-level rise	209	1	3	Use of airborne LiDAR without combination with satellite-borne elevation data
Application of analytical hierarchy process (AHP) for flood risk assessment: a case study in Malda district of West Bengal, India	205	1	3	Focus on hazards other than coastal hazards
Migration induced by sea-level rise could reshape the US population landscape	186	2	3	Use of airborne LiDAR without combination with satellite-borne elevation data
Dynamics of sea level rise and coastal flooding on a changing landscape	171	2	3	Use of airborne LiDAR without combination with satellite-borne elevation data
Continental scale mapping of tidal flats across east Asia using the landsat archive	164	2	3	Focus on (Palaeo)Geomorphology
Increased nuisance flooding along the coasts of the United States due to sea level rise: Past and future	161	2	3	Absence of elevation data use
Comparing the cost effectiveness of nature-based and coastal adaptation: A case study from the Gulf Coast of the United States	160	2	3	Use of airborne LiDAR without combination with satellite-borne elevation data

2) Population findings: Better justification/explanation needs to be provided for the population dataset that is used in the global reanalysis. Because conclusions rely so heavily on this dataset, the reader needs to be clear on a) why this was selected, above

any other, to evaluate population impacts; and b) the limitations in basing conclusions on one global dataset (specific to 2020) and/or any caveats associated with associated results/outcomes (e.g., noting that a 2020 population estimate is being used to compare against a future condition that is unlikely to be reached for many decades).

Thank you for pointing this out, also in line with comments by reviewer #1. We fully agree and in this revised submission, we have now included two LandScan Global 2020 and 2023 datasets in addition to WorldPop 2020 (N.B. LandScan HD does not provide continuous global coverage) and computed all global and regional statistics for all DEMs. We inserted a general section in the paper (methods) that discusses the differences and uncertainties between the different population datasets, discuss the limitation of using contemporary population estimates to project future conditions and report the broader range of population exposed (i.e. using the range from all three datasets) in the result reporting throughout the paper.

3) Novelty: By conducting their own global meta-analysis, the authors generate helpful conclusions to illustrate how underprediction may affect many coastal locations. While this was undoubtedly a considerable undertaking, the approach used is not novel (the code evaluated showed two instances of simple raster subtractions for a GIS environment) and benefitted from the use of many global datasets that have recently been produced. The novelty of the work comes not in the approach but in the findings from the literature review, that they then use the meta-analysis to help to support; again however, transparency regarding meta analysis limitations is needed, as well as better distinctions made between their findings and those from their literature screening (see comments on abstract below).

We agree with the reviewer that novelty in our paper not solely lies in the proper corrections of vertical datum of global elevation models, but in the combination of the literature review and quantification of revealed issues by means of meta-analysis. Throughout literature, there are only few examples where vertical datum correction using mean dynamic topography (MDT) data was performed without error and documented correctly (most previous studies that conducted datum conversion to MDT overlooked to correct for geoid offset before the alignment of coastal sea level and coastal elevation). We have reported a proper conversion strategy earlier in Seeger et al. (2023, 2024), which indeed in the end is a relatively straightforward subtraction of different surfaces. Note that for this paper we slightly altered the specific approach by first converting both elevation and sea-level data to the recent GOCO06s geoid model instead of using the EGM96 as common datum (described in Methods).

In this revision we have extended the meta-analysis (e.g. adding other population datasets, assessing the newest (Aug 2025) released geoid model) to reduce earlier limitations and we have extended our reporting on the meta-analysis procedure and limitations throughout the manuscript: we extended our discussion on other sources of elevation/SL uncertainty and inaccuracy and how they are independent from the investigated vertical reference issues; 2) in addition to our own meta-analyses, we added an extensive analysis and direct comparison of our meta-analysis and existing impact assessments in IPCC-referenced studies, including the point on which the two compared studies deviate (see also comments on the IPCC comparison referee #1); 3) we make the distinction clear in the manuscript between values stemming from our meta-analysis findings and other literature findings. In addition, to investigate and quantify the different sources of uncertainty in coastal impact assessments, we have added a regional assessment of the Mekong Delta (Supplementary Table 3), building on a recent paper in review (<https://doi.org/10.21203/rs.3.rs-7706762/v1>) and extending this work with a comparable exposure impact assessment as we performed at global scale (Supplementary Table 4). This case enables us to detail the impacts across a large range of uncertainties (e.g. individual DEM vertical uncertainties, differences between DEMs, differences between population datasets) and vertical datum issues and provide a relative assessment of each contributor to the final impact assessment statistics, showcasing the large influence of vertical datum in respect to other uncertainty-creating factors.

In addition, based on point 5 below on a presumed 'bathtub approach', which would indeed come with a number of additional assumptions and limitations: we would like to point out here that our meta-analysis is not aiming to be an inundation assessment, as such we strictly report 'population below sea level' and do not present it as inundation or hydrodynamic assessment. We included an explicit sentence to point this out in the Methods.

References:

K. Seeger, P. S. J. Minderhoud, A. Peffeköver, A. Vogel, H. Brückner, F. Kraas, Nay Win Oo, D. Brill, Assessing land elevation in the Ayeyarwady Delta (Myanmar) and its relevance for studying sea level rise and delta flooding. *Hydrol. Earth Syst. Sci.* **27**, 2257–2281 (2023). <https://doi.org/10.5194/hess-27-2257-2023>.

K. Seeger, A. Peffeköver, P. S. J. Minderhoud, A. Vogel, H. Brückner, F. Kraas, Nay Win Oo, D. Brill, Evaluating flood hazards in data-sparse coastal lowlands: highlighting the Ayeyarwady Delta (Myanmar). *Environ. Res. Lett.* **19**(8), 084007. <https://doi.org/10.1088/1748-9326/ad5b07>.

K. Seeger, P. S. J. Minderhoud, Attributing uncertainties in elevation assessments for data-sparse coastal lowlands using global elevation models: A globally applicable approach

showcasing the Vietnamese Mekong Delta. 30 October 2025, PREPRINT (Version 1) available at *Research Square*. <https://doi.org/10.21203/rs.3.rs-7706762/v1>.

4) Argument framing: The paper would be a stronger and more productive contribution to the scientific discourse (and the studies they critique) if the authors offered more context on a) poor data quality and/or paucity and the advances new global datasets, like those used in their reanalysis provide (and that were arguably unavailable during the time many of the studies published in the last five years they critique were conducted); b) the tradeoffs their results might have with more localized studies that can be produced at higher resolutions and/or include nuance that a global approach cannot; c) their decision to use a 1m SLR increment in their analysis, and how that compares to what the other studies they reference have used in their sea-level rise impact assessments. I especially liked some of the recommendations made for improvement for future work in the conclusions (e.g., constellation issues adding to complexity of transformations; peer-review guidance, including explicit instructions for datum review); stating these earlier in the paper could be used to more productively frame the contributions this manuscript offers.

We thank the reviewer for these concrete suggestions to strengthen the argumentation of the paper and gladly did so. Overall, we have considerably rephrased the manuscript to a more constructive framing, now also highlighting early in the manuscript the outlook towards the recommendations for improvement.

a) We considered the unavailability of current available and more accurate datasets by evaluating whether the investigated literature used the latest available elevation and sea-level data during the respective study performance (threshold: newer data must have been available at least three months before initial submission of the paper). We highlighted this point in the Supplementary Information (e.g., Supplementary Table 2). We note that the investigated main issue in our paper, namely vertical datum conversion omission, is unrelated to the availability of newer datasets, as the earliest global mean dynamic topography (MDT) dataset was already available since 2004, well before our 15-year time window of analysis. As such in our window of investigation, the vertical datum conversion to local sea level is not hampered and independent of the elevation and sea-level dataset used, and create a separate additional error when omitted irrespective of whether an old (less accurate) or newer (more accurate) elevation or sea-level height product was used. We emphasised this point in the manuscript and elaborated on the sources of uncertainties involved into the usage of elevation and sea-level information in coastal impact assessments.

- b) We note that a large part of the evaluated studies in the literature review was regional, and vertical reference issues are unrelated to the spatial scale of the impact assessment. However, it is certainly the case that regional or local studies (e.g. national studies using higher accuracy elevation data (i.e. LiDAR) and local sea level referencing) perform better than studies relying on satellite-derived information. We address this point in the introduction and reiterate it again in the discussion, specifically mentioning national-based (using data which is often not available in the public domain, see also discussion of Minderhoud et al., 2019) and the implications (positive) these may have (when existing and performed properly) for existing national exposure assessments and adaptation strategies with respect to the evaluated literature. In addition, we also provide now an example with the Mekong Delta case study in which we compare global satellite-derived elevation models to local elevation-data and tide-gauge controlled sea level, thereby showing that the newest generation of global, satellite-derived DEMs is approaching the vertical accuracy present in local, high-accuracy datasets, underscoring the relative even larger influence of datum conversion issues with newest, more accurate elevation products. In addition, we highlight the large influence of vertical datum conversion issues by assessments of further, major low-elevated coastal-deltaic areas around the world in Supplementary Figs. 11–14.
- c) The 1m SLR increment follows the example other available impact assessments and enables the direct comparison to several studies which are included in the IPCC reports. We describe the choice for our approach now explicitly in the methods and the newly added IPCC-reference literature table, in which we compare our results directly to other studies that follow the same approach, also explicitly shows the direct comparison (including mentioning of other study discrepancies, if existing).

Thank you for the feedback on our recommendations. Overall in our revision, we structured and framed the paper more towards these recommendations.

- 5) Limitations of inundation approach: The authors do not address the limitations of bathtub or inundation approaches in general (of which theirs is one), which have been well documented in the literature as underrepresenting dynamic coastal change present in many of the low-lying areas in which they focus (see: Passeri et al., 2015). At the very least, their results should be caveated with this information.

We are fully aware of this point and agree with the reviewer that inundation involves more complexity than our assessment is able to address. For this reason, we decided again following any inundation approach at all in our meta-analysis and stick to a relative elevation

assessment only. We exclusively focus on coastal sea-surface height in combination with area/population below sea level and do not perform any bathtub, inundation or other hydrodynamic approach. In the reporting of our meta-analysis ensured to never use terms such as “flooding” or “inundation” (etc.) but explicitly referred to area falling below mean sea level following 1 m of relative sea-level rise and within the 10 m low-elevation coastal zone. We added a distinct statement in the Methods.

6) Evaluation of use of flawed studies: Finally, it seems a missed opportunity that the authors did not conduct a comparison of how many of the flawed studies that they highlight are cited in larger scale assessments (for example, the most recent IPCC) to provide substantiation for their claims that these studies often form the basis for larger assessment findings. Even the evaluation of just one assessment as an exemplar would be of benefit.

We thank the referee for this excellent suggestion, also made by referee #1, and for this revision, we investigated the inclusion of all the reviewed literature into IPCC reports. We have cross-searched the entire AR6 WG1–3 and the SROCC reports against our systematically-identified literature (385 papers) and found that 46 were included as references in the IPCC reporting (Search protocol in the Supplementary Information; Results in Supplementary Table 5 and Supplementary Figs. 20 and 21). We then grouped these studies according to their evaluated categories (correct implementation of sea-level reference (n=1); incorrect implementation of sea-level reference (n=9); absence of sea-level reference (n=36)) (Supplementary Table 5) and performed a comparison with several representative studies from each category to our results to show the potential magnitude of error in the existing, IPCC-referenced impact assessments as a results of vertical datum issues. In addition, following also the suggestion of referee #1, we also included the IPCC-referenced paper of Haasnoot et al. (2021) to our additionally evaluated literature and included a direct comparison with our analysis.

Line Items:

In the abstract, as well as in many places in the text, sweeping generalizations about the flawed studies they cite are made, which I believe are instead based on the findings from their global meta-analysis. For example, Lines 18-20 discuss how evaluated studies improperly handled data and transformations, but lines 20-23 seem to refer to findings based on the meta-analysis the authors themselves conducted, and this distinction should be made more clearly. Similarly, in line 25, the authors state: “67% more people (increase of 88 million to 132 million people) will fall below sea level following 1 m of relative sea-level rise,” but it is

unclear this increase is in comparison to what; again I'm inferring this is the comparison of their global reanalysis findings, but if so, this should be stated for context in the abstract.

We have rephrased many parts of the manuscript, making some of the statements more moderate and added context where needed to clarify whether a comparison comes from a comparison to literature or stems from a certain conversion we evaluated in our meta-analysis.

Line 114: "Not accessible or demonstrably.."

Changed as suggested.

Line 135: correct to "demonstrably"

Changed as suggested.

Line 190: past "relative" sea-level changes?

Changed as suggested.

Line 194: "of actual coastal exposure necessitate the accounting of relative sea-level changes..."

Changed as suggested.

Lines 274: These statements have been made at least two other times in the article and the repetition is getting tedious.

Changed as suggested, repetition throughout the manuscript is reduced.

Line 290: Can you say something more about how these data gaps might get filled?

The reference we cite includes more information on the uncertainties and provides some brief information how these can be tackled. We added another reference to the paper currently in review in which we detail and showcase how the different uncertainties (and data gaps) can be addressed. Here in the manuscript, we point to these uncertainties that include data-inherent inaccuracy, sea-level and elevation change effects as showcased for the Vietnamese Mekong Delta and provide their quantification and attribution in Supplementary Table 3.

Line 295: Replace "suffer from" with "be affected by" as more neutral terminology.

Changed as suggested.

Lines 333-334: See concerns above regarding the secondary screening.

We carefully re-analysed both screenings and came to the same results as the submitted version. We thank the reviewer for spotting the copy-paste (for layout) related error in the list of inclusion and exclusion criteria which may have caused the concerns. We apologise that we did not notice this during proof-reading prior to submission and corrected the error (see response above). Furthermore, to enable the reproducibility of the literature screening, we detailed the inclusion and exclusion criteria and provide a documentation in Supplementary Fig. 1 and Supplementary Table 1 and 2.

Line 375: What are “the very coastal stretches”?

We understand that this phrasing might be misunderstood and changed it to “the coastline”.

Line 380: Replace “allows to investigate” with “can be used to adjust”.

Changed as suggested.

Lines 401-405: This is verbatim the same sentence as the last in the paragraph above; is this intentional?

This paragraph focusses on a different dataset than the previous paragraph where we refer to the resampling of the geoid data. In lines 401–405, we refer to the mean dynamic topography data instead.

Line 416: Awkward phrasing: “a distance threshold for which we deem this approach”

Changed to “a distance threshold for which we consider this approach”.

Line 418: Replace “We considered to transpose” with “We transposed”.

Changed as suggested.

Line 431: Elevation threshold of -7m above mean sea level?

Rephrased to 7m below mean sea level

Line 448: Again, see comments above regarding justification and caveats around using this dataset and conclusions drawn from it.

We included a section in the Methods on the integration of additional population data (i.e. LandScan 2020 and 2023) and the potential uncertainties that may arise from population datasets, along which we provide references to respective studies.

References:

Passeri, D. L., Hagen, S. C., Medeiros, S. C., Bilskie, M. V., Alizad, K., & Wang, D. (2015).

The dynamic effects of sea level rise on low-gradient coastal landscapes: A review. *Earth's Future*, 3(6), 159-181.

Referee #2 (Remarks on code availability)

The code evaluated showed two instances of simple raster subtractions for a GIS environment; it is a usable resource in current form though very straightforward and something most that use GIS could create very easily.

We agree the code and transformations are straightforward. We provide the codes for full transparency and reproducibility, given our observation that these two steps have not been considered in the majority of the studies assessed in the literature review. The codes provide clear transparency on the data processing conducted (even though the calculations are simple subtractions and straightforward) and maximise the facilitation of the reader to reproduce a similar conversion on their own, either in a GIS environment or outside a GUI.

Referee #3 (Remarks to the Author)

I am pleased about the opportunity to review this interesting manuscript. Building on a large structured literature review and own modeling work reproducing errors found in existing studies, the paper finds that the majority of the current literature on sea level assessments and sea level rise show severe shortcomings in their methodological reporting and data use. The paper therefore argues to find that sea levels are higher than assumed in most coastal hazard and impact assessments. Interestingly, the paper finds that the discrepancies are widest in many regions of the Global South which suffer from particularly high sea level rise exposure but also from data scarcity. Looking into the future, the paper therefore argues based on its results that current sea level rise impact assessments underrepresent the actual exposure and risk from sea level rise, meaning that more locations and more people will be at risk than previously assumed.

As such, the paper provides a daring approach and innovative findings. However, I feel that the paper suffers from a few substantial shortcomings in terms of its coherency, the way it presents its data and the conclusions that are drawn from it. In my view, these points would need to be revised before the manuscript can be considered for publication.

In lines 38 to 40, the authors argue that “less than 1% of the evaluated literature provided thorough documentation and handled coastal sea-level and elevation data properly”. I find this statement problematic, inconsistent and misleading - at the very least imprecise. How

can the authors know that studies allegedly did not handle data properly, if thorough documentation was lacking? Two sentences further, a similar formulation is likewise very problematic. The later parts of the paper provide more precise numbers on the reporting shortcomings in the evaluated papers. Nevertheless, the main concern I raise here remains. Figure 1, for example, shows the large proportion of studies with documentation shortcomings. But how can the authors deduce from there, that all these studies have errors and only one out of the 385 evaluated studies contained no conversion and alignment errors (lines 122-123)? I cannot follow this conclusion and find the negative interpretation very problematic - maybe/potentially invalid. In fact, the authors themselves argue in the conclusion that in many cases, the missing documentation makes it impossible to evaluate the correctness of the analysis undertaken in the respective paper. However, this should be more carefully separated from the sweeping allegation of “widespread misuse” which is raised two sentences earlier.

We thank the reviewer for critical feedback and this made us carefully reconsider how we are presenting our results and explain the underlying assumptions and considerations more extensively. In addition, the reviewer will see that the argumentation of the manuscript has been redrafted considerably to a more neutral presentation of the findings. Moreover, we abolished the use of negative wordings such as ‘misuse’ as we realised that these can be interpreted more negatively than we intended (originally, we used the word ‘misuse’ as short for ‘incorrect use’, but we understand it may imply more).

We were also surprised by the outcomes of our evaluation that less than 1% of the evaluated literature provided thorough documentation and handled coastal sea-level and elevation data properly. The largest group of literature we encountered did not show any data or evidence of considering any type of datum conversion or sea-level data. In the original manuscript, we document the large group of literature that (largest group in literature and higher degree of error), as Vertical datum documentation: *Absent*, Vertical Datum Conversion: *Correctness not assessable*, Vertical datum type: *Not assessable*, Actuality of SL ref: *Not assessable*. We are highly confident that the papers categorised in this group did not consider nor performed a datum conversion (see below), but as we are unable to proof this for each individual paper without a complete re-analysis of each work (see below), we decided to label them in an objective way. Thanks to the referee’s feedback we realise the original labels fall short in communicating our findings properly and we have modified them.

To elaborate on this, it is important to note that a datum conversion is not common practice in the community nor exists of a simple standard processing step. A vertical datum conversion always involves additional specialised data (less widespread) in addition to coastal elevation

and sea level (e.g. geoid, offset between geoid(s) and tidal level) as well as expert knowledge on conversion and a procedure to process this data, including a certain software environment. We deem it very unlikely that authors who are aware of the importance of vertical datums and vertical datum conversion and go through the entire procedure of conversion end up not writing anything on vertical datum type, additional data they used to make the conversion or the processing steps they followed in their papers and/or supplementary information/data. Therefore, we assume that the absence of any documentation of those datasets or calculation steps in a manuscript is equivalent to the absence of datum conversion. And for each instance we checked the results of such a paper, the results presented (maps, statistics, etc.) provided us with confirmation that this assumption is correct. Note that creating hard proof on this confirmation would require to reproduce each specific assessment entirely, and likely even multiple times as the absence of data documentation leaves a multitude of potential approaches. Naturally this is outside of our capacity and the responsibility of the original authors. However, in the revised version we do compare a number of studies from different error categories directly to our results (Supplementary Table 5), and every time we made a comparison to a paper in the group with datum conversion omitted, our assumption was confirmed.

In the updated manuscript, we now explicitly explain this assumption:

Main text: “We expect that authors aware of the necessity and with the expertise to successfully perform a vertical datum conversion using additional sea-level data do properly document these crucial (and often time-consuming) methodological steps and additional datasets. Therefore, we presume that the absence of any documentation of sea-level information (25%) and/or methodological conversion steps (65%) means inclusion of sea-level height data or datum conversion to a sea-level reference was not considered and omitted. Repeated evaluation across multiple studies confirmed the validity of this presumption (e.g.¹², Supplementary Table 5; Supplementary Tables 3 and 4).”

Methods: “As vertical datum conversion always requires additional data next to a DEM-based coastal elevation dataset (e.g. vertical datum information, offset between geoid(s) and tidal level, sea-level height data) as well as a conversion procedure (including a specific GIS or coding environment or conversion service), we expect researchers who perform a datum conversion to document the additional datasets and procedures in their papers and/or supplementary information/data. In case a paper does not provide any documentation of additional datasets or required conversion steps, we presume that no datum conversion was performed. This assumption was confirmed to be correct for numerous papers where the absence of datum conversion becomes apparent in the results of the study.”

In addition, we have revised and extended the labelling of the different categories in the revised version, making the categorisation more elaborate and explicit. Throughout the entire revised manuscript, we ensured that the writing correctly reflects the nuances between hard findings versus the assumed findings and thereby hope to rule out any inconsistencies or misunderstandings which stemmed from the initial manuscript.

Rather than repeating the numbers of the Sankey charts provided in Fig. 1, the text should concentrate on explaining the problematic convolution of missing documentation and the assumption on erroneous data treatment.

Thank you. We have modified the section surrounding the Sankey diagram and added additional explanation and substantiation of Fig. 1. The implications of the findings presented in the Sankey diagram are now also more extensively described and explained in the next section.

For non-technical readers, a conceptual figure or infographic explaining how the discrepancies between sea level and most-used geoid height arise might be helpful.

Thank you for this suggestion. To accommodate this, we have provided such a conceptual infographic in Supplementary Information, showing the discrepancies between sea level and geoid heights (Supplementary Fig. 5a). Furthermore, we now make references to Seeger et al. (2023) and Seeger and Minderhoud (2025) that include further conceptual figures, respectively.

References:

K. Seeger, P. S. J. Minderhoud, A. Peffeköver, A. Vogel, H. Brückner, F. Kraas, Nay Win Oo, D. Brill, Assessing land elevation in the Ayeyarwady Delta (Myanmar) and its relevance for studying sea level rise and delta flooding. *Hydrol. Earth Syst. Sci.* **27**, 2257–2281 (2023). <https://doi.org/10.5194/hess-27-2257-2023>.

K. Seeger, P. S. J. Minderhoud, Attributing uncertainties in elevation assessments for data-sparse coastal lowlands using global elevation models: A globally applicable approach showcasing the Vietnamese Mekong Delta. 30 October 2025, PREPRINT (Version 1) available at *Research Square*. <https://doi.org/10.21203/rs.3.rs-7706762/v1>.

Also, it would be helpful to explain more clearly on how two findings go together: Figure 2, for example, suggests that there are positive and negative discrepancies between coastal sea level and widely used geoids, particularly EGM96. This applies also to regions identified as most problematic such as Southeast Asia. However, in the later course of the paper, only the findings regarding the alleged underestimation of current sea levels are highlighted, arguing

coastal sea level worldwide is on average 0.2-0.5 m higher than assumed in coastal hazard assessments with errors.

Thank you for noting this, we agree that a more elaborate discussion on the local scale effects is warranted, beyond the worldwide or regional average values for which sea level is on average higher. Beside the underrepresentation of sea level at larger scales, we now also highlight the local opposite phenomenon, and its implication: *“While the geoids on average underrepresent sea-surface height at global and regional scale, locally the discrepancies can also range in opposite direction (e.g. northern Mediterranean coast, Antarctica and some islands in the Atlantic and the Pacific (EGM2008 only) (Fig. 2), consequently resulting in an overrepresentation of sea-surface height.”*

In addition, the role of data scarcity would deserve a more considerate discussion. On the one hand the authors argue that there is a widespread misuse of data, generating most severe discrepancies in areas of the Global South which also suffer from high data scarcity. However, the question whether and to which extent it is justified to talk about the “misuse” of data in situations in which data is insufficient or not available is not discussed. But this discussion is of great relevance for the overall direction of the paper and its messaging: While the paper currently accuses the overarching part of the research community for the alleged misuse of data, it might be the data scarcity that is to blame. Taking this perspective would of course shift the conclusions and recommendations of the paper quite a bit.

It is true that data scarcity is an important factor in the development and quality of individual products, such as geoids, which in the revised manuscript we now also explicitly discuss in relation to geoid versions and accuracy. In addition, we would like to point out that the investigated main issue in our paper, namely vertical datum conversion omission, is unrelated to data scarcity, data inaccuracy or availability of newer data, as both global DEMs and global mean dynamic topography (MDT) datasets have been openly available since 2004, well before the 15-year time window we selected for our analysis. As such, data scarcity is not to blame and did not hamper anyone’s ability of performing a vertical datum conversion to local sea level. In addition, vertical datum conversion is independent of the elevation, geoid and sea-level dataset used (old, new, less accurate due to data paucity in certain regions) and conversion itself creates a separate additional error when omitted irrespective of other courses of uncertainty. We highlight the large influence of vertical datum conversion issues by assessments of major low-elevated coastal-deltaic areas around the world in Supplementary Figs. 11–14. To substantiate this in more detail, we have further emphasised this point in the manuscript and elaborated on the other sources of uncertainties

involved when elevation and sea-level information in coastal impact assessments, further illustrated by the newly added local case study of the Mekong Delta.

The discussion on the potential reasons behind the alleged misuse should be strengthened. The authors put forward the interesting hypothesis that Global North authors might be overconfident in the general validity of the geoid models used, based on the fact that in the Global North those models perform well. Here, the authors should further elaborate on whether they have tested this hypothesis as part of their work.

We have extended the discussion on the potential reason behind the issues we discovered (abolishing ‘misuse’ or placing specific blame). The hypothesis we present on Global North authors stems from our own observations and some anecdotal evidence when speaking to geodesists wondering why they should use “the less accurate MDT data” instead of the better performing geoid models. We did not further investigate nor tested the hypothesis, which is also beyond the scope of this study, and state this now explicitly in the manuscript. However, we provide an example of incompatible North–South transferability by showing the CoastalDEM v1.1 (Kulp and Strauss, 2018), of which the neural-network approach was used to create the CoastalDEM was Global North-trained (USA) and -validated (Australia), but performed considerably worse elsewhere. The approach, for example, places half of the entire Mekong Delta already well below present-day sea level (Supplementary Fig. 19), thereby greatly overestimating consequent population exposure to high-water levels in the region.

Referee #4 (Remarks to the Author)

General

This study offers timely and valuable insights for researchers in the field, as well as the broader scientific and policy-making communities. Through a systematic/meta-analysis of peer-reviewed publications using the PRISMA framework, it looks to have identified key methodological and empirical gaps in sea-level and hazard assessment research. These gaps highlight the potential for serious misinterpretations and poorly informed decision-making, emphasizing the need for closer scrutiny and more rigorous approaches in future studies. The manuscript is coherent, well-written, and supported by comprehensive supplementary materials, including data extraction tables, raw data, literature search protocols, codes, and additional figures and tables. Its rigorous approach and broad relevance make it a significant contribution, suited for publication in Nature. However, a few

minor issues remain that warrant clarification and further justification to strengthen the paper's quality.

We thank the reviewer for these positive comments on our paper. We address the identified minor issues below.

Specific comments and issues for clarification

Title:

The title is informative and effectively conveys the main message of the paper. However, it would benefit from explicitly highlighting that sea level has been underestimated, as this is a key finding clearly demonstrated in the study. Using the term “much higher” may be misleading, since no clear threshold is provided to determine whether the values fall within that range, and no evidence on statistical significance was provided.

Thank you for acknowledging the effectiveness of the current title to reflect our main message. The title has been the result of extensive deliberations in which we considered a great number of titles and key words, including versions that included the suggested ‘underestimated’. We decided against using ‘estimation’ in conjunction with sea level, as this would imply that we have either better data or found a way to decrease uncertainty and are thereby able to provide a better estimate of sea-level height with respect to earlier work, while this is not the case. The main issue we are dealing with relates to (geodetic) vertical reference frames, which in principle do not come with an uncertainty or error as they are used as fixed frames, similar as a coordinate system, or the x- and y-axis of a graph, do not come with an error but present an absolute reference. As such, a geodetic reference frame doesn't come with statistics. Our finding that sea level height turns out to be much higher is the result of a structural bias or offset in this fixed framework, rather than higher quality data or an improved estimate. The original accuracies and uncertainties remain the same, but the zero point shifts).

It is completely true that the vertical datum offset issue leads to downstream underestimations in sea-level-related impacts and coastal exposure assessments, and if the title would revolve around the consequent increased population exposed by the identified offset, we would indeed prefer to use ‘underestimated’, since their values do have uncertainties, ranges and statistical underpinnings. But as we focus on sea-level height offset stemming from not converting the vertical reference frame (no estimate involved), we think the current title is most proper and a clear summary of the main finding in the paper.

We understand the reviewer's comment concerning the use of the wording 'much' without providing a clear statistical threshold. Here we provide the reasoning which made us select this word: The MDT data which we used to correct to local mean sea level datum exhibits near-coastal vertical uncertainty/error of ~4 cm. The offset we identified (global average = 0.3 m) is up to seven times as large as the MDT measurement error, surpassing all matrices of statistical significance. Also, the offset equals roughly half a century sea-level rise, thereby potentially expediting previous end-of-century projections to be reality in a few decades. As such we argue for keeping "much" in the title as we feel that a sudden upward shift of assumed sea level by several decimetres, with respect to known rates of mm/yr, will be widely considered to as much.

In case the reviewer and/or editor do not agree with the above argumentations and justification to preserve the current title and insist on changing it, we are willing to adjust it, although we feel it would mean sacrificing part of its descriptive value for the main findings in the manuscript.

Abstract:

The abstract is also informative, but it could be improved by including a brief overview of the approach followed to synthesize evidence. The novelty of this synthesis lies in its data handling and critical evaluation of the methodologies commonly used in previous studies, an aspect that should be more clearly emphasized. For example, on line 17, instead of stating here we.... It would be better if the methodology employed was stated e.g. Using a systematic meta-analysis of peer-reviewed publications we reveal.... Also on line 70 e.g. through a systematic review of 385 papers contemporary methods for SLR impacts and coastal assessments were evaluated, finding that these crucial steps were often not considered

Thank you for providing these concrete suggestions to improve the abstract and introduction by emphasising the methodology employed. We have included the suggested descriptions on methodology in both the abstract and introduction.

Reflection on the key message - strong criticism on the reviewed papers!

The claim that all (except one) assessed peer-reviewed publications on sea-level rise (SLR) and hazard assessments are erroneous and invalid seems overly bold. These studies were conducted under diverse natural and human-influenced contexts. The systematic review and meta-analysis attempts to compare them across varying spatial and temporal scales which introduces inherent complexity and necessitates careful contextual consideration, although

the concern on proper application of standardization approaches is valid. Details of the contexts that were considered for each paper could be better described.

Thank you for pointing out (it also resonates with similar comments from other referees), we realise our initial presentation of the results may have been too blunt. Based on these feedbacks, as you will see, we have reshaped the argumentation in our paper to be more moderate, less criticising and nuanced presentation of the findings.

We are aware of the temporal component and in our internal evaluation process we are accounting for this by evaluating whether the investigated literature used the latest, *at that time*, available elevation and sea-level data during the respective study performance (threshold: newer data must have been available at least three months before initial submission of the paper). We highlighted this point in the Supplementary Information (e.g., Supplementary Table 2). Furthermore, we also point out that all data to perform a success conversion (global DEMs and global MDT product) has been publicly available since 2004 (MDT), well before the past 15 years of which we have evaluated literature, hence the issue is strictly related to proper application, not spatio-temporal unavailability of data.

Moreover, we focus in our assessment on the methodology and description of the data processing, expecting scientific rigor to provide full documentation and data description to enable the reproduction of a study, a universal scientific cornerstone which has not changed over the past decades, nor should be different in different contexts. By focussing solely on data description and methodological processing steps in the evaluated papers, we are confident that it is justified to compare and weight the different studies along the same scale of assessment (main methodological evaluation criteria, see Fig. 1). In this revision, we have further detailed the code description of the different evaluation categories, provide concrete examples of this detailing in the main manuscript and for the evaluated study-specific aspects we provide the additional information per evaluated study in an additional column in the evaluation spreadsheet.

Advances in modelling approaches have improved uncertainty handling, but the use of newer models alone does not automatically invalidate earlier work. Similarly, the finding that about 60% of the reviewed studies lacked documentation on datum conversion is indeed a methodological shortcoming, but it does not necessarily render their findings invalid. This should be highlighted in the paper

We understand the comment here and we realised that our initial labelling of the findings (e.g. in Fig.1) did not provide enough context and explanation and could be interpreted differently.

To elaborate on this, it is important to note that a datum conversion is not common practice in the community nor exists of a simple standard processing step. A vertical datum conversion always involves additional specialised data (less widespread) in addition to coastal elevation and sea level (e.g. geoid, offset between geoid(s) and tidal level) as well as expert knowledge on conversion and a procedure to process this data, including a certain software environment. We deem it very unlikely that authors who are aware of the importance of vertical datums and vertical datum conversion and go through the entire procedure of conversion end up not writing anything on vertical datum type, additional data they used to make the conversion or the processing steps they followed in their papers and/or supplementary information/data. Therefore, we assume that the absence of any documentation of those datasets or calculation steps in a manuscript is equivalent to the absence of datum conversion. And for each instance, we checked the results of such a paper, the results presented (maps, statistics, etc.) provided us with confirmation that this assumption is correct. Note that creating hard proof on this confirmation would require to reproduce each specific assessment entirely, and likely even multiple times as the absence of data documentation leaves a multitude of potential options. Naturally this is outside of our capacity and the responsibility of the original authors. Note that in the revised version we do compare a number of studies from different error categories directly to our results (Supplementary Table 5), and every time we made a comparison to a paper in the group with datum conversion omitted, our assumption was confirmed.

To emphasise this for the reader, we have added explicit explanation of this assumption to the main text of the manuscript. In addition, we have revised and extended the labelling of the different categories in the revised version, making the categorisation more elaborate and explicit (see new Fig. 1). Throughout the entire revised manuscript, we ensured that the writing correctly reflects the nuances between hard findings versus the assumed findings and thereby hope to rule out any inconsistencies or misunderstandings which stemmed from the initial manuscript.

The review rightly highlights challenges posed by data scarcity, lack of methodological clarity, transparency, and reproducibility, which can lead to underestimation of SLR. It also raises a critical concern about applying Global North models (developed and tested in the data-rich regions) in Global South contexts without adequate adaptation to local contexts and dynamics. This practice can lead to serious misinterpretations and poorly informed decisions and deserves closer scrutiny in future research.

We are happy that the referee agrees with these potential issues regarding Global-North-to-South transferability and in the revised manuscript we have added an explicit relating example to further substantiate and call for more future attention.

Main text: *“The fact that geoid models are developed and performing relatively well in reflecting local sea-level height in the Global North (e.g., USA, Western Europe) may perhaps explain the overconfidence placed in geoid-model performance by Global North-based scientists when performing assessments on global scale or in other regions of the world where geoid models perform less well. While we did not further investigate nor tested this hypothesis, we raise this to call for further scrutiny on the Global North to South transferability of scientific approaches and datasets in future research. A noteworthy example of incompatible North–South transferability occurred with CoastalDEM v1.1 (Kulp and Strauss, 2018), of which the neural-network approach used to create the CoastalDEM was Global North-trained (USA) and -validated (Australia), but performed considerably worse elsewhere. The approach, for example, places half of the entire Mekong Delta already well below present-day sea level, thereby greatly overestimating consequent population exposure to high-water levels in the region (Supplementary Fig. 19).”*

As mentioned in the conclusion section, while this review identifies significant gaps, completely dismissing earlier studies, and, by implication, questioning the integrity of the peer-review process in general seems inappropriate. A more balanced approach would be to advocate for systems that enhance assessments and projections with localized, real-time data, and to emphasize the importance of addressing methodological weaknesses, such as lack of clarity, uncertainty quantification, and reproducibility, issues that can have far-reaching consequences for decision-making and policy. The conclusion would be better framed as identifying current gaps and forward looking to improvements that could be made by authors, reviewers, editors etc. This would create a more useful and positive outcome of the work.

Thank you for these suggestions to shape the manuscript more towards a positive outlook. We agree and have reconstructed our argumentation throughout the paper to be more constructive and forward-looking. We now also highlight early on in the introduction the outlook with concrete recommendations to improve community research practices and future mitigation of errors in coastal hazard assessments (e.g. through guided peer-review...). In addition, we also added recognition to earlier pioneering works and partial steps in the right direction to highlight progress and first signs of community awareness.

Finally, while presenting strong criticisms, the review does not clarify whether the assessed studies acknowledged their own methodological and contextual limitations. Recognizing such boundaries reflects scientific integrity and transparency, helping to guide informed decisions

based on the findings. If some of the reviewed studies included such disclaimers, this should be acknowledged, as it fosters constructive dialogue and progress in the field.

We have only come across two instances (Edmonds et al., 2020; Nicholls et al., 2021) in which authors wrote single sentences in the discussion reflecting on the use of a geoid as reference frame for sea level height. It is noteworthy that these statements were stand alone as single line in the discussion and not integrated or reflected upon in the methodological design of the study (perhaps because the statement was added at a later stage to the paper in response to a reviewer comment?). We have added the following sentence on this to the main manuscript: *“The most prominent issue encountered in the evaluated literature (demonstrably present in 25% and presumably present in 63% more; Figs. 1 and 2; Supplementary Fig. 3) was the neglect of datum conversion from geoid (in some cases even ellipsoid) to a sea-level reference, thereby implicitly assuming a geoid height of 0m to match local sea-level height. Of all literature containing this issue, we only encountered two papers that reflected on the potential discrepancy between geoid and actual sea level^{39,65}.”*

In addition, we have now also included a paragraph which highlights the early signs of community awareness and progress/improvement of practices in the field: *“This group of literature also contains few studies that pioneered the use of MDT data to create a sea-level reference^{28,29,31,45}, indicating first signs of community awareness on the necessity for correct land–sea-level alignment. While these studies arguable improved on the bulk of geoid-based studies neglecting sea-level alignment, they suffer from conversion documentation shortcomings and demonstrated³¹ incorrect datum conversion (Supplementary Fig. 8).”*

One final question for the authors: Do you identify any limitations in your review that may have influenced your findings or conclusions? It be worthwhile to include a separate section discussing these limitations

Thank you for this question. Our main limitations in our review may come along with the choice of the Scopus search engine, in which several studies that met the selection criteria were not included (e.g. Haasnoot et al., 2021, which was raised by referee #1). However, Scopus constitutes one of the most transparent search engines. Moreover, which we also highlight in the manuscript now, is that we do not aim to provide a review of the full body of literature but to create an independently reproducible and representative review to perform our analysis unbiased. To check for potential Scopus-engine bias in the review, we have evaluated an additional, non-systematically collected literature dataset, in which we included studies we encountered that fit the criteria but did not show up in Scopus. This independent dataset of 96 papers (90 after excluding publications by the authors) provided near identical results (see Supplementary Fig. 16) which strengthened our belief that our systematic review

provides representative results that reflect the larger literature body. We have included the above reasoning and aspects in the revised manuscript.

Another literal limitation is the fact that we only evaluated scientific peer-reviewed articles while many more other studies such as (non-)governmental reports and consultancy assessments were not include, while they do show the same errors (see the additionally reviewed literature in Supplementary Data 1). Therefore, the literature we evaluate in this paper is likely just the tip of the iceberg, of which we took every measure possible to ensure the tip represents the body below the water line well: *“This study likely reveals only the tip of the iceberg, as the evaluated publications just form a representative selection of the entire volume of coastal hazards assessments.”*

Methodological comments/questions:

The methodology is comprehensively described and there is good supporting documentation to enable transparency which give confidence in the results. There are however, a few additional details and clarifications that are needed

Could you elaborate on whether and how you accounted for possible biases associated with the study period and geographic location of the included studies in your meta-analysis?

These factors may have influenced the outcomes and warrant discussion.

Thank you for your evaluation and confidence placed in our findings. In response to potential biases from study period and/or geographic location, the methodological evaluation we performed for all reviewed studies is independent from a geographical location or setting. In addition, the required data (global DEM + global MDT – which both can be downscaled to any region of interest) to perform the conversion correctly has been publicly available since 2004 – well before our period of assessment (last 15 years). It is true that more recent studies have access to more recent data, which may influence one of the four evaluation categories (i.e. actuality of sea level reference). To avoid any bias because of this effect, we have accounted for in the review process (see Supplementary Table 2, sea-level reference) by cross-checking the publication submission date with available data 3 months prior to this date.

Regional differences in the observed sea level discrepancy are a result of varying data quality and accuracy (e.g. geoid uncertainty), which is resulting from differences in gravitational data availability, most notably between the Global North and South. This we describe and discuss in more detail in the updated main manuscript.

Please see also our earlier response here in the above section: *Reflection on the key message - strong criticism on the reviewed papers*

Since the literature review was conducted following the PRISMA guidelines, it is recommended to include a PRISMA flow chart to clearly illustrate the screening and selection process.

Thank you, we agree. Originally, we included a Sankey Diagram that entails all the information of the literature screening and selection process (see Supplementary Fig. 1), and we have now revised the figure's layout to meet the typical layout of a PRISMA flow chart.

The methods section indicates inclusion of review papers alongside peer-reviewed articles and data papers to narrow the dataset after 2008. However, review papers are usually excluded in systematic reviews to prevent double counting of primary research. A recommended approach is to extract and include the primary studies referenced in those reviews. This could also help reduce the volume of documents and possibly enable expanding the publication timeframe, unless the cutoff at 2008 is specifically justified. Could you please elaborate on the 2008 cutoff and why this was chosen?

We agree with the reviewer that usually review articles should not be included in a review. However, we included them in the screening as we observed that some papers were categorised as review papers while at the same time conducting their own coastal impact assessment in addition to the review. Therefore, we included review papers that did their own data calculations, while classic review papers without primary computations were excluded in the next screening steps. See also the following inclusion/exclusion criteria in Supplementary Table 1: Inclusion criterion "*Review articles with primary data calculation*", exclusion criterion "*Review articles without primary data calculation*". The 2008 cut-off was selected to enable a leveled playing field to compare the studies. While global elevation models have been available for several decades, there were no global products of sea-level height available until the first global mean dynamic topography (MDT) dataset became available in 2004, introducing a new era of sea level height understanding and data availability. As it takes time for people to get to know new data and realise its potential, we decided on adding a four-year gap to ensure the knowledge on data availability and its potential could spread throughout the scientific communities. As such we selected 2009 as a starting point for our analysis.

Since SCOPUS was used as the only search engine, it would be helpful to provide justification for this choice. Considering that other databases and repositories exist, please explain how SCOPUS sufficiently captures the breadth of relevant literature for your study.

Thank you for pointing this out. The Scopus search engine was used as it constitutes one of the most transparent search engines, but we are aware that the engine does not include all literature available (see also response above). This is not a problem as, which we also highlight in the manuscript now, we do not aim to provide a review of the full body of literature

but to create an independently reproducible and representative literature dataset to perform our unbiased analysis. To check for potential Scopus-engine bias in our review, we have evaluated an additional, non-systematically collected literature dataset, in which we included studies we encountered that fit the criteria but did not show up in our Scopus. This independent dataset of 96 papers (90 after excluding publications by the authors) provided nearly identical results (see Supplementary Fig. 16) which strengthened our belief that our systematic review provides representative results that reflect the larger literature body. We have included the above reasoning and aspects in the revised manuscript.

For the additionally reviewed literature provided in Supplementary Data 1, we also included literature obtained from ISI Web of Science and Google Scholar. However, these search engines are less transparent as they offer less selection and filter categories and, in case of Google Scholar, do not allow for 100% replicable analysis. We included a few lines on this in the Methods section.

The review process, evaluation procedures, and the codes used are clearly and appropriately described; however, there are some issues that need correction. For example, the inclusion criterion “focus on coastal land and coastal sea level” is also listed under the exclusion criteria, which is contradictory. This needs to be changed and clarified

We fully agree with the reviewer and apologise for this (copy-paste) error that occurred during the final layouting of Supplementary Table 1. Naturally this is an inclusion criterion only. We corrected the error and removed ‘focus on coastal land and coastal sea level’ from the exclusion criteria. In addition, we have elaborated other criteria further to increase their clarity.

The use of “maybe” during the screening of titles, abstracts, and keywords is appropriate, as uncertain cases can be passed on to full-text analysis for detailed review. However, it should be stated in the paper that the maybes were carried forward for full text review due to ambiguity

Changed as suggested. We added a sentence in the Methods section.

Line 83, the sub-title needs to be improved as “Most coastal hazard assessments contain errors or are not reproducible” or otherwise!

Changed as suggested.

Line 103 Is 14% ample? how was this judged? I would avoid this descriptor and just state that 14% of evaluated studies were from Africa

Changed as suggested.

Line 131-132, the statement begins with a numeral (%), which is not generally recommended, and the statement from line 132 - 137 is quite long and it would be good to break it down for more clarity.

The figure capture was rewritten as suggested.

Line 201, the sub-title "Sea level higher than previously assumed in most coastal hazard assessments" is almost same as the main title, thus, may need some kind of rephrasing to specifically align it with the specific contents of the sub-section, or change the general title of the paper (as also suggested above) to reflect the entire content of the paper

Thank you for pointing this out. We agree. The sub-title (and others) has been rephrased to better fit the respective section(s).

Line 218, change the word "effected" to "affected" to make it more formal.

Changed as suggested.

In the meta-analysis, it would be advisable to provide standard errors or confidence intervals (if possible) when discussing the average estimates of the coastal sea level values

In the revised manuscript, we considerably extended our meta-analysis by adding two additional population datasets. All statistics including standard deviations are reported in the supplementary data and we also report additional statistical information where relevant directly in the main manuscript text.

Line 240, figure 4 are the grey outlines unable to be calculated? This needs stating

All areas are included in the calculations and the grey shading colour was chosen to outline the regions and subregions and provide contrast to the continent outlines to maximise the readability of the figure. We added further explanation in the figure captions of both figures 3 and 4 and also updated Supplementary Fig. 15 which indicates the study areas.

Line 319 If the guidelines were followed this should be identified as a systematic review, i.e. a systematic adhering to the standards of the Preferred Reporting Items for Systematic Reviews and Meta-Analyses (PRISMA) process was conducted

Changed as suggested.

Line 328 some examples of the exclusion of certain subject areas, keywords here would be useful and need full detail in supplementary

We included some examples of exclusion/inclusion criteria in the Methods section and give the full lists in Supplementary Fig. 1 and Supplementary Table 1.

Line 334 Who undertook the screening? Both authors? Initials should be provided, i.e. screen was undertaken by KS. Was consistency assessed? Which would be best practice and if so how was it judged. Details are needed on this

For consistency, all screening and evaluation of the literature was conducted by KS based on the priority (jointly) defined inclusion/exclusion criteria. PSJM performed an independent double screening for a subset (1000 references) during the first screening, for which no discrepancies in categorisation were encountered. We added this information in the Methods section.

349 Again, who conducted the evaluation and how was consistency ensured if there was more than one person undertaking this?

See response above. We added this information in the Methods section.

Line 350, Could you clarify the purpose of including the >90 papers accessed outside the search engine? As they were not incorporated into the statistical analysis, it would be helpful to either justify their inclusion and identify how they were used or omit this information to avoid confusion.

Thank you for this remark for further clarification. According to our knowledge, Scopus is among the most transparent and reproducible literature search engines. However, we wanted to investigate whether we would come to the same results beyond the Scopus research and included literature that we found via ISI Web of Science and Google Scholar by applying the same key words and selection criteria. We used it to evaluate certain key publications not present in the review (e.g. Haasnoot et al., 2021 which is an important IPCC-referenced assessment) and also evaluated several scientific and (non-)governmental reports to check for the presence of vertical datum conversion and respective documentation. We used the independent, non-systematically collected additional literature to independently validate the results of the methodological findings of the systematic review. We came to comparable results as for the PRISMA-guided review, which strengthened our belief that the SCOPUS dataset provides an unbiased representative selection of the existing literature. We did not include the results of the literature evaluated into the presentation of the results from the literature obtained from the systematic Scopus research, to ensure the reproducibility of the main review. The Supplementary Information provides a separate Sankey Diagram for the additionally evaluated literature (Supplementary Fig. 16).

In the supplementary under the exclusion criteria it is stated that $n = 326$ (4.5%) met the scope “less well” how was well and less well determined? This seems strange, either the papers met the criteria, did not, or it was unclear. At 1st screening perhaps it is better

phrased as unclear and then explained that these were carried through to 2nd screening. Is this the same as the “maybe” papers mentioned in the main document? If so then please align the terminology

We thank the reviewer for pointing this out and agree is confusion. We revised the terminology to ‘maybe’ to make it consistent with the terminology used in Supplementary Data 1.

Vertical datum documentation evaluation - how was completeness judged? What documentation did you need to see for this to be judged complete? Needs to be precise

We define vertical datum documentation as complete when the name of the vertical datum is provided or a reference cited that documents the name of the vertical datum of the dataset used. We added this specification in Supplementary Table 2: *“Documentation on the vertical datum is complete if the evaluated publication itself provides the name of the vertical datum (e.g. EGM96) or if it cites a reference that documents the name of the vertical datum of the dataset used.”*. We define vertical datum conversion as correct when all datasets used are converted to the same vertical reference frame and datum conversion was conducted correctly, including all steps needed to convert all datasets used to a common vertical datum. This requires that vertical datum conversion is described. We added this specification in Supplementary Table 2: *“All datasets used (including topography, bathymetry and sea-level datasets) are converted to the same vertical reference frame (e.g. EGM96, NAVD88) and datum conversion was conducted correctly, including all steps needed to convert all datasets used to a common vertical datum.”*.

Finally, to address some minor editorials, it is advisable to strictly follow the journal guidelines while revising the manuscript. For example, the captions and the main text are mixed up in most cases, which obscures the flow and clarity, and it is also good to clearly indicate the results and/or discussion sections!

We have made sure to strictly followed the author guidelines as given online on the Nature website. Note that indeed the captions are in the same format as the main text, which may indeed create some unclarity. We formatted the captions in italics to be distinguishable from the text.

Referee #4 (Remarks on code availability)

The code is reported openly and transparently. A few details highlighted above are needed to report the methodology conducted more transparently e.g. how was completeness judged

We thank the reviewer for this feedback and trust we have addressed this in the revised version.

Referee #5 (Remarks to the Author)

I co-reviewed this manuscript with one of the reviewers who provided the listed reports.

We thank the reviewer for the invested time and for the constructive comments and suggestions which have further strengthened our manuscript.

Referee #5 (Remarks on code availability)

The codes used to evaluate and synthesize the reviewed literature are included in the supplementary materials to ensure reproducibility. A few additional comments are noted in the reviewer report I co-reviewed for this paper.

Thank you!

Authors note: Additional revisions made in the revised version:

- The discussion on errors and uncertainties made us realise that it is important to explicitly make a difference between error and uncertainties in assessment of elevation on the one hand and uncertainties in impact assessments using elevation data on the other hand. Here, we consider the issues related to (incomplete, incorrect or absent) datum conversion as an error (or bias/offset) and not as an uncertainty. The datum conversion is separate from existing, and frequently studied, uncertainties related to elevation data in general. Datum conversion itself is not an uncertainty in relative elevation assessment but it does create an additional uncertainty in derivative elevation *impact* assessments. We highlight this point in the manuscript.
- We updated several figures in the main manuscript. In Figure 1, we updated the labelling of the different categories to better represent and convey the findings. We updated Figure 2 to ensure that the portrayed data matched the description in the caption of the figure. We updated figure 4 to include also the number of people currently already below sea level, so enable a clearer visualisation of the additional 1m relative sea-level rise.
- We considerably extended the supplementary data files following the addition of a large number of new analyses using additional population datasets, added the newly released GOCO2025 geoid, added quantification of potential residual errors stemming from a geoid conversion errors (EGM96 vs. EGM-DIR R4), and added the evaluation

procedure and results of the reviewed literature included as references the AR6 IPCC reports, including direct statistical comparisons to our meta-analyses to quantify the potential errors in the existing assessments.

Response to Reviews

We are grateful to all reviewers for their thoughtful and constructive comments, which have helped improve the quality of the manuscript and thank you for the time and effort invested in (re-)reviewing our manuscript.

Referee #1 (Remarks to the Author)

I thank the authors for their serious consideration of my feedback. My overall assessment remains the same: I think this article is certainly an important technical contribution, but I am not entirely convinced of the broader import of this gap. Thus, I think it is an editorial judgement call as to whether this paper is a good fit for a journal like Nature, or would be better suited for a journal like Nature Geoscience.

We thank the referee for the critical assessment of our manuscript which we considered in the revision of the manuscript. We trust the Editorial Board of the Nature journal to refer our manuscript to the most appropriate journal.

For example, in response to reviewers' recommendations, the authors now include sensitivity analyses with respect to different population data sets. For studies that do not correctly convert from the geoid to MDT, the problem is certainly large: across DEMs and population data sets, the number of additional people exposed to 1 m of SLR grows from 34-49 million to 77-132 million from correcting this error: clearly a first order error.

We fully agree with the referee. The fact that the majority of existing coastal hazard and impact assessments (>90 %) suffer from this first order error, as the referee calls it, is one of the key messages of our paper.

For studies with geoid conversion errors, the case is less convincing: if I am interpreting the results correctly, the number of people grows from 69-108 million to 77-122 million: not insignificant, but clearly a second order error compared to uncertainty in population distribution.

As extensively discussed in our manuscript, indeed the global-average magnitude of the error introduced into coastal impact assessments is less for studies with geoid conversion errors (9%) than for the earlier mentioned studies (90%) without any conversion. While the global average statistics highlighted by the referee show relatively moderate yet significant differences, a more problematic issue is not captured by these averages. Specifically, local differences can be very large due to geoid-specific variations, but these extremes are averaged out in global statistics. This means that, in addition to the discussed global increase in people and area exposed, there are also major shifts in local exposure. Significant portions of the population identified as exposed in one assessment may no longer be classified as such in

another, due to changes in spatial distribution. While this effect is not apparent in global statistics, it can be substantial at the local scale and highly impactful for regional and local impact assessments—the very level at which coastal elevation data is typically applied and assessments are used to inform coastal adaptation. An example of an encountered geoid conversion error can be found in Extended Data Figure 9 while its impacts become visible on local scale such as shown for the Ganges-Brahmaputra-Meghna, Ayeyarwady, Mekong and Red River deltas (Extended Data Figure 6). We discuss these issues in the manuscript.

Regardless, I do think the authors need to be more precise in their critique of the literature and statements about implications. For example, they write: "It is concerning that many of the evaluated studies are used to underpin IPCC SLR impact reports (1,19,45,53)." Since the IPCC does not produce 'SLR impact reports,' I was curious as to what they meant by this. Ref. 1 and 53 are to the entire of the IPCC AR6 Synthesis Report and Working Group 1 report. The Working Group 1 report does not assess impacts, so I am unclear as to the relevance.

We thank the referee for raising this point and apologise for our imprecise wording of "IPCC SLR impact reports". We agree that the IPCC does not produce 'SLR impact reports' but refers to SLR impact reports from the available literature. We checked our manuscript for any such imprecise statement and specified it.

The Synthesis Report does state "The low-lying coastal zone is currently home to around 896 million people (nearly 11% of the 2020 global population), projected to reach more than one billion by 2050 across all five SSPs." This is the sole relevant sentence, so I am not sure I would call these a "SLR impact report" either. This sentence is based upon the AR6 WG2 Cities by the Sea CCP, which in turn cites Haasnoot et al. (2021) and the IPCC SROCC.

We agree with the reviewer that this statement from the Synthesis Report is not a SLR impact report. We corrected the imprecise wording and ensured adequate consideration of the statement from the Synthesis Report which refers to population exposure in the low-elevation coastal zone.

It seems to me very unlikely that the WG2 statement "In 2020, almost 11% of the global population—896 million people—resided in C&S within the low-elevation coastal zone (LECZ; coastal areas below 10 m of elevation above sea level that are hydrologically connected to the sea; Haasnoot et al., 2021b), a figure which will potentially increase beyond 1 billion by 2050 (Oppenheimer et al., 2019)" will have a substantially different policy import than the authors' implied proposed replacement, "In 2020, about 12-14% of the global population—970 million to 1.07 billion people—resided in C&S within the low-elevation coastal zone (LECZ; coastal areas below 10 m of elevation above sea level that are hydrologically connected to the sea), a figure which will potentially increase beyond 1.XX billion by 2050." Notably, this is not a formal

assessment statement, which would be marked by use of formal confidence/likelihood language; rather, it provides context for this CCP. However, the statement does merit correction based on the authors' analysis, and the authors' analysis does indicate that a formal assessment of this topic would be warranted.

In our study we evaluated the extents to which vertical datum conversion omissions, errors and imperfections are present in the existing scientific literature. With our meta-analysis we quantified the potential impacts such errors and imperfections have on coastal impact and exposure assessments. The additional evaluation to what extent the investigated literature is referenced in the IPCC AR6 WG I–III and SROCC, provides a first-order assessment on potential far-reaching implications of our findings. One of these implications may be, as also pointed out by the referee, that certain WG2 statements can be corrected (however, we prefer a forward-looking approach and suggest uptake into future IPCC reports rather than correcting earlier ones). But apart from implications on general global statistics on exposure, as mentioned in the response before, we believe that most implications lie at regional and local scale assessments, or global climate risk rankings, as they may significantly change and may be used to underpin direct adaptation investments or fund allocations. Of course it is well out of the scope of this study to evaluate such implications and we agree with the referee that formal assessments of such implications of this study are warranted based on our findings.

Refs. 19 and 45, meanwhile, are not IPCC reports at all.

We agree with the referee and specified the imprecise wording in our manuscript accordingly.

So, in short, I think this is an important technical contribution that merits publication. I am concerned that the writing may be sensationalized for publication in Nature. The key question with respect to IPCC, for example, is not simply whether a paper cited by IPCC contains the errors that the authors identified, but the extent to which correcting the errors would influence the resulting impact assessment (or, even beyond that, the extent to which correcting these errors would influence policy or L&D negotiations). For the authors' strongest claims of import to hold, they need to show that more than one contextual, non-assessment sentence is at stake. Perhaps it is: knowing the context in which each of the references the authors are analyzing are used by AR6, and which if any assessment statements they support, would be key to interpreting this.

We acknowledge the referee's feedback but highlight that the goals of our paper are (i) to unravel whether and how the existing, contemporary literature handles sea-level height and land elevation information to infer sea-level height relative to coastal elevation, and (ii) to characterise to what extent the identified errors and imperfections affect relative sea-level rise impact and coastal exposure assessments in the world. Evaluating to what extent the

investigated literature is referenced in the IPCC AR6 WG I–III and SROCC serves as a first, tentative investigation on the potentially far-reaching implications our findings may have beyond the directly affected assessments. In contrast, understanding the implications of our findings to the full extent is another assessment we deem as a necessary follow-up investigation. However, as the referee’s comments also imply, one that will require another methodological framework, which goes well beyond the scope of this study. We further emphasise that nowhere in the manuscript we make hard claims but always use possibility-focused language when we mention potential downstream implications, e.g. related to IPCC reports. The full implications remain to be investigated and substantiated by follow-up studies, as we aimed to carefully point out in the manuscript.

Referee #2 (Remarks to the Author)

Re-review of: “Sea level much higher than assumed in most coastal hazards assessments” by Katharina Seeger and Philip Minderhoud.

I commend the authors on their responsiveness to review comments, many of which required additional analyses and/or substantive additions and references to the manuscript to validate their statements and conclusions. I also appreciate the more moderated tone of the piece as it now has the potential to serve as a more constructive contribution to the scientific community as to the importance of proper inclusion and handling of datum transformations.

We thank the referee for the positive feedback on our revised manuscript.

While the overhaul is very thorough, and I was glad to see the detailed responses and changes made in response to my feedback (and that of others) such that I feel they are sufficiently addressed, the manuscript now requires some editing to streamline and tighten the piece. As I read, I found many examples of typos, passive voice, and generally awkward grammar and phrasing that I found distracting from the message they are trying to convey.

We thank the referee for pointing this out. We revised the manuscript carefully in order to correct typos, use more active voice and smooth and clear phrasing.

For example, the second sentence in the abstract (line 12) starts with “Their...” as written, it is unclear to the new reader what this was referring to in the previous sentence, sea-level rise and other hazards, or elevation and land cover (I’m aware it is the latter; a new reader might not be). Line 22 the sentences reads “potentially necessitating the accelerated implementation of coastal adaptation strategies” which is both awkward as written and a step too far; I think what the authors intend is that the findings show that more people could be in harm’s way. And

although I think the message is better in the sentences starting on line 86, this could be tightened considerably, thereby strengthening the message and impact. The sentences on lines 365-372 provide other examples of awkward/cumbersome phrasing that could be made more succinct. To address these concerns, I suggest that in addition to perhaps recruiting a reader with a keen editorial eye, that the authors evaluate sentences in the paragraph bodies carefully for clarity, as well as adding some paragraph breaks (there are many instances where a single paragraph spans nearly two full pages, for example lines 199-250) to ensure the messaging is as clear, direct, and succinct as possible to maximize their message (and the impact of the piece).

Thank you for highlighting these textual issues. It is correct that our findings reveal that more people are in harm's way than earlier assessments are showing. Moreover, we suspect that the methodological issues we uncover are far more widespread than only the scientific community, as non-academic studies often follow the scientific standards in methodological approach. We have numerous examples (although not systematically reviewed) that they are also present in a lot of existing non-scientific assessments such as consultancy and NGO reports, vulnerability assessments by international banks and funding agencies (Worldbank, ADB etc.) and governmental assessments in areas lacking local data. These assessments, harbouring the same potential magnitude of error as the scientific studies, may be even much more harmful, as they are closer to application or directly informing coastal adaptation strategies (e.g. funding and implementation of coastal protection) and decision-making (e.g. evaluating peoples' exposure to coastal flooding). As such we prefer to keep the mentioning of these potential implications (although not proven nor easily quantifiable) in the starting paragraph. We have carefully considered the phrasing and ensured to use possibility-focused wording when addressing the potential downstream implications. In addition, we carefully revised the highlighted sentences, included paragraph breaks for readability and checked the wording in the manuscript to exclude any remaining unnatural wording or stylistic weaknesses.

The authors should also pay particular attention to how they describe "complete documentation" vs. "correct conversion and adjustment" such as in figure 1; there is nuance that is captured in the text that gets at this (they describe further in the paper how with incomplete documentation they are able to infer where conversation might be correct/incorrect), but stating this more clearly at the outset would further strengthen their argument.

We appreciate the referee's detailed feedback and agree on the relevance of clarifying the nuance in the wording. We went through our manuscript and especially the introductory part to ensure that this nuancing becomes clear from the beginning throughout the text.

In addition, the caption for Figure 1, the Sankey diagram, needs to be revised to include explainers on what the different categorizations in the figure represent. Although the surrounding text goes into some detail, there are elements of the figure that need to be specifically and clearly defined, and the caption is an appropriate place to do this. For example, what does a sea-level reference with a correct implementation (correct implementation with respect to what) and reference up to date (current?) mean exactly?

We thank the referee for this feedback and adjusted the caption of Figure 1 accordingly.

Again, the authors are commended on the considerable work they have put into revision; it is an impressive effort and good to see how it has evolved.

We appreciate the referee's feedback on the revisions made.

Referee #2 (Remarks on code availability)

I reviewed the code the first time around; as it doesn't appear this was revised, I have not re-reviewed.

The referee is correct that we did not revise the codes. We have now moved the codes for the two model workflows from Supplementary Data 2 and 3 to the Yoda data repository and added the related DOI in the Code Availability Statement.

Referee #3 (Remarks to the Author)

The authors have provided thorough and detailed responses to the comments provided by the reviewers. Overall, the manuscript has improved a lot based on the extensive revisions. The methodological approach of the review and its limitations are now described in a much clearer manner. The language is more nuanced and less accusatory. The assumptions about omissions in the studies reviewed are now laid out more clearly. Most importantly, the additional text around the interpretation and sense-making of the findings is very helpful. Particularly, the additional analysis with a view towards IPCC AR6 is helpful. Also, the limits of the approach are now more clearly discussed. The deeper dive into the Mekong Delta case study supports to illustrate the relevance of the findings. The additional discussion of future steps the research community can take is also of value. The new charts in the appendix add clarity and illustration and hence strengthen the overall contribution.

We appreciate the referee's positive feedback on the revisions we made to our manuscript.

I have a few minor comments the authors still might need to consider in order to improve the manuscript further:

We thank the referee for providing explicit comments on specific passages of our manuscript. We have considered them carefully and provide a point-by-point response below.

- The choice of the Mekong Delta case study needs to be explained more thoroughly. Is being the third largest delta globally really the only – and most convincing – reason? Or can the discrepancy in sea level and impact effects be shown most clearly here? If so, how does VMD compare roughly to other regions?

Thank you for this feedback, we see the need to further justify the Mekong Delta case study. Not only is the Mekong Delta one of the largest flat, low-lying coastal landscapes in the world. It is also densely populated and economically important while being elevated only a few decimetres above sea level and therefore highly exposed to relative sea-level rise (Minderhoud et al., 2019; <https://doi.org/10.1038/s41467-019-11602-1>). Besides being located in the geoid–MDT offset ‘hotspot’ of SE Asia (indeed, the discrepancy is large here), it also has local elevation data available which can serve as a benchmark to compare with global satellite-derived DEMs. The combination of all these factors makes the Mekong Delta an ideal example to investigate more in-depth uncertainties (see also Seeger and Minderhoud, 2025; <https://doi.org/10.21203/rs.3.rs-7706762/v1>). We added further explanation in the main text to clarify the suitability of the Mekong Delta as case study in our global meta-analysis accordingly.

References:

P. S. J. Minderhoud, L. Coumou, G. Erkens, H. Middelkoop, E. Stouthamer, Mekong delta much lower than previously assumed in sea-level rise impact assessments. *Nat. Commun.* **10**, 3847 (2019). <https://doi.org/10.1038/s41467-019-11602-1>.

K. Seeger, P. S. J. Minderhoud, Attributing uncertainties in elevation assessments for data-sparse coastal lowlands using global elevation models: A globally applicable approach showcasing the Vietnamese Mekong Delta. 30 October 2025, PREPRINT (Version 1) available at *Research Square*. <https://doi.org/10.21203/rs.3.rs-7706762/v1>.

- In Figure 4, the sea level height lines (assumed vs. corrected) are not very clear and should be made more visible. In the same figure, the global results (bottom left) might be more clearly provided as numbers rather than fractions of the person icons. Overall, numbers in addition to the icons might be helpful.

We agree. We revised Figure 4 and replaced the sea-level height lines by mannequins. We tried adding numbers but this overloaded the figure with information and reduced clarity,

therefore we choose not to include those and trust the new layout does a better job in conveying the message. We did add a reference in the caption to the related Supplementary Data, where all numbers shown in the figure can be found.

- The terms omnipresent and omnipotent should be avoided. For the former, formulations such as “typical” or “common” would be better.

We checked the manuscript and replaced the words ‘omnipresent’ and ‘omnipotent’ as suggested by the referee.

Referee #4 (Remarks to the Author)

We thank the authors for providing a thorough, point-by-point response document to our previous comments. This gives more justifications and perspectives which is helpful and well explained and/or resolved with modifications. Below we outline specific responses to the key comments/modifications:

We appreciate the referee’s feedback on the revisions we made to our manuscript.

Regarding the title, it is clear that detailed thought went in to this and there appears to be strong rebuttal and justification for approach. However, we are not our specific field of expertise and so defer to other reviews who are.

We thank the referee for the feedback on the title of our manuscript. Given the outcomes of the literature evaluation combined with the global meta-analysis, as well as the feedback from all other referees and the editor, we deem the title of our manuscript to be suitable.

We are happy to see the argumentation in the paper is more “moderate, less criticising and nuanced” presentation of the findings by the investigated literature using the latest, at that time, available elevation and sea-level data

We thank the referee for this feedback.

Happy that the assumptions made regarding the absence of any documentation or calculation steps in a manuscript is equivalent to the absence of datum conversion has been made more explicit

We thank the referee for this feedback.

Happy more on the need for global-south research highlighted more in revised manuscript

We thank the referee for this feedback.

Happy for more positive, forward-looking discussion regarding improving community research practices, as well as this being highlighted in the conclusion

We thank the referee for this feedback.

Happy limitations of the review are highlighted more thoroughly in revised manuscript

We thank the referee for this feedback.

Happy that a PRISMA diagram has now been added - However, the inclusion/exclusion criteria text in the figure is just a repeat, it would be better if figures for the reasons for exclusion could be added for transparency

We thank the referee for this suggestion. We agree that numerical reporting of exclusion reasons would improve transparency. However, during screening many studies often fell under a lot of exclusion criteria and detailed counts per individual exclusion criterion were not recorded. Assigning counts in retrospect would risk misrepresenting the screening decisions. To ensure transparency, all exclusion criteria are fully reported, and total inclusion and exclusion numbers are shown at each screening stage. We have clarified this limitation in the caption of the PRISMA diagram. All additional information on the literature review is given in the Supplementary Information and Supplementary Data 1.

Happy for clarification on use of review papers in this review, this is a sensible approach

We thank the referee for this feedback.

Including the supplementary papers as a way of checking for one database/Scopus bias is unconventional but seems adequate, happy to see the approach detailed in the methods and the additional records provided as supplementary material and the limitation of this mentioned in the discussion. We believe that including another database at this stage is a lot of extra work that would probably not alter the results and so is not necessary as long as the limitations of this approach are highlighted, which the authors have now done.

Thank you for agreeing with our, perhaps unconventional, approach. We also believe that adding another database would not significantly alter the results and tried to carefully describe the limitations of our approach.

Changes to the description of the inclusion/exclusion criteria have been made satisfactorily, as well as other methodological details/concerns

We thank the referee for this feedback.

Overall, it is clear a lot of work has been done in this review, and the authors have addressed the comments thoroughly and provided clear justifications in the response document and

manuscript. However, we must defer to subject experts regarding the boldness of the claim and the appropriateness of the title.

We thank the referee for this feedback and appreciate the openness and trust shown towards the other referees to assess the appropriateness of the title of our manuscript.

Referee #5 (Remarks to the Author)

I co-reviewed this manuscript with one of the reviewers who provided the listed reports.

We thank the referee for contributing constructive feedback by co-reviewing.

Referee #5 (Remarks on code availability)

The codes used to evaluate and synthesize the reviewed literature are included, detailed, and described in the supplementary materials.

We thank the reviewer for the thorough assessment on the code availability and the positive feedback.